# Shape Coexistence in Even–Even Nuclei: A Theoretical Overview

**Dennis Bonatsos** [1,*] , **Andriana Martinou** [1] , **Spyridon K. Peroulis** [1] , **Theodoros J. Mertzimekis** [2] **and Nikolay Minkov** [3]

1   Institute of Nuclear and Particle Physics, National Centre for Scientific Research "Demokritos",
    15310 Aghia Paraskevi, Attiki, Greece; martinou@inp.demokritos.gr (A.M.); s.peroulis@yahoo.gr (S.K.P.)
2   Department of Physics, Zografou Campus, National and Kapodistrian University of Athens,
    15784 Athens, Greece; tmertzi@phys.uoa.gr
3   Institute of Nuclear Research and Nuclear Energy, Bulgarian Academy of Sciences, 72 Tzarigrad Road,
    1784 Sofia, Bulgaria; nminkov@inrne.bas.bg
*   Correspondence: bonat@inp.demokritos.gr

**Abstract:** The last decade has seen a rapid growth in our understanding of the microscopic origins of shape coexistence, assisted by the new data provided by the modern radioactive ion beam facilities built worldwide. Islands of the nuclear chart in which shape coexistence can occur have been identified, and the different microscopic particle–hole excitation mechanisms leading to neutron-induced or proton-induced shape coexistence have been clarified. The relation of shape coexistence to the islands of inversion, appearing in light nuclei, to the new spin-aligned phase appearing in $N = Z$ nuclei, as well as to shape/phase transitions occurring in medium mass and heavy nuclei, has been understood. In the present review, these developments are considered within the shell-model and mean-field approaches, as well as by symmetry methods. In addition, based on systematics of data, as well as on symmetry considerations, quantitative rules are developed, predicting regions in which shape coexistence can appear, as a possible guide for further experimental efforts that can help in improving our understanding of the details of the nucleon–nucleon interaction, as well as of its modifications occurring far from stability.

**Keywords:** shape coexistence; shape phase transitions; islands of inversion; shell model; mean field; SU(3) symmetry; dual shell mechanism



## 1. Introduction

Shape coexistence (SC) in atomic nuclei refers to the special case in which the ground state band of a nucleus lies close in energy with another band possessing a radically different structure. SC has been suggested in 1956 by Morinaga [1], in order to interpret $0^+$ states corresponding to deformed shapes lying close in energy with the spherical ground state bands of $^{16}$O, $^{20}$Ne, and $^{24}$Mg. Many more examples of SC have been suggested since then. Manifestations of SC in odd-mass nuclei have been reviewed in 1983 by Heyde et al. [2], while cases observed in even–even nuclei have been reviewed in 1992 by Wood et al. [3]. A review of both experimental observations and relevant theoretical developments has been given in 2011 by Heyde and Wood [4]; see also [5] for an account of the historical development of the subject, as well as the volume [6,7] for a compilation of recent work. The advent of radioactive ion beam facilities in recent years largely increased the number of nuclei available for experimental investigation. Experimental manifestations of SC have been recently (2022) reviewed by Garrett et al. [8]. In parallel, the progress made in theoretical calculations, both in the mean-field and the shell-model frameworks, has accumulated many new results over the last decade, including massive calculations over entire series of isotopes, or even large regions of the nuclear chart involving several series of isotopes. It is the scope of the present review to summarize these recent theoretical developments, draw conclusions from the existing results and pave the way towards future developments.

Nobel prizes in nuclear structure have been awarded to Wigner, Goeppert-Mayer, and Jensen in 1963 [9], as well as to Bohr, Mottelson, and Rainwater in 1975 [10], corresponding to breakthrough developments of our understanding of atomic nuclei. Wigner was rewarded for the use of symmetries in nuclear structure, Goeppert-Mayer and Jensen for the introduction of the spherical shell model, and Bohr, Mottelson, and Rainwater for the introduction of the collective model describing nuclear deformation. Recent developments along all these three lines will be considered in the present review, which will be limited to even–even nuclei, for the sake of size.

### 1.1. Statement of Purpose

The present review considers SC of the ground state band with $K^\pi = 0^+$ bands lying close in energy and having radically different structures. The term SC has also been used in the literature to refer to pairs and/or bunches of high-lying bands (see, for example, [11–18]), isomeric states (see, for example, [19–24]), superdeformed bands (see, for example, [25–27]), or bands of negative parity, related to the octupole degree of freedom (see, for example, [28–31]). Such bands are not considered here.

The main purpose of the present review is to focus attention on the many theoretical advances made in the description of SC since the last authoritative review, by Heyde and Wood in 2011 [4]. Since the experimental work on SC has been recently reviewed by Garrett et al. in 2022 [8], our emphasis has been placed on the theoretical developments, with a number of experimental papers cited either for historical purposes or thanks to the theoretical calculations contained in them.

The progress made on the theoretical front of SC over the last decade has been tremendous. The advent of both theoretical methods and computational facilities already allows for systematic calculations over extended regions of the nuclear chart using the same set of model parameters for all nuclei, in contrast to old calculations in which individual fits of the parameters had to be made for each nucleus separately. This extends the predictive power of the models beyond the experimentally known nuclei, thus providing predictions which can guide further experimental efforts.

The present work has been focused on spectra and B(E2) transition rates, as well as on potential energy surfaces (PESs). A review of additional experimental quantities (transfer cross sections, decay properties, charge radii) related to the presence of SC has been given recently by Garrett et al. [8]. It should be emphasized that PESs alone do not suffice for establishing SC, since they do not contain all the necessary information, which has to be supplied by spectra, B(E2)s, and/or other experimental quantities.

For the sake of simplicity and space, only even–even nuclei are considered in this review. Relevant literature has been considered up to December 2022, with few exceptions. A similar review on odd-A nuclei is called for.

### 1.2. Outline

In the first part (Section 2) of this article, the theoretical methods used for the description of even–even nuclei are briefly reviewed. The purpose of this part is to introduce the language needed and the basic concepts used, practically using no equations. The readers are directed to the relevant authoritative reviews for technical details. The acronyms used throughout this review are listed in Abbreviations, while the theoretical methods used are listed in Appendix A, along with the section in which they are defined. Some additional methods are also briefly described in Appendix A.

In the second part of this article (Sections 3–12), the series of isotopes from Rn ($Z = 86$) to Be ($Z = 4$) are studied separately. An effort has been made to keep the study of each series of isotopes self-contained, at the expense of some repetition. Important figures from the articles considered are not reproduced in the body of this work, in order to keep its size finite. Detailed citations are given instead, so that the interested reader can easily reach them in this era of easy access to articles in an additional window on the computer in use. A brief summary is presented at the end of each subsection, regarding the relevant series

of isotopes. One can choose to consider only the atomic numbers of their own interest, without any need to go through the rest. The only sections in which the properties of several series of isotopes are discussed together are Sections 5.5, 8.5, and 9.5, in which the relation between SC and SPT is discussed in the regions ($N = 90, Z = 60$), ($N = 60, Z = 40$), and ($N = 40, Z = 34$), respectively, as well as the Sections 11 and 12, in which $N = Z$ nuclei and the islands of inversion are discussed, respectively.

In the third part (Sections 13–15) of this article, the information presented in the summaries of the second part is gathered together and discussed in Section 13, in order to reach the final conclusions of the present study in Section 14, as well as to make proposals for further work in Section 15.

## 2. Theoretical Approaches to Shape Coexistence

### 2.1. The Nuclear Shell Model

The nuclear shell model [32–37] was introduced in 1949 in order to explain the appearance of the magic numbers 2, 8, 20, 28, 50, 82, 126, ..., associated with nuclei exhibiting extreme stability. It is based on the three-dimensional isotropic harmonic oscillator (3D-HO), which possesses the SU(3) symmetry and is characterized by closed shells at 2, 8, 20, 40, 70, 112, 168, ... protons or neutrons [38–40]. To the 3D-HO potential the spin–orbit interaction is added, which destroys the SU(3) symmetry beyond the sd nuclear shell, leading from the 3D-HO magic numbers 2, 8, 20, 40, 70, 112, 168, ... to the spherical shell-model magic numbers 2, 8, 20, 28, 50, 82, 126, .... The spectroscopic notation, labeling the shells with 0, 1, 2, 3, 4, 5, 6, 7, ... oscillator quanta by the letters s, p, d, f, g, h, i, j, ... is used. The Woods–Saxon potential [41] can be used alternatively. It may be noticed that the idea of a shell model for atomic nuclei based on the 3D-HO has already existed since the 1930s [42–44], but it was the crucial addition of the spin–orbit interaction [32,34–36] to it, which led to the successful prediction of the magic energy gaps.

In this primitive version of the shell model, also called the independent particle model, degenerate energy levels within each shell occur. In order to break the degeneracies, one has to consider that in addition to the two-body nucleon–nucleon interaction creating the spherical mean field, a "residual" two-body interaction, appearing when two or more nucleons are present in the same shell, has to be taken into account. As a consequence, the computational effort needed for diagonalizing the relevant Hamiltonian increases dramatically with the size of the shell.

A first compromise in order to reduce the size of the calculation is to limit it to the valence protons and neutrons outside closed shells, with the latter being treated as an inert core. Thus, calculations have to use a limited valence space and an appropriate effective interaction adapted to it. The interaction of the valence nucleons with the core is included in the "residual" interaction. A review of these developments has been given by Caurier et al. [45], while a review of the shell-model applications on SC can be found in [6,46]. Applications of the shell-model approach to specific nuclei [47–51] will be discussed in the relevant sections regarding the corresponding series of isotopes.

Computational limitations have restricted the application of the shell model to light- and medium-mass nuclei near to closed shells. A major step towards large-scale shell-model calculations has been taken by the use of the quantum Monte Carlo technique [52–54], which led to the formulation of the Monte Carlo shell model (MCSM) [55–57] scheme. Within this scheme, the size of the calculation is radically reduced by selecting out of the huge number of many-body basis states a much smaller number of highly selected many-body basis states, in what can be considered as an "importance truncation" [58]. The question of uncertainty quantification, which is of obvious importance for predictions in experimentally unknown regions, has also been considered [59] within this approach. Applications of this technique to specific nuclei, including state-of-the-art calculations over extended series of isotopes, can be found in [58,60–75]. These will be discussed in the relevant sections regarding the corresponding series of isotopes.

An alternative path has been taken through no-core shell-model calculations [76,77]. Within this scheme, one is trying to reduce the size of the calculation by taking advantage of the underlying SU(3) symmetry of the 3D-HO, thus ending up with a symmetry-adapted no-core shell model (SANCSM) [78–82], which will be further discussed in the SU(3) Section 2.4.

Within the shell-model framework, SC can be accounted for by two-particle–two-hole (2p-2h) excitations across major shells [83–87]. In a nucleus lying near to closed shells, the excitation of two protons or two neutrons above the Fermi level effectively adds two more valence pairs (a pair of particles above the Fermi surface and a pair of holes below the Fermi surface), thus largely increasing the collectivity of the nucleus, which is based on the numbers of valence protons and valence neutrons outside closed shells. In even–even nuclei, 2p-2h and 4p-4h excitations are the most common ones. Mixing [88–90] of the ground state configuration (which has no p-h excitations) and the excited configuration leads then to the two coexisting bands.

The last statement is the basis of the two-state mixing model (see Section 2.2 of [8]), widely used [88–101] by experimentalists in order to directly detect SC from the data. The two-state mixing model is based on the assumption that two observed states bearing the same angular momentum and parity can be understood as linear combinations of two unperturbed states, with coefficients (mixing amplitudes) depending on a mixing angle, which is a measure of the occurring mixing and can be determined from the spectroscopic data.

### 2.1.1. The Tensor Force

A major step forward in understanding shell evolution in exotic nuclei was made by Otsuka et al. in 2005 [102] with the introduction of the tensor force, based on the exchange of $\pi$ and $\rho$ mesons, in the framework of the meson-exchange process introduced by Yukawa [103] in 1935. The main effect caused by the tensor force is the influence of protons on the neutron gaps, as well as the influence of neutrons on the proton gaps. Using for the nuclear orbitals the symbols $j_> = l + 1/2$ and $j_< = l - 1/2$, it turns out that the interaction between orbitals with different $>, <$ subscripts is attractive, while the interaction between orbitals with the same subscript is repulsive.

Let us see how the tensor force influences the shell gaps through an example [102]. Below the $Z = 28$ gap one finds the proton orbital $1f_{7/2>}$, while above it the proton orbital $1f_{5/2<}$ is lying. Lower than both of them in energy, the neutron orbital $1g_{9/2>}$ lies, since the neutron potential is deeper than the proton potential, see, for example, Figure 6.4 of [104]. The tensor force between $1g_{9/2>}$ and $1f_{7/2>}$ is repulsive, thus $1f_{7/2>}$ is pushed up in energy by the tensor component of the proton–neutron interaction. On the contrary, the tensor force between $1g_{9/2>}$ and $1f_{5/2<}$ is attractive, thus $1f_{5/2<}$ is pushed down in energy by the tensor component of the proton–neutron interaction. As a result, the two proton orbitals are approaching each other, and therefore the $Z = 28$ gap gets reduced, thus making the creation of proton particle–hole excitations across the $Z = 28$ gap easier and therefore facilitating the appearance of SC [105].

Recent reviews on the properties of the tensor force and its role in the occurrence of SC and in shell evolution can be found in [105–107], respectively. The terms *type I shell evolution*, corresponding to shell evolution along a series of isotopes or isotones, and *type II shell evolution*, depending on the balance of the central and tensor components of the effective nucleon–nucleon interaction, and influencing the appearance of SC, have been coined [105]. The shape of the tensor force as a function of the nuclear radius is given for various interaction models in Figure 3 of [102], as well as in Figure 15 of [107]. It is interesting to compare the shape of the tensor force due to the $\pi$ and $\rho$ meson exchange to the shape of the central force, see Figure 1 of [64]. While the central force has a Gaussian shape with minimum at $r = 0$, the tensor force has a hard core and a minimum at $r > 0$, roughly resembling the Kratzer potential [108].

### 2.2. Self-Consistent Mean-Field Models

In the shell-model approach one assumes that the nuclear mean field has the form of a specific single-particle potential and then performs configuration mixing involving all possible states appearing in the area of its application, called the shell-model space. An alternative approach is to start with the self-consistent determination of a nuclear mean field fitted to nuclear structure data [109]. The simplest variational method for the determination of the wave functions of a many-body system is the Hartree–Fock (HF) method [110,111]. In the case of nuclei, the pairing interaction has to be taken into account, leading into the Hartree–Fock–Bogoliubov (HFB)approach [110,111]. The inclusion of pairing is simplified by the BCS approximation, which allows the formation of pairs of degenerate states connected through time reversal, as in the BCS theory of superconductivity in solids [112]. Three different choices have been made for the effective nucleon–nucleon interaction: the Gogny interaction [113–115] and the Skyrme interaction [116–118], used in a non-relativistic framework, and the relativistic mean-field (RMF) models [119–128], the latter having the advantage of providing a natural explanation of the spin–orbit force. A compilation of various parametrizations of the effective interactions in use can be found in Table I of the review article [109]. Extensive codes have been published [129], facilitating the use of RMF all over the nuclear chart.

Applications of the Gogny interaction to specific nuclei can be found in [130–140], while applications of the Skyrme interaction to specific nuclei can be found in [69,133,141–148]. Applications of the RMF method to specific nuclei can be found in [149–171]. These will be discussed in the relevant sections regarding the corresponding series of isotopes.

The self-consistent mean-field approach used in nuclear structure [172,173] strongly resembles the density functional theory used for the description of many-electron systems [174,175]. The main difference is that in electronic systems one can derive very accurately the energy density functionals by *ab initio* methods from the theory of electron gas, while in nuclear structure the effective energy density functionals are motivated by *ab initio* theory [176], but they contain free parameters fitted to the nuclear structure data.

The simplest case of mean-field methods is the static mean-field approach, in which the lowest order nuclear binding energy is obtained [109]. In order to describe a larger set of data, correlations have to be included, in what is called the *beyond-mean-field* approach. SC in the beyond-mean-field method has been considered, for example, in [131,136–138,148,177,178]. The most important correlations are due to large amplitude collective motion, and are dealt by the *generator coordinate method*, which is a variational method in which the single-particle and collective nuclear dynamics are considered within a single coherent quantum mechanical formulation (see [109] and references therein). An alternative approach is to consider a much larger time-dependent (HFB) (TDHFB) phase space and extract from it a collective submanifold. This TDHFB approach [179] has been used in several light and medium-mass nuclei exhibiting SC [180,181].

A non-relativistic approach widely used [22,182–196] for the study of nuclei exhibiting SC is the excited VAMPIR [197]. This is an (HFB) approach with spin and number projection before variation, extended to non-yrast states.

In addition, a five-dimensional quadrupole collective Hamiltonian (5DQCH) has been derived [198] microscopically from large amplitude collective motion and applied to several nuclei exhibiting SC [199–205].

The relativistic self-consistent mean-field approaches determine the ground state properties of nuclei. A method allowing the calculation of excitation spectra and B(E2) transition rates has been introduced by Otsuka and Nomura [206–210]. The parameters of the Interacting Boson Model (IBM) [211–215] are determined by mapping the total energy surface obtained from the RMF approach onto the total energy surface determined by IBM. One is then free to use the IBM Hamiltonian with these microscopically derived parameters for the calculation of spectra and electromagnetic transition rates. This method has already been applied to wide regions of the nuclear chart exhibiting SC [216–230].

In parallel, a 5DQCH with parameters microscopically determined from relativistic density functional theory [124,231–237] has also been used for the study of extended regions of the nuclear chart in which SC appears [238–244].

*2.3. Nuclear Deformation and the Nuclear Collective Model*

The primordial shell model introduced by Mayer and Jensen in 1949 [32–37] did explain the nuclear magic numbers, but it failed to explain the large nuclear quadrupole moments measured away from closed shells. It was soon realized by Rainwater [245] in 1950 that a spheroidal nuclear shape was energetically favored over the spherical one, offering greater stability to the nucleus. The collective model of Bohr and Mottelson [110,246–249] was then introduced in 1952, allowing the description of deformed nuclear shapes in terms of the collective variables $\beta$ (describing the departure from sphericity) and $\gamma$ (an angle representing the departure from axial symmetry).

The realization of the importance of nuclear deformation immediately led to the introduction of a deformed shell model by Nilsson [104,250,251] in 1955, in which an axially deformed three-dimensional harmonic oscillator is employed, characterized by a deformation parameter $\epsilon$, expressing the departure from isotropy. The plots of the single-particle energies of the protons and of the neutrons as a function of the deformation parameter $\epsilon$, known as the Nilsson diagrams [252], have accompanied experimental efforts for understanding the microscopic structure of nuclei for several decades, becoming an integral part of the nuclear terminology.

Soon thereafter the pairing-plus-quadrupole (PPQ) model of Kumar and Baranger was formulated [253–255] in 1965, incorporating the two most important features of the "residual" interaction, namely the pairing effects due to nuclear superfluidity [256] and the deformation effects due to the quadrupole–quadrupole interaction. Despite the fact that the true "residual" interaction might be much more complicated in form, the success of the PPQ model suggests that these two terms offer a very good approximate manifestation of the real interaction.

The importance of nuclear pairing has been demonstrated by Talmi [37,257–261] in 1962, through the introduction of the seniority quantum number $v$, which counts nucleon pairs in a nuclear shell not coupled to zero angular momentum. The seniority scheme is best manifested in semi-magic nuclei, having a magic number of protons and therefore only neutrons as valence particles. It provided clear explanations of the linear dependence of neutron separation energies on the mass number in the semi-magic Ca ($Z = 20$), Ni ($Z = 28$), and Sn ($Z = 50$) series of isotopes, as well as of the nearly constant excitation energies of the $2^+$ states in the even–even Sn isotopes.

The importance of the quadrupole–quadrupole interaction has been most dramatically demonstrated by Casten in 1985 through the $N_p N_n$ scheme [262–265], in which $N_p$ ($N_n$) is the number of valence protons (neutrons) measured from the nearest closed shell. The $N_p N_n$ scheme demonstrates that each nuclear observable exhibits a smooth behavior as a function of $N_p N_n$, independent of the region in which the nucleus sits in the nuclear chart.

A measure of nuclear collectivity, expressing the relative strength of the quadrupole–quadrupole interaction over the pairing interaction is the *P*-factor [265,266]

$$P = \frac{N_p N_n}{N_p + N_n}. \tag{1}$$

Deformed nuclei possess $P > 4$–5, while nuclei with $P < 4$ cannot be deformed. In relation to SC, if we consider a nucleus with $N_p = 4$, $N_n = 6$, we obtain $P = 2.4$. Assuming 2p-2h excitations in the protons, we are going to effectively have $N_p = 8$, $N_n = 6$, yielding $P = 3.43$ , which is much higher than 2.4.

Another, easy to use, measure of nuclear collectivity is the ratio

$$R_{4/2} = \frac{E(4_1^+)}{E(2_1^+)} \tag{2}$$

of the first two excited states of the ground state band of an even–even nucleus, which obtains values 2.0–2.4 for vibrational (near-spherical) nuclei, 2.4–3.0 for intermediate $\gamma$-unstable nuclei, and 3.0–3.33 for rotational (deformed) nuclei. For excited $K = 0$ bands, the ratio takes the form $R_{4/2} = (E(4) - E(0))/(E(2) - E(0))$.

The proton-neutron interaction of the last proton and neutron in an even–even nucleus can be obtained from double differences of binding energies [267–277]

$$\delta V_{pn}(Z, N) = \frac{1}{4}[\{B(Z, N) - B(Z, N - 2)\} - \{B(Z - 2, N) - B(Z - 2, N - 2)\}], \quad (3)$$

where $B(Z, N)$ stands for the binding energy of the nucleus consisting of $Z$ protons and $N$ neutrons. In the notation of the Nilsson model, in which orbitals are labeled by $K[NN_z\Lambda]$, where $N$ is the total number of oscillator quanta, $N_z$ is the number of quanta along the axis $z$ of cylindrical symmetry, and $K$ ($\Lambda$) is the projection of the total (orbital) angular momentum on the symmetry axis, it has been found [268,273,274,278] that the proton–neutron pairs exhibiting maximum interaction are characterized by Nilsson labels differing by $\Delta K[\Delta N \Delta N_z \Delta\Lambda] = 0[110]$, which are called 0[110] Nilsson pairs.

The Bohr collective Hamiltonian with a sextic potential possessing two minima has been used [279–284] for the description of SC of spherical and prolate deformed shapes. A similar effort regarding SC of spherical and $\gamma$-unsable shapes has also been made [285]. More general potentials possessing two minima have been considered within the Bohr collective Hamiltonian in [286–288], while oblate–prolate SC has been considered in [289].

*2.4. The SU(3) Symmetry*

2.4.1. The SU(3) Symmetry of Elliott

A bridge between the microscopic picture of the shell model and the macroscopic picture of the collective model of Bohr and Mottelson has been provided in 1958 by El-liott [290–294], who proved that nuclear deformation naturally occurs within the sd shell of the shell model, which is characterized by a U(6) algebra possessing an SU(3) subalgebra, generated by the angular momentum and quadrupole operators [38,40,295]. Elliott's papers stimulated extensive work on the development of SU(3) techniques appropriate for the description of deformed nuclei, recently reviewed by Kota [296], as well as on the use of algebraic techniques for the description of nuclear structure in general [297]. In what follows we are going to need Elliott's notation for the irreducible representations of SU(3), which is $(\lambda, \mu)$, where $\lambda = f_1 - f_2$ and $\mu = f_2$, with $f_1$ ($f_2$) being the number of boxes in the first (second) line of the relevant Young diagram.

The $p$, $sd$, $pf$, $sdg$, $pfh$, $sdgi$, $pfhj$ shells of the isotropic 3D-HO are known to possess the U(3), U(6), U(10), U(15), U(21), U(28), U(36) symmetry, respectively, each of them having an SU(3) subalgebra [39,298]. In short, the shell with $n$ oscillator quanta is characterized by a U((n + 1)(n + 2)/2) algebra. However, the SU(3) symmetry beyond the $sd$ shell is destroyed by the spin–orbit interaction, which within each major shell acts most strongly on the orbital with the highest total angular momentum $j$, pushing it down into the shell below. Since the parity of the $n$ shell is given by $(-1)^n$, each shell modified by the spin–orbit interaction ends up consisting of its original orbitals (except the one which escaped to the shell below), called the normal parity orbitals, plus the orbital invading from the shell above, which has the opposite parity and is called the abnormal parity or the intruder orbital.

2.4.2. The Pseudo-SU(3) Symmetry

A first attempt to restore the SU(3) symmetry beyond the sd shell was the development of the pseudo-SU(3) scheme [299–306], in which the normal parity orbitals remaining in each shell are mapped onto the full set of orbitals of the shell below, through a unitary transformation [307–309] introducing the quantum numbers of pseudo-angular momentum and pseudo-spin for each orbital. For example, in the 28-50 shell the Nilsson orbitals 1/2[301], 1/2[321] are mapped onto the pseudo-orbitals 1/2[200], 1/2[220] respectively, the orbitals 3/2[312], 1/2[310] are mapped onto 3/2[211], 1/2[211], and the orbitals 3/2[301],

5/2[303] are mapped onto 3/2[202], 5/2[202]. Remembering the Nilsson notation $K[NN_z\Lambda]$ and taking into account that the projection of the spin on the symmetry axis is $\Sigma = K - \Lambda$, we remark that, when making the pseudo-SU(3) transformation, $N$ is reduced by one unit (and therefore parity is changing sign), $n_z$ and $K$ remain intact, while $\Sigma$ is changing sign and $\Lambda$ is modified accordingly in order to keep the rule $\Sigma = K - \Lambda$ satisfied. Tables for the correspondence of the normal and pseudo-SU(3) levels can be found in [310,311]. In this way, the SU(3) symmetry is restored for the normal parity orbitals, thus SU(3) techniques can be applied to them, while the intruder orbitals still have to be treated by shell-model methods. The pseudo-SU(3) scheme has been very successful in describing many aspects of medium-mass and heavy nuclei, like spectra and B(E2) transition rates in even–even [303,312] and odd [313,314] nuclei, the pairing interaction [315,316], double $\beta$ decay [317,318] and superdeformation [26], in which a full-shell-model calculation would have been intractable.

### 2.4.3. The Quasi-SU(3) Symmetry

A second attempt to extend the SU(3) symmetry beyond the sd shell has been made in 1995 by Zuker et al. [319,320] with the introduction of the quasi-SU(3) symmetry. The basic idea of this symmetry is that in the $pf$ shell the intruder orbital $1g_{9/2}$, coming down from the sdg shell, does not suffice to create the necessary collectivity when combined with the normal parity orbitals $2p_{1/2}, 2p_{3/2}, 1f_{5/2}$, remaining in the $pf$ shell after the sinking of the $1f_{7/2}$ orbital to the shell below. In order to enhance collectivity, one has to include in the calculation in addition the orbitals $2d_{5/2}$ and $3s_{1/2}$ of the $sdg$ shell, thus including in the collection all orbitals of the $sdg$ shell successively differing by $\Delta j = 2$ from the $1g_{9/2}$ orbital, where $j$ stands for the total angular momentum. The $\Delta j = 2$ rule is justified by looking at the matrix elements of the quadrupole operator in L-S and j-j coupling [319]. The quasi-SU(3) symmetry has been recently extended to the $sdg$ shell [321,322], the orbitals of which $(3s_{1/2}, 2d_{3/2}, 2d_{5/2}, 1g_{7/2}, 1g_{9/2})$ should be combined not only to the $1h_{11/2}$ intruder orbital, but in addition to its $\Delta j = 2$ partner orbital $2f_{7/2}$ in order to create the desired collectivity.

### 2.4.4. The Proxy-SU(3) Symmetry

A third attempt to restore the SU(3) symmetry beyond the sd shell has been made in 2017 with the introduction of the proxy-SU(3) symmetry [323–328]. The proxy-SU(3) symmetry takes advantage of the Nilsson 0[110] pairs mentioned above, for which it is known that they correspond to orbitals having maximum spatial overlap [268]. One can then observe that in each shell the intruder orbitals (except the one with the highest $K$) are 0[110] partners of the orbitals which deserted the shell by going into the shell below. For example, in the 28–50 shell, the intruding 1g9/2 orbitals 1/2[440], 3/2[431], 5/2[422], 7/2[413] are 0[110] partners of the deserting $1f_{7/2}$ orbitals 1/2[330], 3/2[321], 5/2[312], 7/2[303], while the invading orbital 9/2[404] has no partner. 0[110] partner orbitals possess the same angular momentum and spin projections $K$, $\Lambda$, $\Sigma$, while they differ only by one quantum in the $z$-direction (and as a consequence, also by one quantum in $N$, thus having opposite parity). Because of this similarity, the intruder orbitals can act as "proxies" of the deserting orbitals, thus restoring the U(10) symmetry of the $pf$ shell and its SU(3) subalgebra. The intruder orbital with the highest value of $K$ remains alone and has to be treated separately, but, luckily enough, as one can see from the standard Nilsson diagrams [252] it is the orbital lying highest within the 28–50 shell, thus it should be empty for most nuclei within this shell. The same process, which is equivalent to a unitary transformation [329], applies to all higher shells. The accuracy of the approximation has been tested [323] by comparing the results of Nilsson calculations before and after the replacement, the outcome being that the accuracy of the approximation is improved with increasing size of the shell. The connection of the proxy-SU(3) scheme to the shell model [329,330] has proved that Nilsson 0[110] pairs correspond to $|\Delta n \Delta l \Delta j \Delta m_j\rangle = |0110\rangle$ pairs in the shell-model language, where $n$ is the number of oscillator quanta, $l$ ($j$) is the orbital (total) angular momentum, and $m_j$ is the projection of $j$ on the symmetry axis $z$.

The $|0110\rangle$ pairs of orbitals are identical to the proton–neutron pairs identified by de Shalit and Goldhaber [331] in 1953 during studies of $\beta$ transition probabilities. In particular, they observed that nucleons of one kind (neutrons, for example) have a stabilizing effect on pairs of nucleons of the other kind (neutrons in the example), thus favoring the development of nuclear deformation.

The proxy-SU(3) symmetry has been able to predict quite accurately the dependence of the deformation variable $\beta$ along several series of isotopes of medium-mass and heavy nuclei, without using any free parameters [324,326]. This has been achieved by using the relations [332,333] between the Elliott quantum numbers $\lambda$, $\mu$, characterizing the SU(3) irreps $(\lambda, \mu)$, and the collective variables $\beta$, $\gamma$ describing the nuclear shape. These relations have been obtained by establishing a mapping [332] between the invariant quantities of the rigid rotor, which are the second- and third-order Casimir operators of SU(3), $C_2(\lambda, \mu)$ and $C_3(\lambda, \mu)$ [38,40], and the invariants of the Bohr Hamiltonian, which are $\beta^2$ and $\beta^3 \cos 3\gamma$ [246,247,249]. In addition, it provides a natural explanation of the dominance of prolate over oblate shapes in the ground states of even–even nuclei [324,328], and it successfully predicts [324] the transition from prolate to oblate shapes in the rare earths around $N = 114$, again in a parameter-free way. As we shall see below in Section 2.4.6, it also provides parameter-free predictions for regions of the nuclear chart in which SC can be expected [334,335].

### 2.4.5. The SU(3) Symmetry in the Symplectic Model

An SU(3) subalgebra also exists within the symplectic model [336–338], characterized by the overall Sp(6,R) non-compact algebra (called Sp(3,R) in the papers in which it was introduced). The symplectic model is a generalization of Elliott's SU(3) model, in which many oscillator major shells are taken into account. The advantage of the model is that it allows the determination of the shell-model configurations necessary for building collective rotations, as well as quadrupole and monopole vibrations in nuclei, thus providing a bridge between the shell model and the collective model of Bohr and Mottelson.

The SU(3) symmetry plays a crucial role within the symmetry-adapted no-core shell model (SANCSM) [78–82], which takes advantage of the appearance [339,340] of the symplectic symmetry in no-core shell-model calculations [76,77], thus allowing the use of SU(3) techniques within shell-model calculations. Such calculations have been carried out up to now in light nuclei, including $^6$He, $^6$Li, and $^8$Be [341], $^{12}$C and $^{16}$O [339,340,342–344], $^{20}$O, $^{20,22,24}$Ne, and $^{20,22}$Mg [345]. Recent SANCSM calculations in $^6$Li and $^{20}$Ne have revealed the importance of a few basic equilibrium shapes (deformed or not) in producing the structure of each nucleus [346].

### 2.4.6. The Dual-Shell Mechanism

Within the framework of the proxy-SU(3) symmetry a dual-shell mechanism [334,335] has been developed, allowing for the prediction of regions in the nuclear chart, in which the appearance of SC can be expected.

The three-dimensional isotropic harmonic oscillator (3D-HO) is known [38–40] to exhibit shell structure, with closed shells appearing at the harmonic oscillator (HO) magic numbers 2, 8, 20, 40, 70, 112, 168, . . . .

On the other hand, the connection [329] of the proxy-SU(3) symmetry to the shell model shows that the addition of the spin–orbit interaction to the 3D-HO modifies the shell structure, leading to the magic numbers 6, 14, 28, 50, 82, 126, 184, . . . (see Table 7 of [329]), which for the sake of brevity we call the spin–orbit (SO) magic numbers, although in reality they are 3D-HO magic numbers modified by the spin–orbit interaction.

In spherical nuclei (i.e., in nuclei with zero deformation), the HO magic numbers appear up to the *sd* shell, while the SO magic numbers take over above it, resulting in the well known set of shell-model magic numbers 2, 8, 20, 28, 50, 82, 126, . . . [32,34–36].

As soon as deformation sets in, the magic gaps in the single-particle energy diagrams, known as the Nilsson diagrams [252], soon disappear. However, this does not mean that the

HO and/or the SO magic numbers themselves disappear. The HO and SO magic numbers are still valid: just the single particle energy levels become distorted by the interactions, leading to nuclear deformation. This is how the dual-shell mechanism emerges.

Looking into the details of the relevant Hamiltonian [334], one ends up with the conclusion that SC can appear only within certain regions of neutrons or protons, namely 7–8, 17–20, 34–40, 59–70, 96–112, 146–168, .... These regions appear as horizontal and vertical stripes on the nuclear chart (see Figure 25 of [334]). It should be emphasized that these stripes, which can be seen in Figures 1–6, express a necessary condition for the appearance of SC, and not a sufficient one.

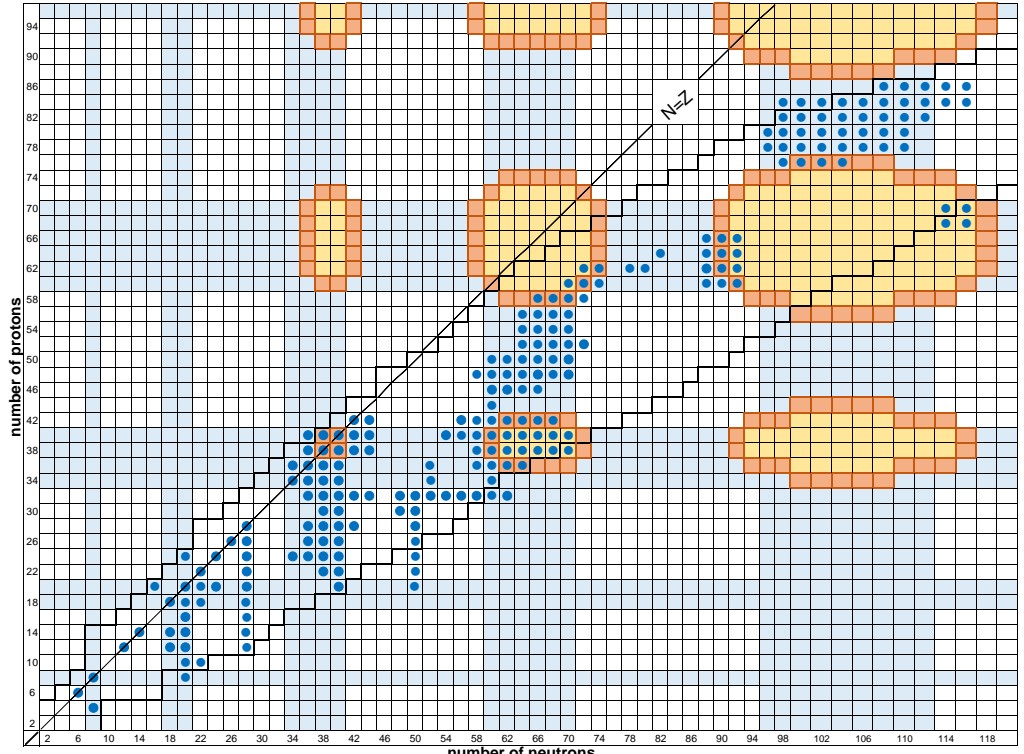

**Figure 1.** Nuclei exhibiting SC according to the evidence discussed in Section 13.1. The nuclei mentioned in the conclusion or summary of each subsection in Sections 3–10 are included. Most nuclei fall within the stripes predicted by the dual-shell mechanism [334] developed in the framework of the proxy-SU(3) symmetry [323,327], shown in azure, with notable exceptions corresponding to the islands of inversion (see Section 12) and the $N = Z$ line (see Sections 11 and 11.1). Contours corresponding to $P \approx 5$ [266,347,348] are also shown in orange, with deformed nuclei lying within them. The proton (neutron) driplines, shown as thick black lines, have been extracted from the two-proton (two-neutron) separation energies appearing in the database NuDat3.0 [349], involving standard extrapolations. See Section 13.1 for further discussion.

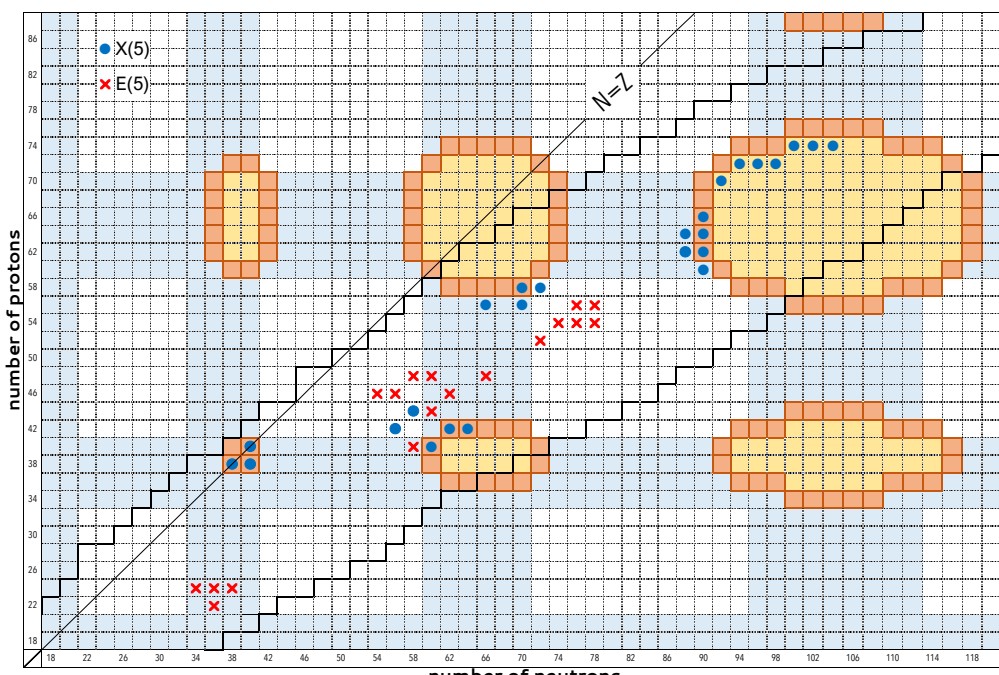

**Figure 2.** Nuclei suggested in the literature as candidates for the X(5) CPS, related to the SPT from spherical to deformed nuclear shape are shown as blue circles. The nuclei shown are $^{76}_{38}$Sr$_{38}$ [350], $^{78}_{38}$Sr$_{40}$ [350], $^{80}_{40}$Zr$_{40}$ [350], $^{100}_{40}$Zr$_{60}$ [350], $^{98}_{42}$Mo$_{56}$ [351,352], $^{104}_{42}$Mo$_{62}$ [350,351,353], $^{106}_{42}$Mo$_{64}$ [351], $^{102}_{44}$Ru$_{58}$ [352], $^{122}_{56}$Ba$_{66}$ [354,355], $^{126}_{56}$Ba$_{70}$ [356], $^{128}_{58}$Ce$_{70}$ [357], $^{130}_{58}$Ce$_{72}$ [356,358,359], $^{150}_{60}$Nd$_{90}$ [360–362], $^{150}_{62}$Sm$_{88}$ [363], $^{152}_{62}$Sm$_{90}$ [358,360,364–373], $^{152}_{64}$Gd$_{88}$ [374], $^{154}_{64}$Gd$_{90}$ [375,376], $^{156}_{66}$Dy$_{90}$ [375,377,378], $^{162}_{70}$Yb$_{92}$ [379], $^{166}_{72}$Hf$_{94}$ [379,380], $^{168}_{72}$Hf$_{96}$ [381], $^{170}_{72}$Hf$_{98}$ [382], $^{176}_{76}$Os$_{100}$ [383], $^{178}_{76}$Os$_{102}$ [383], $^{180}_{76}$Os$_{104}$ [383]. Contours corresponding to $P \approx 5$ [266,347,348] are also shown in orange, with deformed nuclei lying within them. We remark that candidates for X(5) lie on or next to the $P \approx 5$ contours. The stripes in which SC can be expected according to the dual-shell mechanism [334] developed in the framework of the proxy-SU(3) symmetry [323,327] are shown in azure. In addition, nuclei suggested in the literature as candidates for the E(5) CPS, related to the SPT from spherical to $\gamma$-unstable shape are shown as red X's, for comparison. The nuclei shown are $^{58}_{22}$Ti$_{36}$ [384], $^{58}_{24}$Cr$_{34}$ [384–387], $^{60}_{24}$Cr$_{36}$ [384], $^{62}_{24}$Cr$_{38}$ [384], $^{98}_{40}$Zr$_{58}$ [388], $^{104}_{44}$Ru$_{60}$ [389], $^{100}_{46}$Pd$_{54}$ [390], $^{102}_{46}$Pd$_{56}$ [372,391,392], $^{108}_{46}$Pd$_{62}$ [393,394], $^{106}_{48}$Cd$_{58}$ [391], $^{108}_{48}$Cd$_{60}$ [391,395], $^{114}_{48}$Cd$_{66}$ [396], $^{124}_{52}$Te$_{72}$ [391,397,398], $^{128}_{54}$Xe$_{74}$ [391,399], $^{130}_{54}$Xe$_{76}$ [400,401], $^{132}_{54}$Xe$_{78}$ [400], $^{132}_{56}$Ba$_{76}$ [402], $^{134}_{56}$Ba$_{78}$ [366,391,394,402–405]. The proton (neutron) driplines, shown as thick black lines, have been extracted from the two-proton (two-neutron) separation energies appearing in the database NuDat3.0 [349], involving standard extrapolations. See Section 13.1 for further discussion.

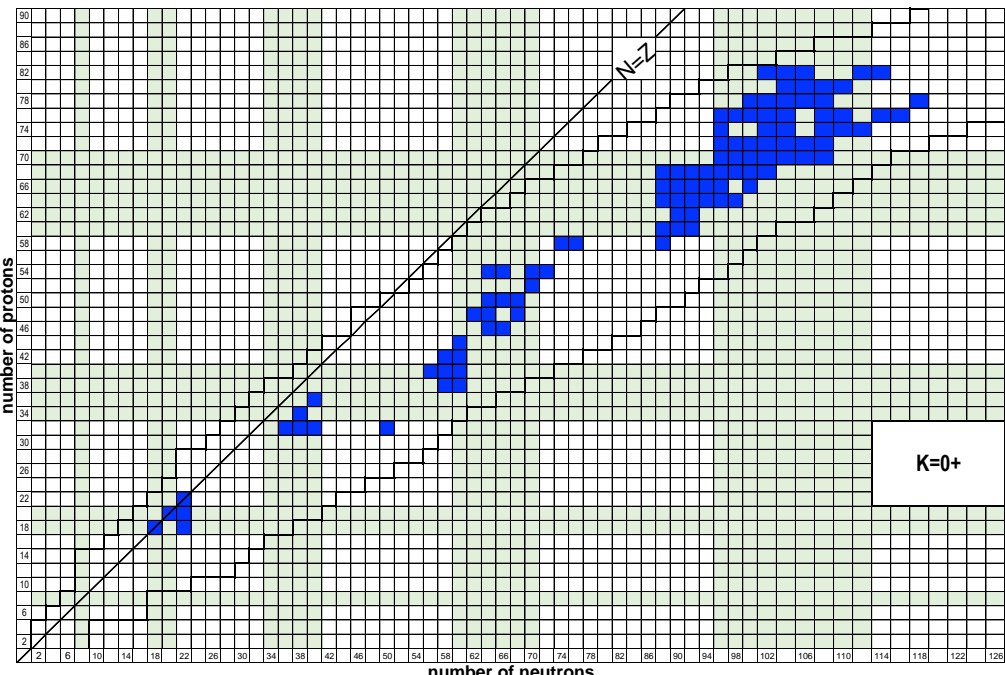

**Figure 3.** Nuclei with well-developed $K = 0$ bands (blue boxes) (listed in Table I of [335], based on data taken from Ref. [406]). The stripes within which SC can be expected according to the dual-shell mechanism [334] developed in the framework of the proxy-SU(3) symmetry [323,327] are also shown in light green. The proton (neutron) driplines, shown as thick black lines, have been extracted from the two-proton (two-neutron) separation energies appearing in the database NuDat3.0 [349], involving standard extrapolations. Adapted from [335]. See Section 13.2 for further discussion.

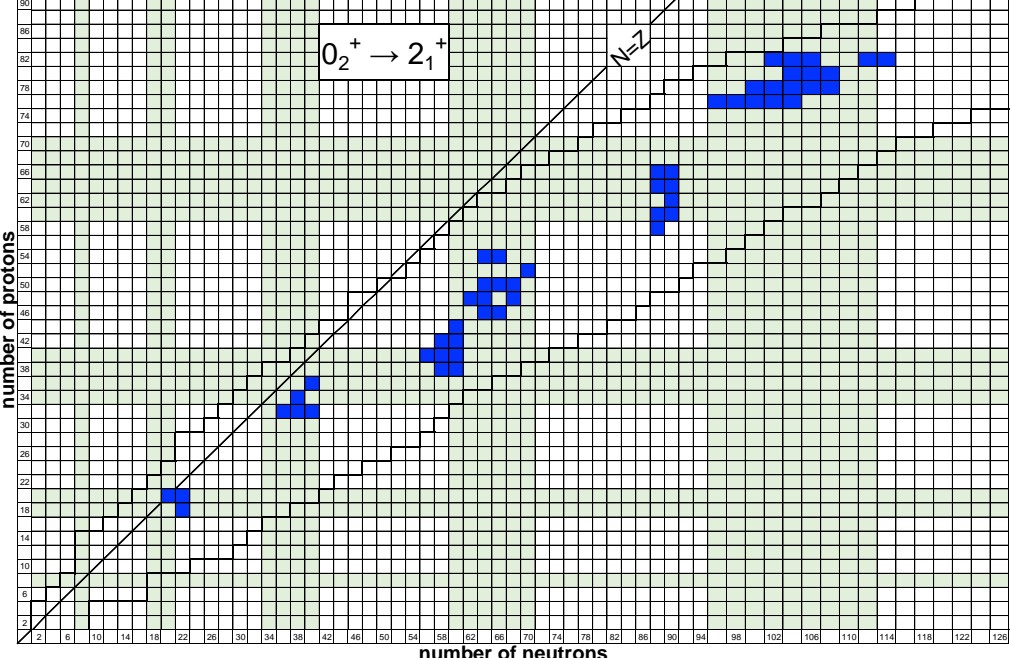

**Figure 4.** Nuclei with energy differences $E(0_2^+) - E(2_1^+)$ less that 800 keV (blue boxes), based on data taken from Ref. [406]. The stripes within which SC can be expected according to the dual-shell mechanism [334] developed in the framework of the proxy-SU(3) symmetry [323,327] are also shown in light green. The proton (neutron) driplines, shown as thick black lines, have been extracted from the two-proton (two-neutron) separation energies appearing in the database NuDat3.0 [349], involving standard extrapolations. Adapted from [335]. See Section 13.2 for further discussion.

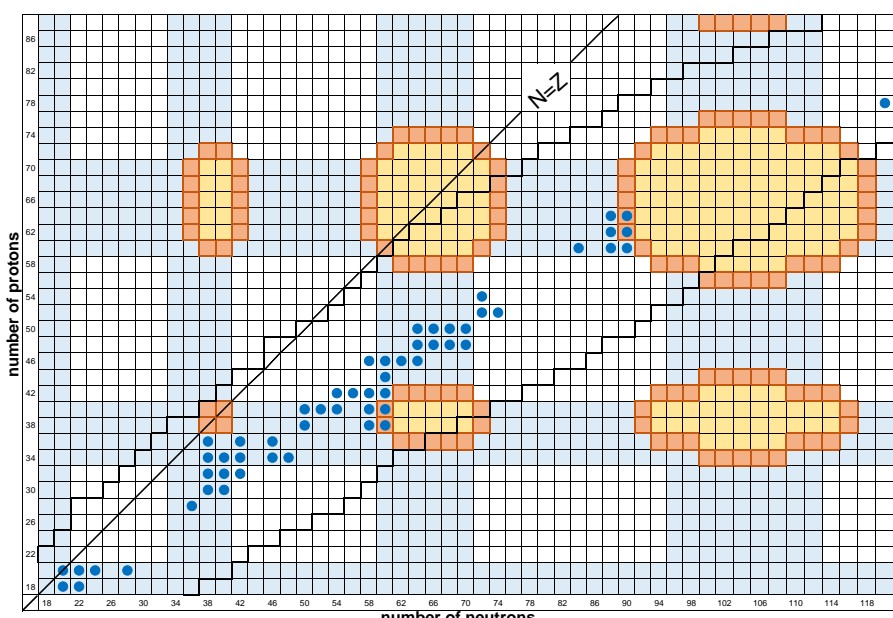

**Figure 5.** Nuclei with experimentally known [406] $B(E2; 0_2^+ \rightarrow 2_1^+)$ transition rates and $K^\pi = 0^+$ bands, satisfying the conditions $R_{4/2} < 3.05$ (Equation 2), $R_{0/2} < 5.7$ (Equation (4)), and $B_{02} > 0.1$ (Equation (5)), and known to exhibit SC according to the references listed in the relevant to each series of isotopes subsection of Sections 3–10, are shown as blue circles. The azure stripes within which SC can be expected to occur according to the dual-shell mechanism [335] developed within the framework of the proxy-SU(3) symmetry [323,327] are also shown, along with the contours (indicated by orange boxes) corresponding to $P \approx 5$ [266,347,348]. The proton (neutron) driplines, shown as thick black lines, have been extracted from the two-proton (two-neutron) separation energies appearing in the database NuDat3.0 [349], involving standard extrapolations. Adapted from [407]. See Section 13.2 for further discussion.

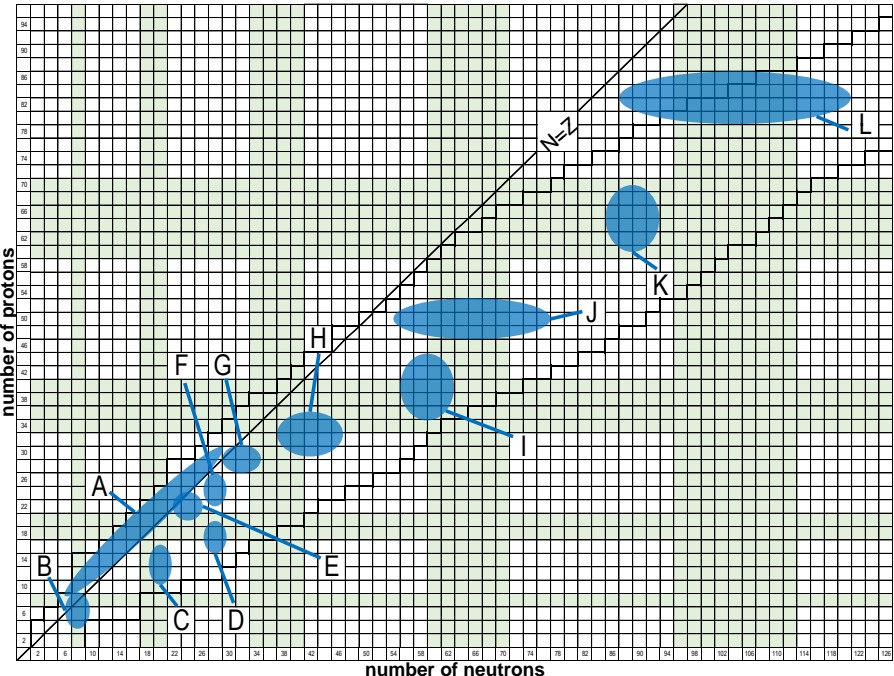

**Figure 6.** The regions of SC reported in Figure 8 of the review article [4], shown as blue ovals, are compared to light green stripes predicted by the dual-shell mechanism [335] developed within the framework of the proxy-SU(3) symmetry [323,327]. See Section 13.3 for further discussion.

### 2.5. Algebraic Models Using Bosons

A different path for reducing the size of shell-model calculations has been taken through representing correlated fermion pairs by bosons. This idea led to the introduction of the Interacting Boson Model (IBM) by Arima and Iachello in 1975 [211–215], the basic ingredients of which are *s* and *d* bosons with angular momentums of 0 and 2, respectively, a reasonable assumption since the quadrupole degree of freedom is dominant in nuclear collectivity. The bosons composing a nucleus correspond to valence fermion pairs counted from the nearest closed shell. In the simplest version of the model, labeled as IBM-1, no distinction is made between protons and neutrons. This distinction is taken into account in the version called IBM-2 or pn-IBM. Odd nuclei (not included in the present review) are described by considering a single fermion in addition to the bosons representing the even–even part of the nucleus, thus forming the Interacting Boson–Fermion Model (IBFM) [215]. Nuclear supersymmetries, describing quadruplets of even–even, even–odd, odd–even, and odd–odd nuclei by the same Hamiltonian have also been developed [215].

IBM-1 is characterized by a U(6) overall symmetry, having three different chains of subalgebras containing the angular momentum algebra SO(3), so that angular momentum can be used for labeling the nuclear energy states. Each chain of subalgebras corresponds to a different dynamical symmetry, representing different physics. The U(5) dynamical symmetry [408] is appropriate for vibrational (near spherical) nuclei appearing near to closed shells, while the SU(3) dynamical symmetry [409] corresponds to axially deformed prolate (rugby ball like) nuclei appearing away from closed shells. A variant called $\overline{SU(3)}$ corresponds to axially symmetric oblate (pancake like) nuclei. The O(6) dynamical symmetry [410] corresponds to nuclei with intermediate values of deformation, characterized as $\gamma$-unstable, since they are soft with respect to the value of the collective variable $\gamma$, thus exhibiting unstable triaxial shapes.

A convenient way to visualize the three dynamical symmetries of IBM is to place them on the vertices of a triangle [265,411,412], with each point within the triangle corresponding to a different set of parameter values of the IBM Hamiltonian.

Partial dynamical symmetries [413,414] have also been developed in the IBM framework, regarding situations in which a symmetry is obeyed by a subset of soluble eigenstates, but is not shared by the Hamiltonian. SC of spherical and deformed shapes, as well as of prolate and oblate shapes, have been predicted [415,416] within the PDS framework.

IBM-1 can describe only states with positive parity. Negative parity states can be accommodated with the addition of p and f bosons with negative parity and angular momenta of 1 and 3, respectively [417,418]. The resulting spdf-IBM can be used for the description of octupole bands appearing in the rare earth [419,420] and in the actinide [421,422] region.

SC can be described within the extended IBM framework [423–430] by allowing two-particle–two-hole (2p-2h) excitations to occur, which obviously increases the number of bosons by 2, since two additional pairs are taken into account. Mixing [431,432] of the ground state configuration of N bosons with the excited configuration of N + 2 bosons leads then to the two coexisting bands. This scheme is called the IBM with configuration mixing (IBM-CM). It has already been applied for the description of SC in several regions of the nuclear chart [433–446].

### 2.6. Shape/Phase Transitions and Critical Point Symmetries

Shape/phase transitions (SPTs) [347,348,447–451] in chains of isotopes occur when the shape of the nucleus is undergoing a radical change by the addition of two neutrons. A spectacular example is seen in the Sm series of isotopes, in which the $R_{4/2}$ ratio jumps from the near-vibrational value of 2.316 in $^{150}$Sm to the near-rotational value of 3.009 in $^{152}$Sm [406].

In the IBM a first order SPT (according to the Ehrenfest classification) is known since 1981 [411] to occur between the dynamical symmetries U(5) and SU(3) [452], while a second-order SPT is found between the dynamical symmetries U(5) and O(6) [411]. No SPT occurs between SU(3) and O(6), but O(6) has been suggested as the critical point of

the transition from prolate SU(3) shapes to oblate $\overline{SU(3)}$ shapes [453–455]. A convenient way to form a picture of these SPTs is to place them on a triangle [265,411,412], with the vertices of the triangle corresponding to the U(5), SU(3), and O(6) dynamical symmetries of IBM, and with each point within the triangle corresponding to a different set of parameter values of the IBM Hamiltonian. This triangle is doubled [453–455] by the addition of the $\overline{SU(3)}$ symmetry.

In Bohr's collective Hamiltonian, a critical point symmetry (CPS) called E(5) has been suggested by Iachello [450] in 2000 between spherical and $\gamma$-unstable nuclei, while a CPS called X(5) has been suggested by Iachello [451] in 2001 between spherical and prolate deformed nuclei. A convenient way to demonstrate these critical point symmetries is to place them on a triangle [456], the vertices of which correspond to spherical, deformed, and $\gamma$-unstable nuclei, with each point within the triangle corresponding to a different set of parameter values of the Bohr Hamiltonian. In addition, a Y(5) CPS between axial and triaxial nuclei [457] and a Z(5) CPS between prolate and oblate shapes [458] have been suggested. Fixing $\gamma$ to the value of maximum triaxiality, $\pi/6$, the CPSs X(5) and Z(5) reduce to the CPSs X(3) [459] and Z(4) [460], respectively. Equal shape/phase space mixing of the $\gamma$-stable X(5) and the $\gamma$-rigid X(3) solutions leads to the X(4) [461] CPS. A link between X(4) and Z(4) is provided by T(4) [462], while a link between E(5) and X(5) is given by F(5) [463].

The best experimental manifestations of X(5) have been found in the $N = 90$ isotones $^{150}$Nd [361,362], $^{152}$Sm [366,372], and $^{154}$Gd [375,376]. Several other candidates have been suggested, reviewed in Refs. [347,356,464], with relevant references given in the caption of Figure 2.

The only fully confirmed experimental manifestation of E(5) is found in $^{134}$Ba [402–405]. Several candidates have been suggested, reviewed in Refs. [347,391,464], with relevant references given in the caption of Figure 2.

The way the energy levels are rearranged by SPTs has been clarified in terms of quasidynamical symmetries in a series of papers by Rowe et al. [465–468].

The close connection between SPTs and SC [469] has been studied in the region around $N = 60$ [439]. Further discussion is deferred to the relevant Sections 5.5, 8.5, and 9.5.

In relation to the microscopic origin of SPTs, the crucial role played by the proton–neutron interaction in the creation of deformation in medium-mass and heavy nuclei, clarified by Federman and Pittel in 1977 [470–473], offers an answer. In light, intermediate-mass, rare earth, and actinide nuclei the $(1d_{5/2}, 1d_{3/2})$, $(1g_{9/2}, 1g_{7/2})$, $(1h_{11/2}, 1h_{9/2})$, and $(1i_{13/2}, 1i_{11/2})$ proton–neutron pairs of orbitals, respectively, play a major role at the onset of deformation, while the pairs $(1d_{5/2}, 1f_{7/2})$, $(1g_{9/2}, 1h_{11/2})$, $(1h_{11/2}, 1i_{13/2})$, and $(1i_{13/2}, 1j_{15/2})$ play a major role in the increase of deformation after its onset, respectively. Notice that the first four couples of orbitals are $|\Delta n \Delta l \Delta j \Delta m_j\rangle = |0010\rangle$ shell-model pairs, while the last 4 couples are $|0110\rangle$ shell-model pairs. Notice also that the same proton orbitals appear in both sets. In the first set they are coupled to neutron orbitals of the same parity, while in the second set they are coupled to neutron orbitals of the opposite parity, which are the orbitals of maximum $j$ in their shell, pushed down to the shell below by the spin–orbit interaction and thus becoming the intruder orbitals. In other words, the intruder neutron orbitals play a major role in the development of deformation across the nuclear chart. Therefore they are expected to be the protagonists in the regions of the sharp SPT from spherical to deformed nuclei occurring at $N = 90$, as well as at $N = 60$ and $N = 40$, and in the appearance of SC there, as we shall see in the relevant Sections 5.5, 8.5 and 9.5.

A recent major step forward in the development of CPSs in atomic nuclei has been the introduction in their study of the conformable fractional derivatives by Hammad [474] in 2021. While in the standard fractional calculus [475–477], fractional derivatives do not obey the familiar basic properties of derivatives, the conformable fractional derivatives [478] do obey them, thus offering obvious analytical advantages. Analytical models using conformable fractional derivatives contain an extra parameter, the order of the derivative, which allows closer approach to the data of critical nuclei [474,479–481].

### 2.7. The O(6) Symmetry

Since the first article by Elliott [290] in 1958, in which the description of nuclear deformation in the *sd* shell of the shell model that has an overall U(6) symmetry, in terms of the SU(3) symmetry has been introduced, it is known that an alternative formulation of the *sd* shell in terms of the O(6) symmetry is possible (see the Appendix of [290]). However, while the SU(3) subalgebra is present in all algebras $U((n+1)(n+2)/2)$ characterizing the *n*-th shell of the isotropic 3D-HO [39,40,298], this is not the case for the O(6). As a consequence, while SU(3) can be applied to heavier shells of the shell model through the use of pseudo-SU(3) and/or quasi-SU(3) and/or proxy-SU(3), no similar generalization regarding O(6) exists so far.

However, the use of O(6) in heavier nuclei is possible in the framework of the IBM [214] (see Figure 2.4 of [214] for the regions of the nuclear chart in which experimental manifestations of the O(6) dynamical symmetry have been seen).

O(6) is related to two SPTs. One of them is the second-order SPT in IBM between U(5) and O(6) [411], characterized by the common O(5) subalgebra of these two dynamical symmetries [482], and labeled by E(5) [450] in the framework of the Bohr Hamiltonian, as mentioned in the preceding section.

The other one is the SPT in IBM between prolate (rugby-ball like) and oblate (pancake-like) deformed nuclei, characterized by the dynamical symmetries SU(3) and $\overline{SU(3)}$ [453] respectively. It has been argued [453–455] that O(6) is the critical point of the SPT from prolate to oblate shapes, with relevant experimental support provided in [483]. Within the framework of the Bohr Hamiltonian the prolate-to-oblate SPT has been described by the Z(5) [458] CPS. The prolate-to-oblate transition is predicted [324] in a parameter-free way within the proxy-SU(3) symmetry.

An analytic approach to SC within the U(5) to O(6) transition has been given in [484,485], with the role played by their common O(5) subalgebra being emphasized.

It may be mentioned that SC has also been considered [486] within a simplified shell model with O(4) symmetry, a multi-shell version of which has been studied through TDHFB theory. In addition, SC has been considered [487] within a simplified model with protons in two subshells and neutrons in one subshell, with each subshell possessing the SO(8) symmetry of the Ginocchio model [488,489], which guarantees that a space of *s* and *d* pairs is closed, i.e., it does not bring in pairs of higher angular momentum.

### 2.8. The Nature of $0^+$ States

The most common manifestation of SC in even–even nuclei regards the occurrence of a $0^+$ band lying close in energy with the ground state band and having radically different structure from it, for example with one of the bands being spherical and the other one deformed. It is therefore of interest to clarify the nature of low-lying $0^+$ bands in even–even nuclei.

At least since 2001 [490] it has been clear that most of the first excited $0_2^+$ states are not $\beta$ vibrations of the Bohr–Mottelson collective model. In order to characterize a $0^+$ band as a $\beta$ vibrational band, one should have appreciable B(E2) transition rates connecting this band to the ground state band (appreciable $B(E2; 0_\beta^+ \to 2_{gs}^+)$ or $B(E2; 2_\beta^+ \to 0_{gs}^+)$. Furthermore, the $E0$ transition connecting the two bands should also be strong [491]. However, in order to unambiguously characterize a $0^+$ state as a $\beta$ vibrational band-head, the presence of anharmonicity contibutions should be taken into account.

The number of $0^+$ states in an even–even nucleus can be large. The first example found experimentally in 2002 was $^{158}$Gd, in which 13 excited $0^+$ states have been found below 3.1 MeV [492–495]. Many theoretical interpretations have been suggested for these $0^+$ states, including two-phonon octupole character [496], the pseudo-SU(3) model [497–499], the quasiparticle–phonon model [500], and the pairing plus quadrupole model [501].

Further experimental investigations found many $0^+$ states in several rare earth nuclei [502–507], and theoretical interpretations have been provided using the pseudo-SU(3) symmetry [508,509], the quasiparticle–phonon model [504], the projected shell model [504],

the pairing plus quadrupole model [510,511], as well as simple band mixing calculations [512]. Special attention has also been given to the Cd isotopes [513–515], as well as to the Sn and Te isotopes [516,517], since SC appears there, as well as to $^{152}$Sm [518] (using the multiphonon approach), since it is a textbook example of the X(5) CPS. Examples have also been found in the actinides [519], interpreted in terms of the quasiparticle–phonon model [520] and the $spdf$ version of the IBM [519].

The vibrational character of the $0_2^+$ state in $^{154}$Gd was disputed in 2008 by Sharpey-Schafer [521–525], by suggesting that this state is not a $\beta$ vibration based on the ground state, but a second vacuum, each with its own $\gamma$ and octupole vibrations. Similar arguments have been presented for the neighboring $N = 90$ isotones, $^{150}$Nd, $^{152}$Sm, and $^{156}$Dy, which, along with $^{154}$Gd, happen to be the best manifestations of the X(5) CPS [347,464]. Further discussion on the relation between the X(5) CPS and SC is deferred to the relevant Section 5.5.

Strong electric monopole transitions, characterized by the monopole strength $\rho^2(E0)$, connecting excited $0^+$ states to the ground state, are an important signature of SC [491], since they provide a clear sign of shape mixing. The relevant data have been reviewed in [526,527], while novel experimental approaches for their identification, based on conversion electron spectroscopy, have been reviewed by Zganjar [528].

*2.9. Multiple Shape Coexistence*

In some nuclei more than one excited low-lying $0^+$ states have been observed, leading to the suggestion of multiple SC.

The first example has been found in 2000 in $^{186}_{82}$Pb$_{104}$ [529], in which three coexisting bands with spherical, oblate, and prolate shapes have been determined. Similar observations have been made in other Pb isotopes [530].

Three different configurations have been found in 2018 in $^{96}_{38}$Sr$_{58}$ [531], in accordance to earlier (2012) predictions by Petrovici within the complex excited VAMPIR approach [190]. In the same paper [190], triple SC has been predicted also in the neighboring nucleus $^{98}_{40}$Zr$_{58}$, in which the influence of multiple SC in $^{98}_{40}$Zr$_{58}$ on the $\beta^-$ decay from $^{96}_{39}$Y$_{57}$ to it is discussed [196].

Four coexisting bands have been found in 2019 in $^{110,112}_{48}$Cd$_{62,64}$ [177,178], interpreted in terms of beyond-mean-field approach with the Gogny D1S interaction [177]. Energy systematics suggest that they are built on multiparticle–multihole proton excitations across the $Z = 50$ gap [178].

Triple coexistence of spherical, prolate, and oblate shapes has been found in 2022 in $^{64}_{28}$Ni$_{36}$ [532], interpreted in terms of MCSM calculations. Triple coexistence of spherical, prolate, and oblate shapes has also been predicted [415,416] in the framework of partial dynamical symmetries [413,414] of the IBM.

Multiple SC has also been predicted in 2011 in $^{80}_{40}$Zr$_{40}$ [138], using the beyond-mean-field approach with the Gogny D1S interaction, as well as in 2017 in $^{76,78}_{36}$Kr$_{40,42}$ [224], using an IBM Hamiltonian with parameters derived microscopically from a Gogny D1M energy density functional.

A mechanism for multiple SC, being an extension of the dual-shell mechanism described in Section 2.4.6 has been recently suggested [335]. In short, the simultaneous presence of the harmonic oscillator (HO) magic numbers and the spin–orbit (SO) magic numbers, described in Section 2.4.6, can lead to four different combinations of magic numbers of protons and neutrons, namely SO-SO, SO-HO, HO-SO, and HO-HO, leading to four different shapes for the same nucleus [335].

*2.10. Islands of Inversion*

In light nuclei the term *islands of inversion* (IoIs) has been coined [533] in the $N = 20$ region, in order to describe the situation in which in a group of semimagic or even doubly magic nuclei, expected to have a spherical ground state, turn out to have a deformed ground state. This effect is attributed [534] to the occurrence of highly correlated many-particle–

many-hole configurations, called intruder configurations, pushed low in energy by the proton–neutron interaction, so that they go lower in energy than the spherical configuration that otherwise would have been the ground state. This mechanism is identical [236] to the particle–hole excitation mechanism leading to SC. It also clarifies the relation between SC and SPTs from spherical to deformed ground states, with the SPT expected to take place on the shores of the island of inversion.

Well-documented islands of inversion are known to occur [535,536] at $N = 8$ [537], $N = 20$ [533,537–543], $N = 28$ [49,544,545], $N = 40$ [50,546–551], and $N = 50$ [534]. Merging of the IoIs occurring at $N = 20$ and $N = 28$ has been suggested in [49,545], while merging of the IoIs occurring at $N = 40$ and $N = 50$ has been proposed in [534,551].

Islands of inversion will be further discussed in Section 12.

### 3. The $Z \approx 82$ Region

*3.1. The Hg (Z = 80) Isotopes*

The Hg isotopes are the ones in which SC has been first observed.

The plot of experimental energy-level systematics of the Hg isotopes, appearing in similar form in many papers (see Figure 6 of [552] , Figure 10 of [4], Figure 1 of [435], Figure 1 of [553]), and shown here in Figure 7a, is the hallmark of SC. Excited $K^\pi = 0^+$ bands invade the practically vibrational spectra of the ground state bands of Hg isotopes, forming parabolas with minima at $N = 102$, which is close to the midshell $N = 104$. The parabolas raise towards infinity at $N = 96$ on the left and at $N = 110$ on the right, providing strong evidence for SC of a deformed excited $0^+$ band with a spherical ground state band.

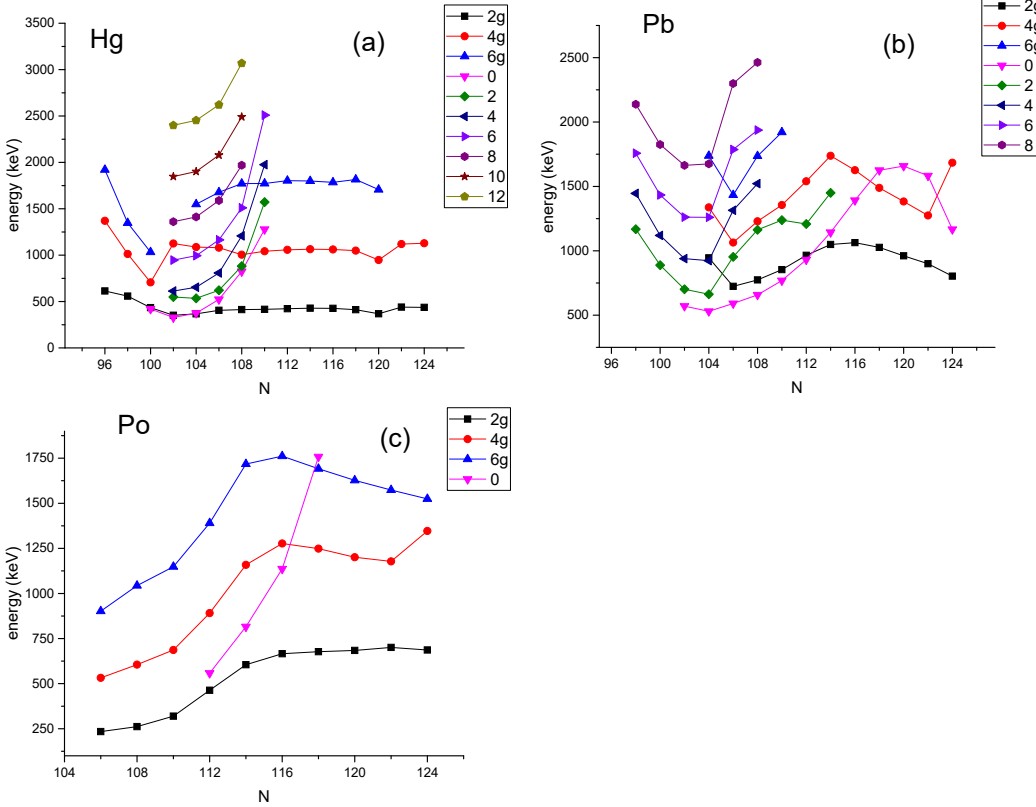

**Figure 7.** Experimental energy levels in keV [406] for the Hg ($Z = 80$) (**a**), Pb ($Z = 82$) (**b**), and Po ($Z = 84$) (**c**) series of isotopes. The levels of the ground state band bear the subscript g, while the levels of the excited $K^\pi = 0^+$ band bear no subscript. See Sections 3.1–3.3 for further discussion.

Indeed $^{184}$Hg$_{104}$ and $^{186}$Hg$_{106}$ were the first Hg isotopes in which SC has been observed in 1973 [554–557], confirmed through both spectra and lifetimes. Further experimental studies confirmed the occurrence of SC in these isotopes [558–561].

SC in the neighbors of the previous isotopes, $^{182}$Hg$_{102}$ and $^{188}$Hg$_{108}$, has been seen in [28,553,554,561–565]. Further to the left, SC has been seen in the isotopes $^{178,180}$Hg$_{98,100}$ in [28,566–571].

The lower border has been reached in [566], in which SC is seen in $^{178}$Hg$_{98}$ but not in $^{176}$Hg$_{96}$. The upper border has been reached in [552,564]. In [564] a systematic study has been carried out in $^{188–200}$Hg$_{108–120}$, making clear that the onset of state mixing, and therefore SC, occurs at $^{190}$Hg$_{110}$, while no state mixing and SC is seen in $^{192–200}$Hg$_{112–120}$.

In conclusion, experimental findings support the occurrence of SC within the region $^{176–190}$Hg$_{96–110}$, while no SC is seen outside it.

From the theoretical point of view, SC in $^{178–188}$Hg$_{96–110}$ has been predicted and/or confirmed by many calculations using different methods, including mean field [148,170,572–575], particle plus rotor [576], and extended IBM [424] calculations. $^{190}$Hg$_{110}$ has been suggested as the upper limit of SC in mean-field calculations by Zhang and Hamilton in 1991 [577] and Nazarewicz in 1993 [578]. Recent calculations using covariant density functional theory (CDFT) with the DDME2 functional with standard pairing included, have found that proton particle–hole excitations induced by the neutrons occur in $^{176–190}$Hg$_{96–110}$ [579,580].

The advent of computational facilities made possible in 2013 extended IBM calculations by Nomura et al. [220,222] with parameters microscopically derived from RMF theory with the Gogny–D1M interaction. Detailed calculations all over the region $^{172–204}$Hg$_{92–124}$ have revealed the onset of a second minimum in the potential energy surface at $^{176}$Hg$_{96}$, coexistence of prolate and oblate minima in $^{178–190}$Hg$_{98–110}$, and the disappearance of the prolate minimum at $^{192}$Hg$_{112}$.

Extended calculations have also been performed in 2014 in the IBM-CM by García-Ramos and Heyde [435]. The isotopes $^{172–200}$Hg$_{92–120}$ have been considered. Clear cases of SC have been found in $^{180–188}$Hg$_{100–108}$. The prolate minimum is found to disappear at $^{190}$Hg$_{110}$, in remarkably close agreement with the RMF results of [220].

Earlier extended RMF calculations have been carried out in 1994 by Patra et al. [160] in $^{170–200}$Hg$_{90–120}$. An oblate ground state has been found in most cases, but a transition from oblate to prolate shape has been seen at $^{178}$Hg$_{98}$ and a transition from prolate back to the oblate shape has been seen at $^{188}$Hg$_{108}$. These results are in agreement with the more recent calculations predicting SC of prolate and oblate shapes in the $^{178–188}$Hg$_{98–108}$ region, but they differ in relation to the nature of the ground state in this region, which appears to be oblate and not prolate, as pointed out by Heyde et al. [152], based on the data [566–568], and in agreement with earlier calculations [572,578]. As pointed out in Ref. [165], the energy difference between the prolate and oblate configurations is very sensitive to the choice of the input parameters, as well as to the amount of pairing involved.

In conclusion, the theoretical predictions closely agree with the experimental findings, indicating SC in the region starting at $N = 96$ or 98 and ending up at $N = 108$ or 110.

### 3.2. The Pb (Z = 82) Isotopes

Following the observation of proton particle–hole excitations across the $Z = 82$ magic number in the Hg isotopes, it was expected to search for similar excitations in the Pb isotopes, lying exactly at the $Z = 82$ energy gap. Some findings are summarized in Figure 7b, in which again intruder states forming parabolas with minima at $N = 104$ are seen.

Low-lying $0^+$ states in the Pb isotopes have been first identified in 1984 in $^{192–200}$Pb$_{110–118}$ [581–583], and the right branch of a parabola formed by them, going down in energy from the upper border of the 82–126 neutron shell towards the midshell ($N = 104$) has been formed [583]. It has been realized that the $0^+$ state becomes the first excited state in $^{192,194}$Pb$_{110,112}$, sinking lower than the $2^+$ member of the ground state band [582]. The microscopic nature of these $0^+$ states, based on proton 2p-2h excitations, has also been clarified [581–583]. It was even realized that "the general idea of obtaining low-lying intruder states near or in single-closed-shell nuclei with a maximal number of valence nucleons of

the other type is shown to be correct" [582]. The isotopes $^{188-198}$Pb$_{106-116}$, forming the right branch of the parabola, have been reviewed recently in [565].

The next important step was the identification in 1998 of coexisting spherical, prolate, and oblate configurations in $^{188}$Pb$_{106}$ [530,584–586], offering the first example of triple SC. Triple SC has also been seen in 2000 at the neutron midshell, i.e., in $^{186}$Pb$_{104}$ [529], by realizing that the three lowest in energy states are spherical, oblate, and prolate. An oblate band has been identified [587] in $^{186}$Pb$_{104}$, in addition to the prolate ground state band.

The left branch of the parabola of intruder $0^+$ states has been completed by searches in $^{184}$Pb$_{102}$ [588], $^{182}$Pb$_{100}$ [589], and $^{180}$Pb$_{98}$ [590], reviewed in [591], in which an updated figure of the parabola is given (Figure 1 in [591], Figure 3 in [590]). It is seen that the prolate intruder $0^+$ band reaches a minimum in energy at midshell ($N = 104$) and keeps rising sharply as the neutron number is reduced.

It should be noticed that $^{188}$Pb$_{106}$ and $^{194}$Pb$_{112}$ represent two rare cases in which the $0_2^+$ band-head of a well-developed $K^\pi = 0^+$ band lies below the $2_1^+$ state of the ground state band [335]. It has been found [335] that such nuclei tend to be surrounded by nuclei exhibiting SC.

Early theoretical considerations over extended series of Pb isotopes included calculations [84] in the $^{186-204}$Pb$_{104-122}$ region, confirming the equivalence of the spherical and deformed shell-model approaches to the interpretation of intruder $0^+$ states through the particle–hole excitation mechanism, as well as RMF calculations in the $^{178-208}$Pb$_{96-126}$ region, predicting oblate ground states in the $^{188-196}$Pb$_{106-114}$ and prolate ones outside it [170]. This prediction has been questioned by Ref. [152], leading to the realization [165] that the ordering between prolate and oblate configurations is very sensitive to the input parameters, the pairing interaction included. In addition, calculations have been performed in the $^{190-196}$Pb$_{108-114}$ region, studying particle–hole excitations in the extended IBM formalism [424], as well as in the particle-plus-rotor model, in which configuration mixing in relation to SC has been considered [576].

Theoretical investigations focused on predicting the triple SC of spherical, prolate and oblate shapes in $^{186}$Pb$_{104}$ have been performed in 2003 in the mean-field framework using the Skyrme SLy6 interaction [142], as well as in 2004 in the IBM-CM framework [433]. Soon thereafter, calculations covering the region $^{182-194}$Pb$_{100-112}$ have been performed in the mean-field framework using the Gogny D1S force [130,140], the Skyrme interaction [148], the density-dependent cluster model [592], the total-Routhian-surface (TRS) method [574], the configuration-constrained potential-energy-surface method [575], the IBM-CM [444] with parameters fitted to the data, and the IBM with parameters determined through the Gogny D1M energy density functional [219], consistently describing SC in these nuclei and reproducing the parabolic behavior of the intruder $0^+$ states (see Figure 7 of [140], Figure 3 of [219], Figure 7 of [444]).

It was suggested in Ref. [130] that the low-lying prolate and oblate $0^+$ states are predominantly due to neutron correlations, while the protons behave rather as spectators, in contrast to the traditional interpretation of proton 2p-2h excitations [581–583]. The recent findings in the framework of covariant DFT calculations [579,580] seem to bridge this gap, suggesting that SC in this region is indeed due to proton particle–hole excitations, but the cause of these particle–hole excitations is based on the increased role played by the neutrons near midshell.

Recent (2022) extended calculations [153] using the deformed relativistic Hartree–Bogoliubov (RHB) theory in the $^{172-302}$Pb$_{90-220}$ region, have reconfirmed SC in $^{184-188}$Pb$_{102-106}$, in addition to finding a peninsula near the neutron drip line, consisting of the $^{278-296,300}$Pb$_{196-214,218}$ isotopes. Furthermore, recent calculations using CDFT with the DDME2 functional with standard pairing included have found that proton particle–hole excitations induced by the neutrons occur in $^{180-190}$Pb$_{98-108}$ [579,580].

In summary, the theoretical predictions closely agree with the experimental findings, indicating SC in the region starting at $N = 98$ and ending up at $N = 112$.

### 3.3. The Po (Z = 84) Isotopes

Following the observation of proton particle–hole excitations across the $Z = 82$ magic number in the Hg isotopes, lying two protons below the $Z = 82$ magic number, it was reasonable to search for similar excitations in the Po isotopes, lying two protons above the $Z = 82$ energy gap. Some findings are summarized in Figure 7c, in which intruder states forming the right branch of a parabola are seen.

Since 1991, experimental examples of SC due to proton 4p-2h excitations and configuration mixing with the proton 2p ground state have been found in $^{192-196}$Po$_{108-112}$ in several works [584,593–597]. In addition, experimental examples of SC have been seen in $^{198,200}$Po$_{114,116}$ [593,597–599]. In Ref. [598], a transition was suggested to occur between $^{202}$Po$_{118}$ and $^{200}$Po$_{116}$, corresponding to a transition between heavier Po isotopes, in which the positive parity states can be described as two-phonon multiplets within a quadrupole vibrational model, and lighter Po isotopes, in which proton 4p-2h excitations appear, mixing with the proton 2p ground state band and leading to SC, see also [565] in relation to this transition. The intruder $0_2^+$ states are seen to form the right branch of a parabola reaching $^{202}$Po$_{118}$ (see Figure 4 of [597], Figure 9 of [598], Figure 7 of [596], and Figure 1 of [599]).

From the theoretical point of view, several calculations revealing SC have been performed in the region $^{188-200}$Po$_{104-116}$, including extended-IBM [424,425] and particle-core model [600] calculations, based on the proton 4p-2h excitations picture, as well as mean-field calculations in the standard deformed Woods-Saxon plus pairing approach [573,575] and calculations in the density-dependent cluster model [592]. Efforts extending beyond $^{200}$Po$_{116}$ include IBM-CM calculations [437], in which the change from the SC picture involving 4p-2h configurations to regular rotational behavior when passing from $^{200}$Po$_{116}$ to $^{202}$Po$_{118}$ has been corroborated, as well as beyond-mean-field calculations with a Skyrme energy density functional [148], which indicate that the ground state is oblate above $N = 106$ and prolate at and below it, in agreement with the findings within the particle-core model [600], in which it has been found that in $^{192}$Po$_{108}$ the intruder band becomes the ground-state configuration. Recent calculations using CDFT with the DDME2 functional with standard pairing included have found that proton particle–hole excitations induced by the neutrons occur in $^{182-192}$Po$_{98-108}$ [579,580].

In conclusion, both experimental and theoretical pieces of evidence converge to the conclusion that SC based on proton 4p–2h excitations is seen from $N = 98$ up to $N = 116$, while the regular rotational character dominates above it.

### 3.4. The Rn (Z = 86) Isotopes

Having seen signs of SC in the Po isotopes, lying two protons above the $Z = 82$ energy gap, it is reasonable to search for similar effects in the Rn isotopes, which lie four protons above the $Z = 82$ gap.

Beyond-mean-field calculations with a Skyrme energy density functional [148] in the region $^{194-204}$Rn$_{108-118}$, indicate oblate ground states in these nuclei (see Figures 3 and 6 of [148] for the relevant potential energy curves). Mixing of the ground state and the $0_2^+$ state has been seen in $^{202}$Rn$_{116}$ using a standard deformed Woods–Saxon plus pairing approach [573]. Therefore, theoretical predictions appear to be similar as in the Po isotopes, suggesting the occurrence of SC up to $N = 116$. However, further experimental efforts are needed to confirm this suggestion, since the $0_2^+$ state in $^{202}$Rn$_{116}$ remains elusive [565].

In summary, some hints for SC exists in the $N = 108\text{-}116$ region.

### 3.5. Heavy Nuclei Beyond Z = 86

Data related to SC become scarce in the actinides beyond $Z = 86$. For example, sets of $0^+$ states have been observed [519] in $^{228,230}_{90}$Th$_{138,140}$ and $^{232}_{92}$U$_{140}$, and their microscopic interpretation in terms of the quasiparticle–phonon model has been given [520]. In addition, the spectrum of $^{238}_{94}$Pu$_{144}$ has been described [282] within the Bohr Hamiltonian with a sextic potential possessing two minima, thus possibly related to a first-order SPT from spherical to prolate deformed shapes and/or SC related to it.

SC in superheavy nuclei with $Z = 100$–$114$ has been considered [161] by self-consistent RMF theory, finding that in some cases near $Z = 114$ and $N = 174$ the superdeformed band may become the ground state. More recent calculations on superheavy nuclei using CDFT with various functionals [601–603] indicate that the nature of the ground state depends on the functional used. For example, in the $Z = 120$, $N = 184$ region, some functionals (NL3*, DD-ME2, and PC-PK1) predict spherical ground states, while other functionals (DD-PC1,DD-ME$\delta$) predict oblate shapes [601].

Recent beyond-mean-field calculations [131] using the Gogny force for the $^{288-298}_{114}\text{Fl}_{174-184}$ superheavy nuclei find that triaxial shapes dominate in this region. A prolate ground state is predicted in $^{288}_{114}\text{Fl}_{174}$, while an oblate ground state is predicted in $^{296}_{114}\text{Fl}_{182}$, with a prolate-to-oblate transition predicted to occur between $^{290}_{114}\text{Fl}_{176}$ and $^{292}_{114}\text{Fl}_{178}$, with a novel type of SC of two different triaxial shapes occurring in $^{290}_{114}\text{Fl}_{176}$.

It seems that superheavy nuclei present a special interest for further studies, since a rich variety of novel phenomena appears to be predicted there.

*3.6. The Pt (Z = 78) Isotopes*

Following the observation of proton particle–hole excitations across the $Z = 82$ magic number in the Hg isotopes, lying two protons below the $Z = 82$ magic number, it was reasonable to examine to which extent similar excitations appear in the Pt isotopes, lying four protons below the $Z = 82$ energy gap.

SC in the Pt isotopes was first seen experimentally in 1986 in $^{184}\text{Pt}_{106}$ [604–606]. It was later found in lighter isotopes, down to $^{174}\text{Pt}_{96}$ [607–611], as well as in heavier isotopes, up to $^{188}\text{Pt}_{110}$ [612,613]. In summary, experimental evidence for SC has been found in the region $^{174-188}\text{Pt}_{96-110}$, in agreement with Figure 3.26 of the review article [3] (see also Figure 3 of [434]), in which the weakly deformed and the strongly deformed states in the even–even Pt isotopes are depicted, showing that the strongly deformed states nearly form a well with walls sharply rising at $N = 96$ on the left and at $N = 110$ on the right.

Several theoretical calculations have been carried out using the IBM-CM [434,443,446], in which the model parameters have been fitted to the data. In addition, IBM calculations have been carried out with parameters determined from energy density functional theory with Gogny-D1S [216] and Gogny-D1M [218] interactions, as well as with the DD-PC1 functional [217]. A comparison between these two approaches has been carried out in [436], reaching the conclusion that SC is seen in the region $^{176-186}\text{Pt}_{98-108}$, in agreement with the data.

An alternative point of view had been given in [614], in which IBM calculations have been carried out in the region $^{172-194}\text{Pt}_{94-116}$, claiming that a good description of these isotopes can be obtained without configuration mixing, i.e., without involving particle–hole excitations. It seems that p-h effects are stronger for nuclei lying at $Z = 82 \pm 2$ (namely Pb, Po, Hg), while they are "fading away" when moving further away from the magic number 82. As a consequence, the SC effects in Pt, which appears to be a "border case", can also be described in a theoretical framework involving free parameters without explicitly including the p-h excitations.

A detailed analysis [101] of $^{194}\text{Pt}_{116}$ in terms of IBM with configuration mixing has shown that the mixing between the ground-state band and the excited band is very small, while the lower band is only slightly more collective than the upper band, showing that the criteria for SC are not satisfied in this nucleus.

Configuration mixing and/or SC of prolate and oblate shapes within the $^{176-188}\text{Pt}_{98-110}$ region have been corroborated by several calculations in the mean-field framework [572,573,575], as well as within the particle plus rotor model [576] and the Bohr collective model [615]. Deformed RMF calculations [170] have confirmed the absence of spherical shapes in $^{176-180}\text{Pt}_{98-102}$, in contrast to the Pb isotopes, in which triple SC of spherical, prolate, and oblate structures has been seen.

Several calculations have focused on the region $^{184-200}\text{Pt}_{106-122}$. Both mean-field calculations [145,616], as well as 5DQCH calculations with parameters determined through

CDFT [244] have found that an oblate-to-prolate transition occurs at $N = 116$, where the prolate and oblate minima have approximately the same depth. In addition, RMF calculations have shown that deformation and SC are favored within the $^{176-190}$Pt$_{98-112}$ region [168], in which oblate shapes are favored [162].

Recent calculations [151] within the deformed RHB theory have indicated that bubble structures with SC can be obtained in the extremely neutron-rich nucleus $^{210}$Pt$_{132}$.

In summary, both experimental findings and theoretical predictions suggest SC in the region bordered roughly by $^{174}$Pt$_{96}$ and $^{188}$Pt$_{110}$.

## 4. The Z = 68–76 Desert

### 4.1. The Os (Z = 76) Isotopes

Experimental results on $^{176-180}$Os$_{100-104}$ [383] have pointed to $^{178}$Os$_{102}$ as a good candidate for the X(5) CPS. SC in $^{178}$Os$_{102}$ of a prolate deformed ground state with $\gamma$-soft excited states has been supported by experimental results in [617], backed up by projected angular momentum deformed HF calculations, as well as by cranked Woods–Saxon model calculations [617]. Shape/phase mixing in the $^{174-180}$Os$_{98-104}$ region has been supported by calculations in a hybrid collective model [615].

The $0^+$ states in $^{190}$Os$_{114}$ have attracted experimental attention [505,506], since the enhanced density of low-lying $0^+$ states has been considered as evidence for shape/phase transitional behavior [505,506]. However, such an enhanced density has been located in $^{154}$Gd$_{90}$ and not in $^{190}$Os$_{114}$ [506].

The $^{186-198}$Os$_{110-122}$ region has attracted much theoretical attention, since SC is expected to occur in relation to the SPT from prolate to oblate shapes seen around $N = 116$ [324], which implies the occurrence of SC around it. Potential energy surfaces revealing this transition have been calculated within a self-consistent HFB approach [616], a Skyrme HF plus BCS framework [145], as well as with an IBM Hamiltonian with parameters determined from the microscopic energy density functional Gogny D1M [218] and Gogny D1S [209] (see Figure 2 of [616], Figure 2 of [145], Figure 1 of [218], and Figure 1 of [209] for the corresponding PES, as well as Figure 5 of [209] for relevant level schemes).

More recently, extended calculations with a 5DQCH with parameters determined from CDFT have been performed in $^{178-200}$Os$_{102-124}$ [244] (see Figure 5 of [244] for the relevant potential energy surfaces and Figure 9 for level schemes). An evolution is seen from well-deformed prolate shapes at $N = 102$ (near to the midshell $N = 104$) to prolate–oblate SC in $^{190,192}$Os$_{114,116}$, and then to oblate shapes at $N = 118$, 120 and spherical shapes at $N = 122$, 124, where the shell closure at $N = 126$ is approached. While SC around the critical point of the prolate-to-oblate SPT has been seen in $^{182,184}$Er$_{114,116}$ and $^{184,186}$Yb$_{114,116}$, this is not the case in $^{190,192}$Os$_{114,116}$ [244], since the transition from prolate to oblate shapes becomes smooth at higher $Z$.

Potential energy surfaces in the region $^{206-216}$Os$_{130-140}$ have been calculated [207] using an IBM Hamiltonian with parameters determined from mean-field calculations with the Skyrme force (see Figure 21 of [207] for the relevant PES). $^{210}$Os$_{134}$ has been suggested [207] as a possible candidate for the E(5) CPS, its level scheme given in Figure 25 of [207].

Recent calculations [151] within the deformed RHB theory have indicated that bubble structures with SC can be obtained in the extremely neutron-rich nuclei $^{196,206,208,256}$Os$_{120,130,132,180}$.

In summary, some experimental and theoretical support exists for SC and shape/phase mixing in the $^{174-180}$Os$_{98-104}$ region.

### 4.2. The W (Z = 74) Isotopes

The $0^+$ states in $^{180-184}$W$_{106-110}$ have attracted considerable experimental attention [490,505,506], since the enhanced density of low-lying $0^+$ states has been considered as evidence for shape/phase transitional behavior [505,506]. However, such an enhanced density has been located in $^{154}$Gd$_{90}$ and not in $^{180,184}$W$_{106,110}$ [506].

The $^{184-196}$W$_{110-122}$ region has attracted much theoretical attention, since SC is expected to occur in relation to the SPT from prolate to oblate shapes seen around $N = 116$ [324],

which implies the occurrence of SC around it. Potential energy surfaces revealing this transition have been calculated within a self-consistent HFB approach [616], a Skyrme HF plus BCS framework [145], as well as with an IBM Hamiltonian with parameters determined from the microscopic energy density functional Gogny D1M [218] and Gogny D1S [209] (see Figure 2 of [616], Figure 2 of [145], Figure 1 of [218], and Figure1 of [209] for the corresponding PES, as well as Figure 5 of [209] for relevant level schemes).

More recently, extended calculations with a 5DQCH with parameters determined from CDFT have been performed in $^{176-198}$W$_{102-124}$ [244] (see Figure 4 of [244] for the relevant potential energy surfaces). An evolution is seen from well-deformed prolate shapes at $N = 102$ (near to the midshell $N = 104$) to prolate–oblate SC in $^{188,190}$W$_{114,116}$, and then to oblate shapes at $N = 118, 120$ and spherical shapes at $N = 122, 124$, where the shell closure at $N = 126$ is approached. While SC around the critical point of the prolate-to-oblate SPT has been seen in $^{182,184}$Er$_{114,116}$ and $^{184,186}$Yb$_{114,116}$, this is not the case in $^{188,190}$W$_{114,116}$ [244], since the transition from prolate to oblate shapes becomes smooth at higher $Z$.

Potential energy surfaces in the region $^{204-214}$W$_{130-140}$ have been calculated [207] using an IBM Hamiltonian with parameters determined from mean-field calculations with the Skyrme force (see Figure 21 of [207] for the relevant PES). $^{208}$W$_{134}$ has been suggested [207] as a possible candidate for the E(5) CPS, its level scheme given in Figure 25 of [207].

Recent calculations [151] within the deformed RHB theory have indicated that bubble structures with SC can be obtained in the extremely neutron rich nuclei $^{194,204,254}$W$_{120,130,180}$.

In summary, SC in the W isotopes is not predicted anywhere, not even in the $N = 114$, 116 region, in which an SPT from prolate to oblate shapes is predicted in the Er and Yb series of isotopes.

*4.3. The Hf (Z = 72) Isotopes*

The $^{166-170}$Hf$_{94-98}$ isotopes have attracted considerable attention, since they lie close to the Nd, Sm, Gd, and Dy $N = 90$ isotones, which are considered as manifestations of the X(5) CPS. Experimental investigations in $^{166}$Hf$_{94}$ [379,380] have shown reasonable agreement of spectra and B(E2)s to the X(5) CPS, which is using an infinite square well potential in the $\beta$ collective variable, while in $^{168}$Hf$_{96}$ [381] a Davidson potential appears to provide better results, and in $^{170}$Hf$_{98}$ [282,382] a sextic potential possessing two minima appears to be the most appropriate one.

The $0^+$ states in $^{174-178}$Hf$_{102-106}$ have attracted much experimental attention [490,492, 505,506,512], since the enhanced density of low-lying $0^+$ states has been considered as evidence for shape/phase transitional behavior [505,506]. However, such an enhanced density has been located in $^{154}$Gd$_{90}$ and not in $^{176}$Hf$_{104}$ [506]. A theoretical description for $^{176}$Hf$_{104}$ has been provided within the pseudo-SU(3) symmetry [508,509].

The $^{182-194}$Hf$_{110-122}$ region has attracted much theoretical attention, since SC is expected to occur in relation to the SPT from prolate to oblate shapes seen around $N = 116$ [324], which implies the occurrence of SC around it. Potential energy surfaces revealing this transition have been calculated within a self-consistent HFB approach [616], a Skyrme HF plus BCS framework [145], as well as with an IBM Hamiltonian with parameters determined from the microscopic energy density functional Gogny D1M [218] (see Figure 2 of [616], Figure 2 of [145], and Figure 1 of [218] for the corresponding PES).

More recently, extended calculations with a 5DQCH with parameters determined from CDFT have been performed in $^{174-196}$Hf$_{102-124}$ [244] (see Figure 3 of [244] for the relevant potential energy surfaces). An evolution is seen from well-deformed prolate shapes at $N = 102$ (near to the midshell $N = 104$) to prolate–oblate SC in $^{186,188}$Hf$_{114,116}$, and then to oblate shapes at $N = 118, 120$ and spherical shapes at $N = 122, 124$, where the shell closure at $N = 126$ is approached. While SC around the critical point of the prolate-to-oblate SPT has been seen in $^{182,184}$Er$_{114,116}$ and $^{184,186}$Yb$_{114,116}$, this is not the case in $^{186,188}$Hf$_{114,116}$ [244], since the transition from prolate to oblate shapes becomes smooth at higher $Z$.

Recent calculations [151] within the deformed RHB theory have indicated that bubble structures with SC can be obtained in the extremely neutron rich nuclei $^{202,234,236,240,250}$Hf$_{130,162,164,168,178}$.

In summary, SC in the Hf isotopes is not predicted anywhere, not even in the $N = 114$, 116 region, in which an SPT from prolate to oblate shapes is predicted in the Er and Yb series of isotopes.

### 4.4. The Yb (Z = 70) Isotopes

The $0^+$ states in $^{168-172}$Yb$_{98-102}$ have attracted much experimental attention [490,506,512], since the enhanced density of low-lying $0^+$ states has been considered as evidence for shape/phase transitional behavior [505,506]. However, such an enhanced density has been located in $^{154}$Gd$_{90}$ and not in $^{170}$Yb$_{100}$ [506]. A theoretical description for the many $0^+$ states in $^{170}$Yb$_{100}$ has been provided within a monopole pairing plus quadrupole–quadrupole model with a spin quadrupole force [511], while for $^{172}$Yb$_{102}$ the pseudo-SU(3) symmetry [508,509] has been used.

The $^{158-162}$Yb$_{88-92}$ isotopes [241,379,618] have attracted considerable attention, since $^{160}$Yb$_{90}$ is in line with the Nd, Sm, Gd, and Dy $N = 90$ isotones, which are considered as manifestations of the X(5) CPS. Calculations have been performed within the cranked HFB approximation [618], and a 5DQCH with parameters obtained from CDFT [241]. Marginal relation to the X(5)CPS has been found [379]. Extensive calculations [143] in $^{152-170}$Yb$_{82-100}$, performed with a 5DQCH with parameters determined from three different mean-field solutions have demonstrated the significant influence of the pairing correlations on the speed of the development of collectivity with increasing neutron number.

The $^{180-192}$Yb$_{110-122}$ region has attracted much theoretical attention, since SC is expected to occur in relation to the SPT from prolate to oblate shapes seen around $N = 116$ [324], which implies the occurrence of SC around it. Potential energy surfaces revealing this transition have been calculated within a self-consistent HFB approach [616], a Skyrme HF plus BCS framework [145], as well as with an IBM Hamiltonian with parameters determined from the microscopic energy density functional Gogny D1M [218] (see Figure 2 of [616], Figure 2 of [145], and Figure 1 of [218] for the corresponding PES).

More recently, extended calculations with a 5DQCH with parameters determined from CDFT have been performed in $^{172-194}$Yb$_{102-124}$ [244] (see Figure 2 of [244] for the relevant potential energy surfaces). An evolution is seen from well-deformed prolate shapes at $N = 102$ (near to the midshell $N = 104$) to prolate–oblate SC in $^{184,186}$Yb$_{114,116}$, and then to oblate shapes at $N = 118$, 120 and spherical shapes at $N = 122$, 124, where the shell closure at $N = 126$ is approached. The level scheme of $^{186}$Yb$_{116}$ can be seen in Figure 11 of [244]. The occurrence of SC around the critical point of the prolate-to-oblate SPT seen in this exotic region bears great similarity to the SC seen in Sections 5.5 and 8.5.

In summary, SC in the Yb isotopes is predicted only in the $N = 114$, 116 region, in which an SPT from prolate to oblate shapes is predicted. Once more, the close relation between SC and SPT is pointed out.

### 4.5. The Er (Z = 68) Isotopes

The $0^+$ states in $^{168}$Er$_{100}$ have attracted much experimental attention [504–506,512], since the enhanced density of low-lying $0^+$ states has been considered as evidence for shape/phase transitional behavior [505,506]. However, such an enhanced density has been located in $^{154}$Gd$_{90}$ and not in $^{168}$Er$_{100}$ [505,506]. Theoretical descriptions for the many $0^+$ states in $^{168}$Er$_{100}$ have been provided within the quasiparticle–phonon model [504,520] and the projected shell model [504], as well as within the pseudo-SU(3) symmetry [508,509].

The development of nuclear deformation, leading to the emergence of simplicity and SC has been theoretically studied in $^{168}$Er$_{100}$ in the framework of the symplectic shell model [619], and in $^{166}$Er$_{98}$ within an algebraic many-nucleon version of the Bohr–Mottelson unified model [620]. Detailed level schemes for $^{166}$Er$_{98}$ [239] have been provided

by a 5DQCH with parameters determined through the relativistic energy density function-als DD-PC1 and PC-F1 (see Figures 5 and 8 of [239] for the corresponding level schemes).

The $^{156-160}$Er$_{88-92}$ isotopes have attracted considerable attention, since $^{158}$Er$_{90}$ is a neighbor of the Nd, Sm, Gd, and Dy $N = 90$ isotones, which are considered as manifestations of the X(5) CPS. Calculations have been performed within a hybrid collective model [615], a quartic potential [503], the cranked HFB approximation [618], and a 5DQCH with parameters obtained from CDFT [241]. Marginal relation to the X(5) CPS has been found.

Extended calculations with a 5DQCH with parameters determined from CDFT have been performed in $^{170-192}$Er$_{102-124}$ [244] (see Figure 1 of [244] for the relevant potential energy surfaces). An evolution is seen from well-deformed prolate shapes at $N = 102$ (near to the midshell $N = 104$) to prolate–oblate SC in $^{182,184}$Er$_{114,116}$, and then to oblate shapes at $N = 118, 120$ and spherical shapes at $N = 122, 124$, where the shell closure at $N = 126$ is approached. The level scheme of $^{184}$Er$_{116}$ can be seen in Figure 11 of [244]. The occurrence of SC around the critical point of the prolate-to-oblate SPT seen in this exotic region bears great similarity to the SC seen in Setions 5.5 and 8.5.

In summary, SC in the Er isotopes is predicted only in the $N = 114, 116$ region, in which an SPT from prolate to oblate shapes is predicted. Once more, the close relation between SC and SPT is pointed out.

## 5. The $Z \approx 60$, $N \approx 90$ Region

### 5.1. The Sm (Z = 62) Isotopes

5.1.1. Sm (Z = 62) Isotopes above $N = 82$

The change of equilibrium (ground state) deformation occurring between $^{150}$Sm$_{88}$ and $^{152}$Sm$_{90}$ has been known since 1966 [621], while the question of SC in $^{152}$Sm$_{90}$ has been raised since 1969 [622], albeit in relation to the $0_3^+$ state.

The idea of two coexisting phases in $^{152}$Sm$_{90}$, related to the transition from spherical U(5) to deformed SU(3) shapes, and corresponding in the IBM framework to a potential energy surface with two minima, was proposed in 1998 [412], followed by extensive experimental work [368,371] supporting this idea. Further evidence in favor of this description has been provided both in the IBM [367] and the geometrical collective model [373] frameworks, while the question regarding the occurrence of a phase transition in a system characterized by discrete integer nucleon numbers has been addressed in [623].

In relation to this SPT, the concept of the X(5) CPS was introduced by Iachello [451] in 2001, using an approximate solution of the Bohr Hamiltonian with an infinite square well potential in the $\beta$ variable (which can be considered as a crude approximation to a double-well potential with two wells of equal depth), and a steep harmonic oscillator potential in the $\gamma$ variable, centered at $\gamma = 0$ and therefore corresponding to prolate deformed shapes. Characteristic parameter-independent predictions of the X(5) CPS is the $R_{4/2}$ ratio for the ground state band, which is 2.91, and the relative excitation energy of the first excited $0^+$ state, $E(0_2^+)/E(2_1^+)$, which should be 5.67, both of them being very close to the experimental values found for $^{152}$Sm$_{90}$. The parameter-independent (up to an overall scale factor) predictions of the X(5) CPS for spectra and B(E2) transition rates involving the ground state and other $K = 0$ bands have been found to be in good agreement with the data for $^{152}$Sm$_{90}$ [356,358,360,366,372]. Good agreement has also been found for the $\gamma$-band, in the case of which a free parameter enters [364].

The nature of the $0_2^+$ state of $^{152}$Sm$_{90}$ has been thoroughly discussed, since it plays a central role in the X(5) CPS. While initially this state was thought as being a clear example of a $\beta$-vibration in a deformed nucleus [365,490], within the X(5) CPS it appears as a spherical state coexisting with the deformed ground state band (gsb). (It is worth noticing at this point that the gsb and the $K = 0$ bands based on $0_2^+$ and $0_3^+$ in $^{152}$Sm$_{90}$ have $R_{4/2}$ ratios of 3.01, 2.69, and 2.53 respectively [406]). The characterization of the bands based on the $0_2^+$ states in the $N = 90$ region as $\beta$-vibrations has been questioned [525], starting with the

$0_2^+$ state in $^{154}$Gd$_{90}$ [522–524]. Furthermore, no experimental evidence for multiphonon structures has been found in $^{152}$Sm$_{90}$ [369,370,518].

As mentioned above, before the introduction of the X(5) CPS in 2001 [451], the term phase coexistence was used [367,368,371,412,623], in order to distinguish this effect from SC, which was already identified as connected to intruder bands arising from particle–hole excitations [490]. The term phase/shape coexistence started also being used [373]. It was also suggested that $^{152}$Sm$_{90}$ indicates a complex example of SC instead of a critical point nucleus [624]. Given the recent work on the close relation between SC and SPTs [439,469], this question appears to have been answered. The study in Ref. [439] is particularly relevant, since it regards the $Z \approx 40$, $N \approx 60$ region, which bears great similarity to the present $Z \approx 60$, $N \approx 90$ region, as we will further see below in Section 8.5.

$^{150}$Sm$_{88}$ has been found [363] to lie very close to the SPT from spherical to prolate deformed shapes. Recent lifetime measurements [625] support the appearance of SC in it. See also Ref. [626] for an early comparison between $^{150}$Sm$_{88}$ and $^{70}$Ge$_{38}$, supporting the occurrence of SC in both.

On the theoretical side, spectra and B(E2)s of $^{152}$Sm$_{90}$ have been successfully described in the geometric collective model [373] and in the dynamic pairing plus quadrupole (DPPQ) model [627].

More recently, extended calculations have been performed in the region $^{144–158}$Sm$_{82–96}$ within several schemes. Early RMF calculations [157] found relatively flat potential surfaces for $^{148–152}$Sm$_{86–90}$, in agreement with the idea of an SPT. Subsequent calculations using the self-consistent Skyrme-HF+BCS approach [144], the self-consistent HFB approximation with the D1S parametrization of the Gogny interaction [135,137], as well as relativistic HB calculations using the DD-ME2 force [158] have found potential energy curves with two minima for $^{152}$Sm$_{90}$ and its neighbors $^{150,154}$Sm$_{88,92}$, approximating the flat potential needed for an SPT (see Figure 9 of [144], Figure 10 of [135], Figure 4 of [137], Figure 2 of [158]). Recent calculations using CDFT with the DDME2 functional with standard pairing included, have found that neutron particle–hole excitations induced by the protons occur in $^{152,154}$Sm$_{90,92}$ [579,580].

More detailed calculations in the region $^{144–158}$Sm$_{82–96}$, or parts of it, in which spectra and B(E2)s are obtained, have been performed with a 5DQCH with parameters calculated from CDFT [231,233,235,241] (for level schemes see Figure 12 of [231] and Figure 5 of [233]). In addition, IBM calculations have also been performed, with parameters determined from the Skyrme interaction [207,226] and from the PC-PK1 energy density functional [228]. In the latter calculation, one can see that the addition of pairing significantly lowers the energy of the excited $K = 0$ bands, thus improving agreement to the data (see Figure 10 of [228]). Finally, the $0^+$ states of $^{156}$Sm$_{94}$ have been successfully described [508,509] in the framework of the pseudo-SU(3) symmetry.

In summary, signatures of SC have been seen in the critical nucleus $^{152}$Sm$_{90}$ and its neighbors $^{150,154}$Sm$_{88,92}$.

5.1.2. Sm ($Z = 62$) Isotopes below $N = 82$

$^{130−144}$Sm$_{68−82}$ have been described [321] in the quasi-SU(3) framework. It is found that for the neutrons, the orbitals of the $sdg$ shell, plus the intruder orbital $1h_{11/2}$, coming down from the $pfh$ shell, do not suffice to explain the large B(E2)s observed in $^{134}$Sm$_{72}$, and that the neutron $2f_{7/2}$ orbital, coming from the 82–126 shell, has to be added to the model space for large enough collectivity to be obtained. The simultaneous inclusion of the $1h_{11/2}$ and $2f_{7/2}$ orbitals, differing by $\Delta j = 2$, is the hallmark of the quasi-SU(3) symmetry. In other words, neutron particle–hole excitations from the $1h_{11/2}$ to the $2f_{7/2}$ orbital are needed in order to account for the increased collectivity, thus supporting microscopically the possible occurrence of SC in $^{134}$Sm$_{72}$.

Within a completely different approach, the isotopes $^{128−142}$Sm$_{66−80}$ have been described [243] by a 5DQCH with parameters determined from the relativistic energy density

functional PC-PK1. In this case, the first two $0^+$ states in $^{136}Sm_{74}$ are found to possess significantly different deformation, again signifying the presence of SC.

In addition, experimental signs of SC have been found in the near spherical nucleus $^{142}Sm_{80}$ [628], as well as in $^{140}Sm_{78}$ [629], where the onset of deformation below $N = 82$ occurs. The role of the intruder neutron orbitals in the onset of deformation has been emphasized in [628].

In summary, evidence for SC starts appearing in the neutron-deficient Sm isotopes with $N < 82$, the isotopes with $N = 72, 74, 78, 80$ obtained as the first candidates.

*5.2. The Gd (Z = 64) Isotopes*

5.2.1. Gd (Z = 64) Isotopes above $N = 82$

The analogy between the $0_2^+$ states of $^{154}Gd_{90}$ and $^{152}Sm_{90}$ was realized early [490], as well as the similarity of the spectrum and B(E2)s of $^{154}Gd_{90}$ to the predictions of the X(5) CPS [356,375,376]. The lack of $\beta$-vibrations in $^{152-156}Gd_{88-92}$ has also been realized [521–525], as in the case of the Sm isotopes. The $0_2^+$ state of $^{152}Gd_{88}$ has been found [374] not to be of intruder nature.

A large number of excited $0^+$ states has been found in $^{158}Gd_{94}$ and $^{156}Gd_{92}$ [492–495], 13 and 6 respectively. The $0_6^+$ state (2276.66 keV) of $^{158}Gd_{94}$ [495], as well as the $0_4^+$ state (1715.2 keV) of $^{156}Gd_{92}$ [492] have been found to correspond to two-phonon $\gamma\gamma$ vibrations. The enhanced density of low-lying $0^+$ states in $^{154}Gd_{90}$ has been suggested [505,506] as a new signature of shape/phase transitional behavior near the critical point (see Figure 4 of [505] and Figure 11 of [506]).

The large number of excited $0^+$ states appearing in $^{156,158}Gd_{92,94}$ has been interpreted in terms of the geometric collective model [496], the IBM [496], the quasiparticle–phonon model [500], the pairing plus quadrupole model [501], and the pseudo-SU(3) symmetry [497–499,508,509].

Calculations extended at least over the $^{152-156}Gd_{88-92}$ region, focusing on the SPT from spherical to prolate deformed shapes, have been performed using the RMF theory with the NL3 interaction [164], the Skyrme-HF plus BCS approach [144], the beyond-mean-field method with the Gogny D1S interaction [135,137], and the relativistic HFB approach using the DD-ME2 parametrization [158]. In all cases a double-well potential energy surface has been obtained in the $^{152-156}Gd_{88-92}$ region, resembling the flat potential needed for a SPT (see Figure 2 of [164], Figure 11 of [144], Figure 11 of [135], Figure 4 of [137], Figure 2 of [158]). In addition, adiabatic TDHFB calculations [181] have been able to reproduce the X(5)-like behavior of $^{154}Gd_{90}$. Recent calculations using CDFT with the DDME2 functional with standard pairing included, have found that neutron particle–hole excitations induced by the protons occur in $^{154,156}Gd_{90,92}$ [579,580].

Recently, more detailed calculations, allowing for predictions of the energy levels and the B(E2)s among them, have been performed in the $^{152-156}Gd_{88-92}$ region and around it, using a 5DQCH with parameters determined through RMF theory [231,233,241]; see Figure 13 of [231] and Figure 6 of [233] for relevant level schemes. The potential energy surface determined for $^{154}Gd_{90}$ in [241] exhibits a soft potential in $\beta$ without two coexisting minima, in agreement with the flat potential expected for an SPT in the framework of the X(5) CPS.

Furthermore, detailed calculations allowing for predictions of the energy levels and the B(E2)s among them have been performed in the $^{152-156}Gd_{88-92}$ region and around it, using the IBM Hamiltonian with parameters determined from CDFT [226,228]. The addition of pairing in these calculations [228] lowers the energy of the excited $0^+$ states, improving agreement to the data (see Figure 11 of [228]).

In conclusion, two competing pictures arise from the theoretical calculations in relation to the critical nucleus $^{154}Gd_{90}$. On one hand, most calculations predict potential energy surfaces with two minima of comparable depth in the $\beta$ direction, which support the presence of SC, and can be considered as an approximation to the flat-bottom potential needed in order to achieve an SPT. On the other hand, some calculations predict flat potential energy

surfaces, which are in perfect agreement with the flat-bottom potential needed in order to achieve an SPT, but do not support SC. The sensitivity of these predictions to the parameter sets used in each case has been pointed out by many authors. Theoretical approaches based on first principles, such as symmetries, can be helpful in resolving this issue.

In summary, signatures of SC have been seen in the $^{152-156}$Gd$_{88-92}$ region, surrounding the critical point of the SPT from spherical to prolate deformed shapes.

### 5.2.2. Gd ($Z = 64$) Isotopes below $N = 82$

Four-particle four-hole excitations have been suggested in $^{146}$Gd$_{82}$ [85], corresponding to proton 2p-2h excitations across the $Z = 64$ gap and neutron 2p-2h excitations across the $N = 82$ gap.

### 5.3. The Dy ($Z = 66$) Isotopes

Experimental attention has been focused on $^{156}$Dy$_{90}$, which has been found [356,375,377,378,503] to be a reasonable candidate for the X(5) CPS, albeit of lower quality [375,378] in comparison to the Nd, Sm, and Gd $N = 90$ isotones, since the $\gamma$ degree of freedom appears to play a more important role here. Twelve $0^+$ states have been observed in $^{162}$Dy$_{96}$ [505,506].

The seven excited $0^+$ states appearing in $^{164}$Dy$_{98}$ have been interpreted in terms of the pseudo-SU(3) symmetry [508,509].

Calculations extended over the $^{148-162}$Dy$_{82-96}$ region, or parts of it, focusing on the SPT from spherical to prolate deformed shapes, have been performed using the RMF theory with the NL3 interaction [164], the Skyrme-HF plus BCS approach [144], the self-consistent HFB method with the Gogny D1S interaction [135], and the relativistic HFB approach using the DD-ME2 parametrization [158]. In all cases a double-well potential energy surface has been obtained in the $^{154-158}$Dy$_{88-92}$ region, resembling the flat potential needed for an SPT(see Figure 3 of [164], Figure 12 of [144], Figure 11 of [135], Figure 2 of [158]). The fact that $^{156}$Dy$_{90}$ represents a poor candidate for the X(5) CPS, in comparison to its $N = 90$ neighbors in the Nd, Sm, and Gd chains of isotopes, has been emphasized [158].

Calculations allowing for predictions of the energy levels and the B(E2)s among them, have been recently performed for $^{156}$Dy$_{90}$ and its neighbors, using a 5DQCH with parameters determined through RMF theory [233,241], see Figure 7 of [233] for a relevant level scheme. The potential energy surface determined for $^{156}$Dy$_{90}$ in [241] exhibits a soft potential in $\beta$ without two coexisting minima, in agreement with the flat potential expected for an SPT in the framework of the X(5) CPS.

Furthermore, detailed calculations allowing for predictions of the energy levels and the B(E2)s among them, have been performed in the $^{150-162}$Dy$_{84-96}$ region, using the IBM Hamiltonian with parameters determined from CDFT [226,228]. The addition of pairing in these calculations [228] lowers the energy of the excited $0^+$ states, improving agreement to the data (see Figure 12 of [228]).

The critical point nucleus $^{156}$Dy$_{90}$ has been adequately described in a hybrid collective model [615], revealing a sizeable shape/phase mixing.

In summary, signatures of SC have been seen in the critical nucleus $^{156}$Dy$_{90}$ and its neighbors $^{154,158}$Dy$_{88,92}$.

### 5.4. The Nd ($Z = 60$) Isotopes
### 5.4.1. Nd ($Z = 60$) Isotopes above $N = 82$

Experimental work has been focused on $^{150}$Nd$_{90}$, which has been found to agree very well [356,358,360–362] with the parameter-independent predictions of the X(5) CPS.

Calculations extended over the $^{142-156}$Nd$_{82-96}$ region, or parts of it, focusing on the SPT from spherical to prolate deformed shapes, have been performed using the RMF theory with the NL3 interaction [164] and the PC-F1 interaction [159], the Skyrme-HF plus BCS approach [144], the self-consistent HFB method with the Gogny D1S interaction [135], the beyond-mean-field method with the Gogny interaction [136], and the relativistic HFB

approach using the DD-ME2 parametrization [158]. In all cases a double-well potential energy surface has been obtained in the $^{148-152}$Nd$_{88-92}$ region, resembling the flat potential needed for an SPT (see Figure 1 of [164], Figure 1 of [159], Figure 8 of [144], Figure 10 of [135], Figure 4 of [136], Figure 2 of [158]). Recent calculations using CDFT with the DDME2 functional with standard pairing included have found that neutron particle–hole excitations induced by the protons occur in $^{150,152}$Nd$_{90,92}$ [579,580].

A step further has been taken in Ref. [231], in which a 5DQCH with parameters determined through RMF theory has been used, allowing for the calculation of spectra and B(E2)s, see Figure 6 of [231] for the relevant level scheme of $^{150}$Nd$_{90}$. In Ref. [231] both the $\beta$ and $\gamma$ degrees of freedom have been included, in contrast to Ref. [159], in which only $\beta$ has been considered. However, the potential energy surfaces obtained from the two calculations are very similar, as one can see by comparing Figure 2 of [231] to Figure 1 of [159].

Spectra and B(E2)s have also been calculated in Ref. [228], using an IBM Hamiltonian with parameters determined from CDFT [228]. The addition of pairing in these calculations lowers the energy of the excited $0^+$ states, improving agreement to the data, as one can see in Figure 9 of [228] for the level scheme of $^{152}$Nd$_{92}$.

Spectra and B(E2)s have also been successfully calculated for $^{150}$Nd$_{90}$, using a hybrid collective model with shape/phase mixing [615], as well as for $^{152}$Nd$_{92}$ with the Bohr Hamiltonian with a sextic potential possessing two minima of equal depth [281,282]. The excited $0^+$ states in $^{152}$Nd$_{92}$ have been considered within the pseudo-SU(3) approach in [508,509].

In summary, signatures of SC have been seen in the $^{148-152}$Nd$_{88-92}$ region, surrounding the critical point of the SPT from spherical to prolate deformed shapes.

### 5.4.2. Nd ($Z$ = 60) Isotopes below $N$ = 82

$^{128-142}$Nd$_{68-82}$ have been described [321] in the quasi-SU(3) framework. It is found that for the neutrons, the orbitals of the $sdg$ shell, plus the intruder orbital $1h_{11/2}$, coming down from the $pfh$ shell, do not suffice to explain the large B(E2)s observed in $^{130,132}$Nd$_{70,72}$, and that the neutron $2f_{7/2}$ orbital, coming from the 82–126 shell, has to be added to the model space for large enough collectivity to be obtained. The simultaneous inclusion of the $1h_{11/2}$ and $2f_{7/2}$ orbitals, differing by $\Delta j = 2$, is the hallmark of the quasi-SU(3) symmetry. In other words, neutron particle–hole excitations from the $1h_{11/2}$ to the $2f_{7/2}$ orbital are needed in order to account for the increased collectivity, thus supporting microscopically the possible occurrence of SC in $^{130,132}$Nd$_{70,72}$.

Within a completely different approach, the isotopes $^{126-140}$Nd$_{66-80}$ have been described [243] by a 5DQCH with parameters determined from the relativistic energy density functional PC-PK1. In this case, the first two $0^+$ states in $^{134}$Nd$_{74}$ are found to possess significantly different deformation, again signifying the presence of SC.

In summary, theoretical evidence for SC starts appearing in the neutron-deficient Nd isotopes with $N < 82$, the isotopes with $N = 70$–74 obtained as the first candidates.

### 5.5. Shape Coexistence and Shape/Phase Transition at $N \approx 90$

In Figure 8 the ground state bands and the $K^\pi = 0^+$ bands invading them are plotted for $N = 88, 90, 92$ in the $Z \approx 60$ region.

In Figure 8a we see that for $N = 88$ the intruder levels form a parabola, presenting a minimum at $Z = 64$, i.e., near the proton midshell ($Z = 66$). In Figure 8b the same happens for $N = 90$, while for $N = 92$ the minimum seems to be shifted towards $Z = 68$, as seen in Figure 8c.

In the case of $N = 88$ the intruder $0_2^+$ states around $Z = 64$ fall below the $4_1^+$ members of the ground state band (gsb), while in the case of $N = 90$ the intruder $0_2^+$ states at $Z = 60$–66 are nearly degenerate with the $6_1^+$ members of the gsb. In the case of $N = 92$ the intruder $0_2^+$ states at $Z = 60$–66 are already lying far above the $6_1^+$ members of the gsb, but they are almost degenerate to them at $Z = 68, 70$.

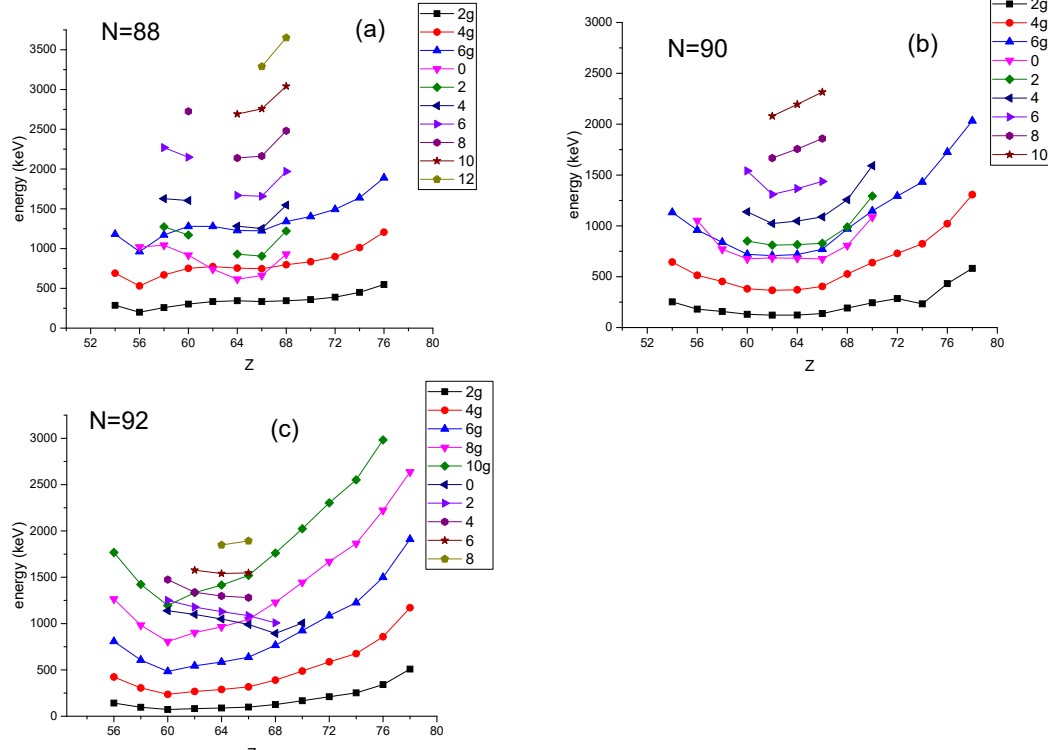

**Figure 8.** Experimental energy levels in keV [406] for the $N = 88$ (**a**), $N = 90$ (**b**), and $N = 92$ (**c**) series of isotones. The levels of the ground state band bear the subscript g, while the levels of the excited $K^\pi = 0^+$ band bear no subscript. See Section 5.5 for further discussion.

Several comments are in place.

The parabolas appearing in Figure 8 vs. $Z$ for the $N \approx 90$ isotones are similar to the parabolas appearing in Figures 7 and 9 vs. $N$, for the $Z \approx 82$ and $Z \approx 50$ isotopes, respectively. In all cases, the minima of the parabolas appear near the relevant midshell. The similarity of the data suggests similarity in the microscopic mechanism leading to the appearance of these effects. In the cases of the $Z \approx 82$ and $Z \approx 50$ isotopes it is known [83–87] that the microscopic origin of the intruder states is lying in proton particle–hole excitations across the $Z = 82$ and $Z = 50$ shell closures, respectively. Therefore it is plausible that in the present case of the $N \approx 90$ isotones the underlying microscopic origin should be related to neutron particle–hole excitations. Indeed, in [579,580] it has been suggested that, in this case, neutron particle–hole excitations occur from normal-parity orbitals to intruder orbitals, which of course bear the opposite parity. It should be noticed that the particle–hole excitations occurring across the $Z = 82$ and $Z = 50$ major shells also correspond to particle–hole excitations between orbitals of opposite parity, since the orbitals involved belong to different major shells.

The appearance of particle–hole excitations across $Z = 82$ and $Z = 50$ for relevant series of isotopes can be attributed to the influence of the changing number of neutrons on the constant set of protons. Thus, SC appearing in these cases has been called *neutron-induced SC* [579,580]. In a similar manner, particle–hole excitations across $N = 90$ for relevant series of isotopes can be attributed to the influence of the changing number of protons on the constant set of neutrons. Thus, SC appearing in these cases has been called *proton-induced SC* [579,580]. This interpretation is consistent with the role played in the shell-model framework by the tensor force [102,105–107], which is known to be responsible for the influence of the protons on the neutron gaps, as well as the influence of the neutrons on the proton gaps, as described in Section 2.1.1.

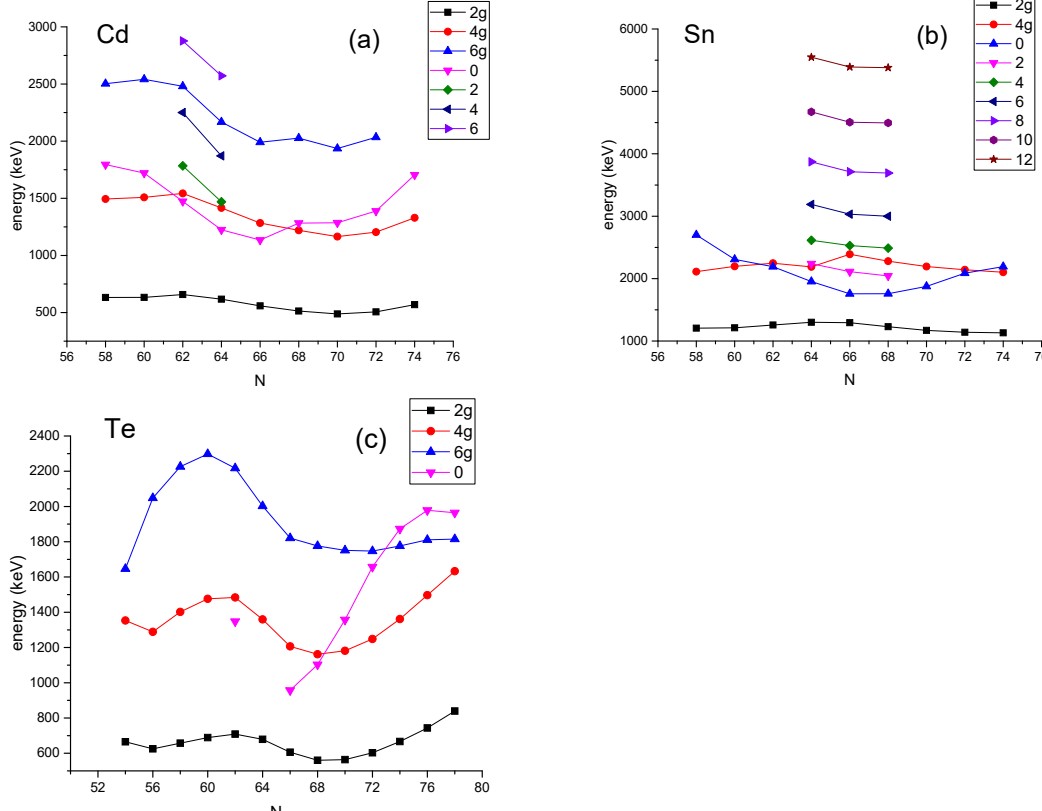

**Figure 9.** Experimental energy levels in keV [406] for the Cd ($Z = 48$) (**a**), Sn ($Z = 50$) (**b**), and Te ($Z = 52$) (**c**) series of isotopes. The levels of the ground state band bear the subscript g, while the levels of the excited $K^\pi = 0^+$ band bear no subscript. See Sections 7.1–7.3 for further discussion.

The fact that for $N = 88$ and $Z = 60$–$64$ the $0_2^+$ intruder levels remain close to the $4_1^+$ members of the ground state bands means that the $K^\pi = 0^+$ intruder band remains close to the ground state band, thus the appearance of SC is possible. SC remains plausible also for $N = 90$, since the $0_2^+$ intruder levels remain close to the $6_1^+$ members of the ground state band, implying again that the $K^\pi = 0^+$ intruder band still remains quite close to the ground state band. In different words, $^{148,150}$Nd$_{88,90}$, $^{150,152}$Sm$_{88,90}$, $^{152,154}$Gd$_{88,90}$, and $^{154,156}$Dy$_{88,90}$ are good candidates for the appearance of SC. This closeness deteriorates for $Z = 60$–$64$ at $N = 92$, which means that $^{152}$Nd$_{92}$, $^{154}$Sm$_{92}$, $^{156}$Gd$_{92}$, and $^{158}$Dy$_{92}$ are not candidates for SC.

The fact that for $N = 90$ and $Z = 60$–$64$ the $0_2^+$ intruder levels remain close to the $6_1^+$ members of the ground state bands implies that the isotones $^{150}$Nd$_{90}$, $^{152}$Sm$_{90}$, $^{154}$Gd$_{90}$, and $^{156}$Dy$_{90}$ are good candidates for the X(5) CPS [451], of which the degeneracy of the $0_2^+$ and $6_1^+$ states is a hallmark. Indeed these isotones have been suggested as being the best manifestations of the X(5) CPS, as discussed in Section 2.6, with detailed relevant references given in the caption of Figure 2. On the other hand, the proximity of the $0_2^+$ states to the $6_1^+$ states seen in $N = 92$ for the $Z = 68, 70$ cases, suggests that $^{160}$Er$_{92}$ and $^{162}$Yb$_{92}$ might be reasonable candidates for X(5). Indeed, such a suggestion exists [379] for the latter.

It should be noticed that the strong similarity between the $N \approx 90$, $Z \approx 64$ and the $N \approx 60$, $Z \approx 40$ regions has been pointed out by Casten et al. already in 1981 [630], suggesting a common microscopic interpretation of the onset of deformation in these two regions.

## 6. The $Z$ = 54–58 Desert

### 6.1. The Ce (Z = 58) Isotopes

On the experimental side, $^{130}$Ce$_{72}$ [356,359] and $^{128}$Ce$_{70}$ [357] have been suggested as examples of the X(5) CPS.

Moving to lighter Ce isotopes, one sees in Figure 9 of [502] that the $0_2^+$ states in $^{128-134}$Ce$_{70-76}$ exhibit a parabolic behavior with a minimum at $^{130}$Ce$_{72}$, while the $0_2^+$ state in $^{128}$Ce$_{70}$ remains close to this minimum, suggesting a region of SC starting at $^{130}$Ce$_{72}$ and extending to the lighter Ce isotopes.

It should be recalled that $^{128-134}$Ce$_{70-76}$ are considered as good examples of the O(6) dynamical symmetry of IBM, as seen in Figure 2.4 of the book [214].

Extended RMF calculations have been performed [171] in the $^{120-142}$Ce$_{62-84}$ region, using different interactions. Potential energy surfaces with two minima of similar depth and different deformation have been seen in $^{126-130}$Ce$_{68-72}$ with the PK1 and NL3 interactions [171], as well as in $^{124-128}$Ce$_{66-70}$ with the TM1 interaction [171].

Recently, $^{126-140}$Ce$_{68-82}$ have been described [321] in the quasi-SU(3) framework. It is found that for the neutrons, the orbitals of the $sdg$ shell, plus the intruder orbital $1h_{11/2}$, coming down from the $pfh$ shell, do not suffice to explain the large B(E2)s observed in $^{124,126}$Ce$_{66,68}$, and that the neutron $2f_{7/2}$ orbital, coming from the 82–126 shell, has to be added to the model space for large enough collectivity to be obtained. The simultaneous inclusion of the $1h_{11/2}$ and $2f_{7/2}$ orbitals, differing by $\Delta j = 2$, is the hallmark of the quasi-SU(3) symmetry. In other words, neutron particle–hole excitations from the $1h_{11/2}$ to the $2f_{7/2}$ orbital are needed in order to account for the increased collectivity, thus supporting, microscopically, the possible occurrence of SC in $^{124,126}$Ce$_{66,68}$.

$^{148}$Ce$_{90}$ has been considered within a wider study of $N = 90$ isotones within a hybrid collective model [615], turning out to be a marginal case neighboring $^{150}$Nd$_{90}$, which is a clear example of the X(5) CPS.

In summary, no clear evidence for SC exists so far in the Ce isotopes, except for signs of neutron particle–hole excitations supporting the occurrence of SC in $^{124,126}$Ce$_{66,68}$, as well as some theoretical indications for a double-minimum potential in the $^{124-130}$Ce$_{66-72}$ region, and/or the occurrence of the X(5) CPS in $^{128,130}$Ce$_{70,72}$.

### 6.2. The Ba (Z = 56) Isotopes

On the experimental side, $^{134}$Ba$_{78}$ [366,391,394,403–405] is considered as the textbook example of the E(5) CPS, being the only unquestionable example of this symmetry identified so far. Further measurements [402] in $^{132,134}$Ba$_{76,78}$ corroborate the transitional character from O(6) to U(5) symmetry in these nuclei, while $^{130}$Ba$_{74}$ [507,631] is found to be close to the O(6) dynamical symmetry [507], in agreement to Figure 2.4 of the book [214], in which $^{126-132}$Ba$_{70-76}$ appear as good examples of the O(6) dynamical symmetry of IBM.

Moving to lighter Ba isotopes, one sees in Figure 8 of [502] that the $0_2^+$ states in $^{122-134}$Ba$_{66-78}$ exhibit a parabolic behavior with a minimum at $^{128}$Ba$_{72}$, while the $0_2^+$ states in $^{124,126}$Ba$_{68,70}$ remain close to this minimum, suggesting a region of SC starting at $^{128}$Ba$_{72}$ and extending to the lighter isotopes $^{124,126}$Ba$_{68,70}$.

Independently, $^{126}$Ba$_{70}$ [356] and $^{122}$Ba$_{66}$ [354,355] have been suggested as examples of the X(5) CPS, which is known to be related to the first-order SPT between spherical and deformed shapes and related to SC in the $N = 90$ region.

In summary, when moving down from the $N = 82$ shell closure to the $N = 66$ midshell one first encounters the E(5) CPS at $N = 78$ and the transition from U(5) shapes above $N = 78$ to O(6) shapes below $N = 78$. Moving further down towards the midshell, one finds the border of a region of SC at $N = 72$ and nuclei exhibiting SC and/or the X(5) CPS below it.

On the theoretical side, potential energy surfaces (PES) for $^{120-138}$Ba$_{64-82}$ have been calculated in the Skyrme-HF plus BCS approach [144], as well as in $^{130-134}$Ba$_{74-78}$ in the self-consistent HFB approximation with the Gogny D1S interaction [135]. Relatively flat PES have been found for $^{132,134}$Ba$_{76,78}$, as seen in Figure 6 of [144] and in Figure 6 of [135],

in agreement with the expectations for an SPT. On the other hand, two minima of different deformations appear in $^{120-126}$Ba$_{64-70}$, which could give rise to SC. Flat PES in $^{132,134}$Ba$_{76,78}$ have also been obtained [238] by a 5DQCH with parameters determined from self-consistent RMF calculations, as seen in Figure 1 of [238], with the corresponding level scheme for $^{134}$Ba$_{78}$ given in Figure 9 of [238]. Flat PES in $^{132,134}$Ba$_{76,78}$ have also been found in IBM calculations with parameters determined by mean-field calculations with the Skyrme interaction [207], as well as in RHB calculations with the DD-ME2 interaction [158], as seen in Figure 10 of [207] and Figure 1 of [158].

IBM calculations in $^{124-132}$Ba$_{68-76}$ with parameters determined from the Gogny D1M energy density functional [227] tend to overestimate the energy of the $0_2^+$ state, as seen in Figure 1 of [227]. The same problem is appearing in $^{124-132}$Xe$_{70-78}$ [227], but it has been resolved in IBM calculations with parameters determined from the PC-PK1 energy density functional [228] in $^{122}$Xe$_{68}$ by taking into account configuration mixing, i.e., by inclusion of the pairing vibrations, as seen in Figure 4 of [228]. Presumably the same remedy would work in the case of the Ba isotopes. The excited $0^+$ states in $^{130-134}$Ba$_{74-78}$ have been considered within a Hamiltonian with pairing, quadrupole–quadrupole, and spin–quadrupole interactions in [510].

Extensive shell-model calculations [321] in the $N = 56$–82 region have shown that the quasi-SU(3) symmetry, implying the inclusion of the $2f_{7/2}$ neutron orbital in addition to the $1h_{11/2}$ one, is needed in order to account for large B(E2) values in some Ce, Nd, and Sm isotopes, as described in the relevant subsections, but such need does not arise for Ba isotopes. However, more recent shell-model calculations [322] in the $^{126-136}$Ba$_{70-80}$ region indicate that the inclusion of both the $2f_{7/2}$ and the $1h_{11/2}$ neutron orbitals is necessary for explaining the oblate to prolate SPT with $N = 76$ as the critical point. It should be remembered that the O(6) dynamical symmetry of IBM has been suggested as lying at the critical point of the oblate-to-prolate transition [453–455]. Therefore the inclusion of quasi-SU(3) in shell-model calculations appears to be connected to the creation of collectivity leading to the O(6) dynamical symmetry of IBM.

This finding points towards an underlying relation between the E(5) CPS, suggested to be present in $^{132,134}$Ba$_{76,78}$, as mentioned in the beginning of this subsection, the O(6) dynamical symmetry of the IBM, and the critical point of the oblate-to-prolate transition. The existence of the subalgebra O(5) in both the E(5) and O(6) symmetries, which guarantees seniority as a good quantum number, appears to play a leading role in this connection, further discussed in Section 2.7. It should be remembered that O(5) not only is a subalebgra of the O(6) and U(5) dynamical symmetries of the IBM, but in addition underlies the whole O(6) to U(5) transition region [482]. Therefore it also underlies the critical point of the relevant second-order SPT [411] from $\gamma$-unstable to spherical shapes, which has been called E(5) [450] in the framework of the Bohr Hamiltonian.

In summary, no clear evidence for SC exists so far in the Ba isotopes, except for some theoretical indications for a double-minimum potential in the $^{120-126}$Ba$_{64-70}$ region and/or the occurrence of the X(5) CPS in this area. On the other hand, strong evidence is accumulated for an SPT from U(5) shapes above $N = 78$ to O(6) shapes below $N = 78$ at $^{132}$Ba$_{76}$ and $^{134}$Ba$_{78}$ and for the occurrence of the E(5) CPS in this area.

*6.3. The Xe (Z = 54) Isotopes*

On the experimental side, $^{128}$Xe$_{74}$ [391,395,399], $^{130}$Xe$_{76}$ [400,401], and $^{132}$Xe$_{78}$ [400], have been considered as candidates for the E(5) CPS.

On the theoretical side, potential energy surfaces (PES) for $^{118-136}$Xe$_{64-82}$ have been calculated both in the Skyrme-HF plus BCS approach [144] and in the self-consistent HFB approximation with the Gogny D1S interaction [135]. Relatively flat PES have been found for $^{128-132}$Xe$_{74-78}$, as seen in Figure 5 of [144] and in Figure 2 of [135], in agreement with the expectations for a SPT. On the other hand, two minima of different deformations appear in $^{118-124}$Xe$_{64-70}$, which could give rise to SC. Flat PES in $^{128-132}$Xe$_{74-78}$ have also been obtained [238] by a 5DQCH with parameters determined from self-consistent RMF

calculations, as seen in Figure 2 of [238], with the corresponding level schemes given in Figures 11–13 of [238]. Flat PES in $^{128–132}$Xe$_{74–78}$ have also been found in IBM calculations with parameters determined by mean-field calculations with the Skyrme interaction [207], as well as in RHB calculations with the DD-ME2 interaction [158], as seen in Figure 11 of [207] and Figure 1 of [158].

IBM calculations in $^{124–132}$Xe$_{70–78}$ with parameters determined from the Gogny D1M energy density functional [227] tend to overestimate the energy of the $0_2^+$ state, as seen in Figure 1 of [227]. This problem has been resolved in IBM calculations with parameters determined from the PC-PK1 energy density functional [228] in $^{122}$Xe$_{68}$ by taking into account configuration mixing, i.e., by inclusion of the pairing vibrations, as seen in Figure 4 of [228].

Extensive shell-model calculations [321] in the $N = 56$–82 region have shown that the quasi-SU(3) symmetry, implying the inclusion of the $2f_{7/2}$ neutron orbital in addition to the $1h_{11/2}$ one, is needed in order to account for large B(E2) values in some Ce, Nd, and Sm isotopes, as described in the relevant subsections, but such a need does not arise for Xe isotopes. However, more recent shell-model calculations [322] in the $^{124–134}$Xe$_{70–80}$ region indicate that the inclusion of both the $2f_{7/2}$ and the $1h_{11/2}$ neutron orbitals is necessary for explaining the oblate to prolate SPT with $N = 76$ as the critical point. It should be remembered that the O(6) dynamical symmetry of IBM has been suggested as lying at the critical point of the oblate-to-prolate transition [453–455]. Therefore, the inclusion of quasi-SU(3) in shell-model calculations appears to be connected to the creation of collectivity leading to the O(6) dynamical symmetry of IBM.

This finding points towards an underlying relation between the E(5) CPS, suggested to be present in $^{128–132}$Xe$_{74–78}$, as mentioned in the beginning of this subsection, the O(6) dynamical symmetry of the IBM, and the critical point of the oblate-to-prolate transition. The existence of the subalgebra O(5) in both the E(5) and O(6) symmetries, which guarantees seniority as a good quantum number, appears to play a leading role in this connection, further discussed in Section 2.7. It should be remembered that O(5) is not only a subalebgra of the O(6) and U(5) dynamical symmetries of the IBM, but in addition underlies the whole O(6) to U(5) transition region [482]. Therefore it also underlies the critical point of the relevant second-order SPT [411] from $\gamma$-unstable to spherical shapes, which has been called E(5) [450] in the framework of the Bohr Hamiltonian.

In summary, no clear evidence for SC exists so far in the Xe isotopes, except for some theoretical indications for a double-minimum potential in the $^{118–124}$Xe$_{64–70}$ region. On the other hand, strong evidence is accumulated for a SPT from oblate to prolate shapes between $^{130}$Xe$_{76}$ and $^{132}$Xe$_{78}$ and/or the occurrence of the E(5) CPS in this area.

## 7. The $Z \approx 50$ Region

### 7.1. The Sn ($Z = 50$) Isotopes

The observation of SC due to proton particle–hole excitations across the magic number $Z = 82$ led to the question if similar excitations also occur across the magic number $Z = 50$. Some findings are summarized in Figure 9b, in which again intruder states forming parabolas with minima at the midshell, here lying at $N = 66$, are seen.

Since 1979, when collective intruder bands based on low-lying $0_2^+$ states corresponding to proton 2p-2h excitations were first observed in $^{112–118}$Sn$_{62–68}$ [632], SC in the $^{110–120}$Sn$_{60–70}$ region has been observed in several experiments [633–641], recently reviewed in [297,516] (see in particular Figure 1.32 of [297], reproduced as Figure 23 in [516]). The neighborhood below this region, namely $^{106,108}$Sn$_{56,58}$, has been investigated in [642–645], in which rotational bands have been found that are much higher in energy and starting at much higher angular momenta, thus providing no evidence of SC with the ground state band.

The low-lying $0_2^+$ collective states in the $^{112–118}$Sn$_{62–68}$ isotopes have been interpreted in terms of proton 2p-2h excitations, mixing with the ground state band, both in a microscopic framework [86,87,646], as well as in extended versions of the IBM [430,647]. Calculations

extending over much wider regions have been recently performed using the MCSM [72] and the shell model taking advantage of the quasi-SU(3) symmetry [321].

In the MCSM calculation [72], covering the $^{100-138}$Sn$_{50-88}$ region, for both protons and neutrons the whole $sdg$ shell, consisting of the $1g_{9/2}, 1g_{7/2}, 2d_{5/2}, 2d_{3/2}, 3s_{1/2}$ orbitals has been taken into account, along with the lower part of the $pfh$ shell, consisting of the $1h_{11/2}, 2f_{7/2}, 3p_{3/2}$ orbitals. The major finding of this work was the explanation of the sudden increase of the $B(E2; 0_1^+ \to 2_1^+)$ value, seen at $^{110}$Sn$_{60}$, in terms of proton excitations from the proton $1g_{9/2}$ orbital, which is in accordance with the earlier explanation of SC in terms of 2p–2h proton excitations in this region, having in mind that the $1g_{9/2}$ orbital is the one lying immediately below the $Z = 50$ magic gap, and further corroborating the experimental fact [642–645] that no SC appears at $N = 58$ and below it.

The shell-model calculations of Ref. [321] corroborate the above findings. Taking into account the full $sdg$ shell plus the $1h_{11/2}$ orbital, one sees that the $B(E2; 0_1^+ \to 2_1^+)$ values are underestimated near the neutron midshell, while this discrepancy goes away as soon as the $2f_{7/2}$ orbital is included in the calculation. This finding also clarifies the essence of the quasi-SU(3) symmetry, which is the inclusion of orbitals differing by $\Delta j = 2$, the $1h_{11/2}$ and $2f_{7/2}$ orbitals in the present case. It is remarkable that in the MCSM calculation of Ref. [72], the full set of orbitals differing by $\Delta j = 2$ from the $pfh$ shell, namely the $1h_{11/2}, 2f_{7/2}, 3p_{3/2}$ orbitals, have been taken into account.

The MCSM calculation [72] also predicts a second-order SPT around $^{116}$Sn$_{66}$ from a moderately deformed phase to the pairing (seniority) phase. As we shall see below, this prediction is in accordance with the earlier suggestion [396] of $^{114}$Cd$_{66}$ (also having $N = 66$) as a candidate for the E(5) CPS.

In summary, the experimental findings and theoretical predictions agree that SC is seen in the region bordered by $^{110}$Sn$_{60}$ and $^{120}$Sn$_{70}$.

*7.2. The Cd (Z = 48) Isotopes*

The Cd isotopes, lying two protons below the magic number $Z = 50$, are the analog of the Hg isotopes, which lie two protons below the magic number $Z = 82$. Therefore, strong evidence for SC is expected to be seen in them. Some findings are summarized in Figure 9a, in which again intruder states forming parabolas with minima at the midshell ($N = 66$), are seen.

Starting in 1977 [648], the first examples of SC in the Cd isotopes, interpreted in terms of proton 2p-4h excitations, were seen experimentally in $^{110-114}$Cd$_{62-66}$ [648–651]. It was also in $^{110,112}$Cd$_{62,64}$ that multiple SC, based on multiparticle–multihole proton excitations, were seen [177,178,652].

Systematics of energy levels revealed a parabola formed by the $0_2^+$ states, related to 2p-4h excitations, having its minimum at the neutron midshell $N = 66$ and extending from $^{106}$Cd$_{58}$ to $^{120}$Cd$_{72}$ (see Figure 1 of [513], Figure 1 of [653], Figure 1 of [654]). B(E2)s connecting the intruder band to the ground-state band corroborate this picture (see Figure 15 of [516] and Figure 5 of [517]).

The lower border of the SC region has been considered as a place of possible manifestation of the E(5) CPS. The $^{106,108}$Cd$_{58,60}$ isotopes have been suggested as possible E(5) candidates [391,395]. Recent investigations [655] suggest that at $^{106}$Cd$_{58}$ the onset of collective rotation occurs (see also Figure 4 of [655], in which the energy systematics suggest that intruder $0_2^+$ and $0_3^+$ bands start from $^{106}$Cd$_{58}$ towards heavier Cd isotopes).

The upper border of the SC region appears to be reached at $^{118}$Cd$_{70}$, which is considered as a textbook example of a near-harmonic vibrational nucleus [656], since, in addition to the two-phonon triplet, the three-phonon quintuplet is also seen. The arguments for and against vibrational motion in the region $^{110-116}$Cd$_{62-68}$, i.e., at the center of the region exhibiting SC, have been considered in detail in [514,515].

Already in 1990 it was seen [653,654] that two sets of low-lying $0^+$ states cross in energy between $^{114}$Cd$_{66}$ and $^{116}$Cd$_{68}$ in what can be interpreted as an early sign of multiple SC, confirmed recently in $^{110,112}$Cd$_{62,64}$ [177,178,652].

$^{114}$Cd$_{66}$ has been suggested as a candidate for the E(5) CPS in [396], in accordance with the recent suggestion [72] in the framework of MCSM calculations of a second order SPT around $^{116}$Sn$_{66}$ (also having $N = 66$) from a moderately deformed phase to the pairing (seniority) phase.

From the theoretical point of view, SC in $^{112,114}$Cd$_{64,66}$ was attributed as early as in 1985 within the shell-model framework to 2p-4h proton excitations and configuration mixing [83,646]. It was within this framework that the close connection between SPTs and SC has been first considered in 2004 [469] in the $N = 66$ isotones, in accordance with the experimental observations mentioned above.

Within the IBM framework, the influence on SC of the O(5) as a common subalgebra in the U(5) and O(6) dynamical symmetries of the model has been considered [484,485], and detailed predictions for $^{108-114}$Cd$_{60-66}$ have been given within the extended IBM framework, involving p-h excitations and configuration mixing.

More recently, calculations discussing SC in the $^{108-116}$Cd$_{60-68}$ region have been performed using parameters determined microscopically through CDFT both in the IBM framework [225] and with a quadrupole collective Hamiltonian [657]. Furthermore, an extended calculation covering the region $^{96-136}$Cd$_{48-88}$ [154] has been performed using CDFT, corroborating the oblate–prolate SC in $^{110-114}$Cd$_{62-66}$.

In summary, the experimental findings and theoretical predictions agree that SC is seen in the region starting at $N = 58$ or 60 and ending at $N = 70$, with special attention paid to the relation between SC and an SPT from moderately deformed to nearly spherical shapes seen at $N = 66$.

*7.3. The Te (Z = 52) Isotopes*

The Te isotopes, lying two protons above the magic number $Z = 50$, are the analog of the Po isotopes, which lie two protons above the magic number $Z = 82$. Therefore, strong evidence for SC is expected to be seen in them. Some findings are summarized in Figure 9c, in which again intruder states forming a parabola with a minimum at the midshell ($N = 66$), are seen.

Extended experimental investigations of the Te isotopes since 1985 by Rikovska et al. [658–661] concluded that SC appears in the region $^{116-124}$Te$_{64-72}$, corresponding to 4p-2h proton excitations, analogous to the 2p-4h proton excitations seen in the Cd isotopes. The upper limit of this region, $^{124}$Te$_{72}$, has been suggested as a manifestation of the E(5) CPS [391,397,398], while behavior similar to E(5), especially in relation to the decays from the low-lying $0^+$ levels, has been seen in the neighboring isotopes $^{122}$Te$_{70}$ and $^{126}$Te$_{74}$ [398].

In recent reviews [516,517], the $0_2^+$ states have been seen to form a parabola having its minimum at the midshell isotope $^{118}$Te$_{66}$ (see Figure 28 of [516] and Figure 17 of [517]). SC has been confirmed in $^{118-124}$Te$_{66-72}$ [516], while a 4p-2h proton excitations character for the $0_2^+$ states has been suggested in [517].

Theoretical investigations within IBM-2 [658,660,662] and the extended IBM [423,430], in which configuration mixing of the 4p-2h configuration with the ground-state 2p configuration has been used, have corroborated SC in $^{116-124}$Te$_{64-72}$, while the relation between SC and SPTs has been considered by [469] in $^{118}$Te$_{66}$, along with other $N = 66$ isotones. The upper limit of the SC region has been discussed [663] within a transitional IBM Hamiltonian taking advantage of the affine $\widehat{SU(1,1)}$ Lie algebra in the transitional region between U(5) and O(6), concluding [663] that SC is seen in $^{118-122}$Te$_{66-70}$, while $^{124,126}$Te$_{72,74}$ are exhibiting E(5) features, in accordance with [391,397,398] mentioned above. Recently the spectral features of $^{118-124}$Te$_{66-72}$ have been successfully reproduced [664] both in the IBM-1 and the dynamic pairing plus quadrupole model, while spectra and B(E2)s of $^{114-124}$Te$_{62-72}$ have been calculated [286] using the Bohr collective Hamiltonian with an infinite square well potential possessing a step.

Early shell-model calculations [646] with configuration mixing corroborated SC in $^{116-120}$Te$_{64-68}$, while recently [321,322] extended shell-model calculations taking advantage of the quasi-SU(3) symmetry, as described above (see the second half of Section 7.1)

for the Sn isotopes, have been performed in $^{108-134}$Te$_{56-82}$. In addition, extended RHB calculations [163] in the $^{104-144}$Te$_{52-92}$ region have corroborated SC in $^{116-120}$Te$_{64-68}$. Recent calculations using CDFT with the DDME2 functional with standard pairing included, have found that proton particle–hole excitations induced by the neutrons occur in $^{116-120}$Te$_{64-68}$ [579,580].

In summary, the experimental findings and theoretical predictions agree that SC is seen in the region bordered by $^{116}$Te$_{64}$ and $^{124}$Te$_{72}$, with possible occurrence of the E(5) CPS at the upper limit.

### 7.4. The Pd (Z = 46) Isotopes

The Pd isotopes, lying four protons below the magic number $Z = 50$, are the analog of the Pt isotopes, which lie four protons below the magic number $Z = 82$. Therefore some evidence for SC is expected to be seen in them.

Low-lying $0^+$ states with intruder bands built on them have been seen in $^{106-112}$Pd$_{60-66}$ [665–667]. Systematics of $0^+$ states in the $^{102-116}$Pd$_{56-70}$ region [517] show a parabola with a minimum at $N = 64$, next to the $N = 66$ midshell. The observation of a strong $\rho^2(E0)$ in $^{106}$Pd$_{60}$ (see Figure 21 in [668]) supports the SC case around this minimum, as also suggested by B(E2) transition rates (see Figure 22 in [517]).

$^{102}$Pd$_{56}$ [391,392,669] and $^{108}$Pd$_{62}$ [393–395] have been suggested as candidates for the E(5) CPS, while in $^{100}$Pd$_{54}$ spectra appear to be closer to the U(5) limit of IBM, but B(E2)s appear to be closer to the O(6) limit, indicating the absence of any IBM dynamical symmetry in this region.

Early theoretical considerations interpreted SC in the $^{108-112}$Pd$_{62-66}$ region in terms of proton p-h excitations in the shell model [86], as well as in the extended IBM framework [430], while the appearance of the E(5) CPS in $^{102}$Pd$_{56}$ has been supported by adiabatic TDHFB calculations [181].

More recently, extended calculations of potential energy surfaces covering the whole $^{96-128}$Pd$_{50-82}$ region, or a large part of it, have been performed using mean-field methods with the Skyrme [144] and Gogny [135] interactions, as well as RMF methods using the DD-ME2 energy density functional [158]. In addition, detailed spectroscopic predictions have been obtained in the IBM framework [670]. These studies were focused on locating candidates for the E(5) CPS, with several suggestions ($^{106}$Pd$_{62}$ [670], $^{108}$Pd$_{62}$ [135,144,158], $^{110}$Pd$_{64}$ [144,158], $^{112-118}$Pd$_{66-72}$ [135]) made, while in [158] it has been suggested that $^{102}$Pd$_{56}$ is not a suitable candidate for E(5).

On the other hand, recent RMF calculations using the DD-PC1 and DD-PCX parametrizations [167] have suggested $^{108}$Pd$_{62}$ as a clear example of prolate–oblate coexistence, locating in addition a prolate-to-oblate SPT at $^{110}$Pd$_{64}$ and an oblate-to-prolate SPT at $^{120}$Pd$_{74}$. In addition, extended IBM calculations [207] with parameters determined microscopically from the Skyrme interaction have found weakly prolate structures for $N = 60–72$, in rough agreement with [167].

In summary, evidence in favor of SC has been presented for $^{106}$Pd$_{60}$ [666–668], $^{108}$Pd$_{62}$ [167,430,667], $^{110}$Pd$_{64}$ [430,665], $^{112}$Pd$_{66}$ [86,665], i.e., in the region $N = 60–66$, while $^{108}$Pd$_{62}$ appears to be the most common suggestion for the E(5) CPS, thus offering a further argument in favor of the connection between SC and SPTs.

## 8. The $Z \approx 40$, $N \approx 60$ Region

As pointed out in [630], the $Z \approx 40$, $N \approx 60$ region is expected to be very similar to the $Z \approx 60$, $N \approx 90$ region. This point will be further discussed in Section 8.5.

### 8.1. The Zr (Z = 40) Isotopes

8.1.1. Zr (Z = 40) Isotopes above $N = 50$

Strong experimental evidence for SC has been accumulated in $^{100}$Zr$_{60}$ [671–676] and $^{98}$Zr$_{58}$ [388,671,677,678], gradually extended to $^{96}$Zr$_{56}$ [88,679,680] and $^{94}$Zr$_{54}$ [681]. The sharp SPT taking place between $^{98}$Zr$_{58}$ and $^{100}$Zr$_{60}$ has been pointed out. In addition, a

detailed study of $^{110}$Zr$_{70}$, lying at the 3D-HO magic numbers $Z = 40$ and $N = 70$, showed no sign of stabilization of harmonic oscillator shell structure there [682].

It should be noticed that $^{96}$Zr$_{56}$ and $^{98}$Zr$_{58}$ represent two rare cases in which the $0_2^+$ band-head of a well-developed $K^\pi = 0^+$ band lies below the $2_1^+$ state of the ground state band [335]. It has been found [335] that such nuclei tend to be surrounded by nuclei exhibiting SC.

The $^{88-102}$Zr$_{48-62}$ isotopes have been (along with the Mo isotopes in this region) the textbook example used by Federman and Pittel [470–473] for connecting nuclear deformation to the enhanced proton–neutron interaction among certain Nilsson orbitals, as described in Section 2.6. The abrupt change in the spectra of the Zr isotopes can be seen in Figure 1 of [470], Figure 1 of [472], and Figure 2 of [473], in which the large energy drop of the $2_1^+$ and $0_2^+$ levels when passing from $^{98}$Zr$_{58}$ to $^{100}$Zr$_{60}$ is clearly seen, signifying both the sudden increase of deformation (indicated by the drop of $2_1^+$) and the onset of SC (indicated by the drop of $0_2^+$), offering a dramatic manifestation of the close connection between SPTs and SC, discussed later in detail by García-Ramos and Heyde [439,683]. It is then not surprising that $^{100}$Zr$_{60}$ has been suggested [350] as a candidate for the X(5) CPS.

Calculations focused on $^{98,100}$Zr$_{58,60}$ have been carried out both in the mean-field framework using the Skyrme interaction [69,146] and the complex excited VAMPIR approach [190], as well as in the MCSM framework [66,69] and within a simple two-state-mixing model [684], confirming both the rapid change of structure occurring between $^{98}$Zr$_{58}$ and $^{100}$Zr$_{60}$, as well as configuration mixing and SC in the latter. Furthermore, triple SC has been predicted in $^{98}$Zr$_{58}$ [190] and $^{96}$Zr$_{56}$ [196] within the complex excited VAMPIR approach, while SC in $^{96}$Zr$_{56}$ has been corroborated through calculations [287,685] using a quadrupole collective Bohr Hamiltonian, later extended to the $^{92-102}$Zr$_{52-62}$ region [288]. Recent calculations using CDFT with the DDME2 functional with standard pairing included, have found that neutron particle–hole excitations induced by the protons occur in $^{98,100}$Zr$_{58,60}$ [579,580].

Within the last decade, calculations covering the whole $N = 50$–70 region, or most of it, have been performed within several different theoretical frameworks.

MCSM calculations [71,679] have corroborated the sharp SPT taking place at $N = 60$, and in addition have exhibited the difference between the SC occurring in $^{100}$Zr$_{60}$, in which a prolate band coexists with an oblate one, and in $^{110}$Zr$_{70}$, in which a prolate band coexists with a triaxial one. The crucial role played by the tensor force, discussed in Section 2.1, has been emphasized.

Extended CDFT calculations [150,155,166] have found SC in the $^{98-110}$Zr$_{58-70}$ region, while in contrast the absence of SC in $^{88-92}$Zr$_{48-52}$ has been demonstrated [150]. Similar results have been obtained using a 5DQCH [242] or an IBM Hamiltonian [221], with the relevant parameters in both cases obtained from CDFT.

Extended calculations [438,439] have also been performed within the IBM-CM framework, confirming the abrupt structural change from spherical to deformed shapes occurring at $^{100}$Zr$_{60}$ and the close connection between SPT and SC. SC in $^{100-110}$Zr$_{60-70}$ is confirmed, with the evolution of the intruder configuration across this region pointed out [439]. The latter point has been studied in detail in [440–442], suggesting that in the Zr isotopes two intertwined SPTs take place, sharing the same critical point at $^{100}$Zr$_{60}$. On one hand, the transition from normal (without particle–hole excitations) to intruder (involving particle–hole excitations) configurations takes place at $N = 60$, signaling also the beginning of the SC region extending up to $N = 70$. On the other hand, the nature of the intruder band itself is varying from nearly spherical in $^{92-98}$Zr$_{52-58}$ to prolate deformed in $^{102-104}$Zr$_{62-64}$ and finally to $\gamma$-unstable shapes in $^{106-110}$Zr$_{66-70}$. It is worth pointing out that the last point is in accordance to the results of MCSM calculations, finding SC of prolate and triaxial shapes in $^{110}$Zr$_{70}$ [71].

In summary, SC is well established both experimentally and theoretically in $^{94-100}$Zr$_{54-60}$, while theoretical predictions are extending it up to $^{110}$Zr$_{70}$.

8.1.2. Zr ($Z = 40$) Isotopes below $N = 50$

Early RMF calculations in the $^{76-90}$Zr$_{36-50}$ region have shown SC of a spherical and a deformed minimum in $^{76-82}$Zr$_{36-42}$ [156]. SC has been considered in $^{80}$Zr$_{40}$ within the complex-excited VAMPIR framework [185,188], as well as within beyond-mean-field calculations involving the Gogny D1S interaction [138], revealing in it multiple SC of five $0^+$ states exhibiting different shapes. $^{80}$Zr$_{40}$ has also been suggested as a candidate for the X(5) CPS [350]. Total-Routhian-surface calculations [686] have revealed SC in the region $^{80-84}$Zr$_{40-44}$. The same holds for Skyrme HF calculations [69] in $^{80-84}$Zr$_{40-44}$ (see Figure 9 of [69]). Recent calculations using CDFT with the DDME2 functional with standard pairing included, have found that neutron particle–hole excitations induced by the protons, as well as proton particle–hole excitations induced by the neutrons, occur in $^{78,80}$Zr$_{38,40}$ [579,580].

In summary, several theoretical predictions support the occurrence of SC in $^{76-84}$Zr$_{36-44}$.

*8.2. The Sr ($Z = 38$) Isotopes*

8.2.1. Sr ($Z = 38$) Isotopes above $N = 50$

Strong experimental evidence for SC in $^{96,98}$Sr$_{58,60}$ has been accumulated [89,531,673, 687–692], while signs of an abrupt shape transition from $N = 58$ to 60 have been identified [687–689,691].

Early theoretical calculations using a two-state mixing model [684] and the VAMPIR mean-field approach [190] have been focused on SC in $^{98}$Sr$_{60}$ and $^{96}$Sr$_{58}$, respectively, finding triple SC in the latter. More recently, extended calculations have been carried out, covering large parts of the region $^{88-108}$Sr$_{50-70}$, using the RHB formalism [150], a 5DQCH with parameters determined from CDFT [242], the IBM with parameters determined from a Gogny energy density functional [221], and the IBM-CM [445]. In all cases an abrupt change of structure is seen between $^{96}$Sr$_{58}$ and $^{98}$Sr$_{60}$, with SC predicted then up to $^{108}$Sr$_{70}$.

In MCSM calculations [691], in which the SPT occurring at $N = 60$ is attributed to type II shell evolution, proton particle–hole excitations from the $pf$ shell into the $1g_{9/2}$ orbital are seen as the mechanism leading to SC. This means that p-h excitations across the $Z = 40$ subshell closure are supposed to play the major role in creating SC, in contrast to the opinion expressed in the review article [4] (see p. 1486 of [4]), that subshell structure does not play a major role in nuclear collectivity, since it is "smeared out" by the pairing interaction into a uniform distribution. The same reservation has been expressed in the review article [8] (see p. 71 of [8]) in relation to the similar region of SC occurring around $N = 90$, $Z = 64$.

In conclusion, SC is well established both experimentally and theoretically in $^{96,98}$Sr$_{58,60}$, while theoretical predictions are extending it up to $^{108}$Sr$_{70}$.

8.2.2. Sr ($Z = 38$) Isotopes below $N = 50$

Experimental evidence for SC has been seen in $^{82}$Sr$_{44}$ [693], while $^{76,78}$Sr$_{38,40}$ have been considered as possible X(5) candidates [350]. Along the $N = Z$ line collectivity has been found [689] to increase strongly between $^{64}$Ge$_{32}$ and $^{76}$Sr$_{38}$.

Early RMF calculations in the $^{74-88}$Sr$_{36-50}$ region have shown SC of a spherical and a deformed minimum in $^{74-80}$Sr$_{36-42}$ [156]. SC has been considered in the complex excited VAMPIR framerwork in $^{74}$Sr$_{36}$ [192,195] and in $^{76,78}$Sr$_{38,40}$ [185,188]. SC of spherical, prolate, and oblate shapes has been found by total-Routhian-surface calculations in $^{76-80}$Sr$_{38-42}$ [686], while shell-model calculations along the $N = Z$ line have shown [60] that a sudden jump of nucleons into the $1g_{9/2}2d_{5/2}$ orbitals occurs at $N = Z = 36$, thus supporting SC in $^{76}$Sr$_{38}$. Recent calculations using CDFT with the DDME2 functional with standard pairing included, have found that neutron particle–hole excitations induced by the protons, as well as proton particle–hole excitations induced by the neutrons, occur in $^{78}$Sr$_{40}$ [579,580].

In summary, SC is supported experimentally and/or theoretically in $^{74-82}$Sr$_{36-44}$.

*8.3. The Mo (Z = 42) Isotopes*

8.3.1. Mo (Z = 42) Isotopes above $N = 50$

Clear experimental evidence for SC has been reported in $^{98}$Mo$_{56}$ [229,230], $^{100}$Mo$_{58}$ [672,674,675], $^{102}$Mo$_{60}$ [694]. $^{104}$Mo$_{62}$ has been suggested [350,353] as a candidate for the X(5) CPS, based on energy spectra, but B(E2)s were found [351] to support a rather rotational behavior. The recent observation of low-lying $0_2^+$ states in $^{108,110}$Mo$_{66,68}$ [695] appears to extend the SC region up to $N = 68$. The lower border of the SC region appears to be $^{98}$Mo$_{56}$, since no evidence of SC has been seen in $^{96}$Mo$_{54}$ [230]. Experimental information reviewed in [8,517,689] supports these borders. The energy of the $0_2^+$ states appears to be dropping fast up to $N = 56$, being nearly stabilized up to $N = 60$ and then increasing very slowly up to $N = 68$ (see Figure 7 of [230] and Figure 29 of [8]).

The $^{92-106}$Mo$_{50-64}$ isotopes have been (along with the Zr isotopes in this region) the textbook example used by Federman and Pittel [470–473] for connecting nuclear deformation to the enhanced proton–neutron interaction among certain Nilsson orbitals, as described in Section 2.6. The abrupt change in the spectra of the Mo isotopes can be visualized in Figure 5 of [473], in which the large energy drop of $0_2^+$ when passing from $^{96}$Mo$_{54}$ to $^{98}$Mo$_{56}$ is clearly seen, signifying the onset of SC at $N = 56$. The drop of the $2_1^+$ state along the Mo chain of isotopes is gradual, in contrast to the Zr chain of isotopes, in which a dramatic drop is seen between $N = 56$ and 58.

Extended theoretical calculations covering the $^{92-112}$Mo$_{50-70}$ region, or even going to higher $N$, have been performed recently using CDFT [149,155,166], predicting SC in the $^{98-108}$Mo$_{56-66}$ region. Similar results have been obtained using a 5DQCH with parameters determined through CDFT using the PC-PK1 parametrization [242], and with an IBM Hamiltonian with parameters determined through CDFT using the Gogny-D1M energy density functional [221]. Recent calculations using CDFT with the DDME2 functional with standard pairing included have found that neutron particle–hole excitations induced by the protons occur in $^{102}$Mo$_{60}$ [579,580].

The shape mixing evolution in the $^{96-100}$Mo$_{54-58}$ has also been described [284] in terms of a Bohr Hamiltonian with a double-well sextic potential. The need for a double-well potential is related to a first order SPT (see, for example, Figure 10 of [449]).

In summary, SC is well established both experimentally and theoretically in $^{98-104}$Mo$_{56-62}$, while theoretical predictions are extending it up to $^{110}$Mo$_{68}$.

8.3.2. Mo (Z = 42) Isotopes below $N = 50$

Theoretical predictions for the occurrence of SC have been provided by complex excited VAMPIR calculations [185] in $^{84}$Mo$_{42}$, as well as by total-Routhian-surface calculations [686] in $^{84,86}$Mo$_{42,44}$.

*8.4. The Ru (Z = 44) Isotopes*

Despite the proximity of the Ru (Z = 44) isotopes to the Mo (Z = 42) ones, in which SC has been observed, there is very little evidence for SC in the Ru isotopes. The evolution of the $0_2^+$ states in the Ru isotopes (see Figure 32 of [8]) looks very similar to the corresponding plot for the Mo isotopes (see Figure 29 of [8]), dropping rapidly up to $^{98}$Ru$_{54}$, then dropping slowly up to $^{102}$Ru$_{58}$, then raising slowly up to $^{110}$Ru$_{66}$. The Ru isotopes beyond $N = 60$ appear to be certainly deformed [517], while signs of triaxiality have been seen in $^{110}$Ru$_{66}$ [689]. In parallel, the presence of multiphonon spherical vibrations in the Ru isotopes has been recently completely ruled out [696].

On the theoretical side, early investigations [423,469] led to the conclusion that $^{110}$Ru$_{66}$ presents a regular behavior, and therefore shows no signs of SC. $^{104}$Ru$_{60}$ has been tested as a candidate for the E(5) CPS by IBM [389] and adiabatic TDHFB [181] methods, but no firm evidence supporting this suggestion has been found.

Recently, extended calculations have been performed, covering the region $^{96-124}$Ru$_{52-80}$, or most of it, using an IBM Hamiltonian with parameters fixed through microscopic calculations involving the Skyrme force [207] and the Gogny energy density functional [221],

with no signs of SC found in the Ru isotopes, in contrast to the Sr, Zr, and Mo isotopes considered within the same framework [221]. The difference can be clearly seen from the configurations used in order to obtain satisfactory agreement with the data, reported on page 5 of [221]. While the mixing of two configurations was required in the case of the Mo isotopes, and mixing of three configurations was found necessary in the case of the Zr and Sr isotopes, a single configuration sufficed in the case of the Ru isotopes.

Extended calculations within the same wide region have been recently performed also using CDFT [149,166]. In both cases, only $^{104}Ru_{60}$ has been singled out as a possible candidate for SC. It is of interest to compare Tables I and II of Ref. [149]. While for most of the Mo isotopes considered in that study, two configurations have been involved (as seen in Table II of [149]), in the case of the Ru isotopes only one configuration was found, as seen in Table I of [149], with the sole exception of $^{104}Ru_{60}$, in which two configurations have been found.

In summary, only $^{104}Ru_{60}$ appears as a possible candidate for SC, in contrast to the Sr, Zr, and Mo series of isotopes, in which several candidates have been found.

### 8.5. Shape Coexistence and Shape/Phase Transition at $N \approx 60$

In Figure 10 the ground-state bands and the intruder levels are plotted for $N = 58$, 60, 62 in the $Z \approx 40$ region.

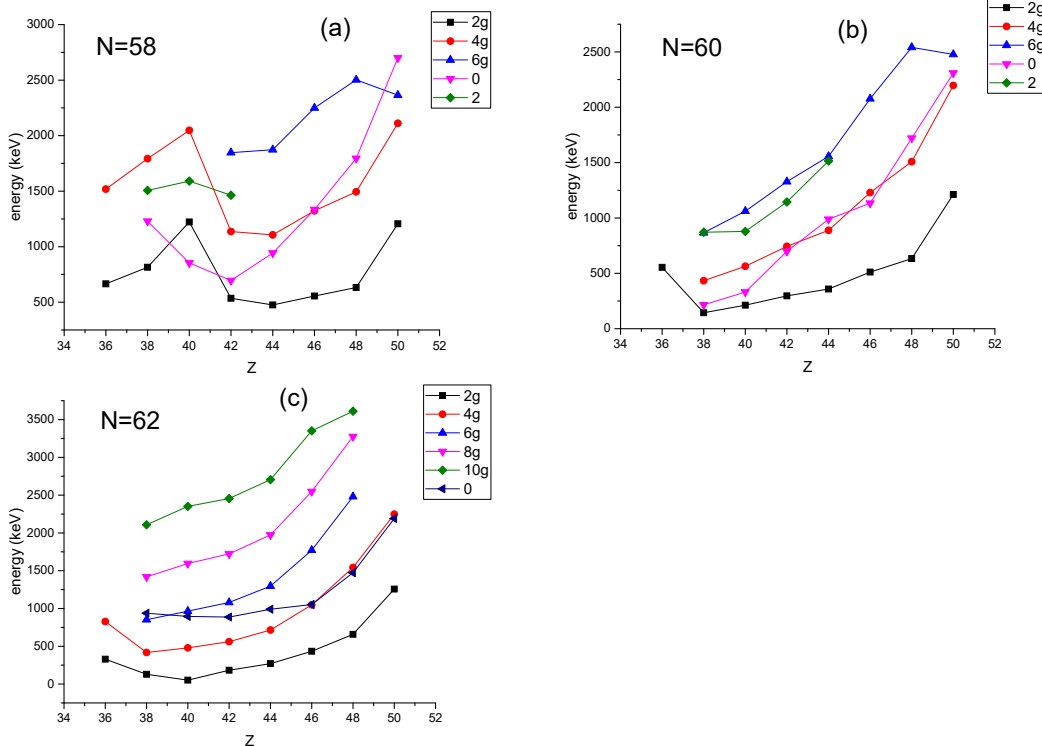

**Figure 10.** Experimental energy levels in keV [406] for the $N = 58$ (**a**), $N = 60$ (**b**), and $N = 62$ (**c**) series of isotones. The levels of the ground-state band bear the subscript g, while the levels of the excited $K^\pi = 0^+$ band bear no subscript. See Section 8.5 for further discussion.

In Figure 10a we see that for $N = 58$ the intruder levels form a parabola, presenting a minimum at $Z = 42$, i.e., near the proton midshell ($Z = 39$). In Figure 10b the same happens for $N = 60$ at $Z = 38$, while for $N = 62$ the minimum seems to be flat, as seen in Figure 10c.

In the case of $N = 58$, the intruder $0_2^+$ states in the region $Z = 38$–44 fall below the $4_1^+$ members of the ground state band (gsb) and approach the $2_1^+$ levels, with $^{98}Zr_{58}$ representing a rare case in which the $0_2^+$ state lies below the $2_1^+$ state, while for $Z = 46$–50 they stay close to the $4_1^+$ state. In the case of $N = 60$, the intruder $0_2^+$ states at $Z = 38$–40 are nearly degenerate with the $2_1^+$ members of the gsb, while at $Z = 42$-50 are nearly

degenerate with the $4_1^+$ members of the gsb. In the case of $N = 62$, the intruder $0_2^+$ states at $Z = 38$–44 are already lying far above the $4_1^+$ members of the gsb at $Z = 38$–44, still remaining below the $6_1^+$ members of the gsb though, but they are almost degenerate to the $4_1^+$ members of the gsb at $Z = 46$–50.

The evolution seen in Figure 10 from $N = 58$ to 62 is very similar to the evolution seen in Figure 8 from $N = 88$ to 92, the main difference being that the intruder levels in Figure 10 appear to have sank lower within the gsb in comparison to what is seen in Figure 8. As a consequence, while for $N = 90$ near degeneracy is seen between the $0_2^+$ and $6_1^+$ states in the region $Z = 60$–66, in the present case for $N = 60$ one sees degeneracy of the $0_2^+$ state to the $2_1^+$ state for $Z = 38, 40$ and to the $4_1^+$ state for $Z = 42$–50.

Several comments are in place.

The parabolas appearing in Figure 10 vs. $Z$ for the $N \approx 60$ isotones are similar to the parabolas appearing in Figures 7 and 9 vs. $N$, for the $Z \approx 82$ and $Z \approx 50$ isotopes, respectively. In all cases the minima of the parabolas appear near the relevant midshell. The similarity of the data suggests similarity in the microscopic mechanism leading to the appearance of these effects. In the cases of the $Z \approx 82$ and $Z \approx 50$ isotopes, it is known [83–87] that the microscopic origin of the intruder states is lying in proton particle–hole excitations across the $Z = 82$ and $Z = 50$ shell closures, respectively. Therefore, it is plausible that in the present case of the $N \approx 60$ isotones the underlying microscopic origin should be related to neutron particle–hole excitations. Indeed, in [579,580] it has been suggested that in this case neutron particle–hole excitations occur from normal parity orbitals to intruder orbitals, which of course bear the opposite parity. It should be noticed that the particle–hole excitations occurring across the $Z = 82$ and $Z = 50$ major shells, also correspond to particle–hole excitations between orbitals of opposite parity, since the orbitals involved belong to different major shells.

The appearance of particle–hole excitations across $Z = 82$ and $Z = 50$ for relevant series of isotopes can be attributed to the influence of the changing number of neutrons on the constant set of protons. Thus SC appearing in these cases has been called *neutron-induced SC* [579,580]. In a similar manner, particle–hole excitations across $N = 60$ for relevant series of isotones can be attributed to the influence of the changing number of protons on the constant set of neutrons. Thus SC appearing in these cases has been called *proton-induced SC* [579,580]. This interpretation is consistent with the role played in the shell-model framework by the tensor force [102,105–107], which is known to be responsible for the influence of the protons on the neutron gaps, as well as the influence of the neutrons on the proton gaps, as described in Section 2.1.1.

The fact that in practically all cases for $N = 58$–62 and $Z = 38$–50 the intruder $0_2^+$ levels remain below the $6_1^+$ members of the gsb, or are almost degenerate to them, suggests that the appearance of SC in all these nuclei is plausible. In different words, $^{96-100}\text{Sr}_{58-62}$, $^{98-102}\text{Zr}_{58-62}$, $^{100-104}\text{Mo}_{58-62}$, $^{102-106}\text{Ru}_{58-62}$, $^{104-108}\text{Pd}_{58-62}$, $^{106-110}\text{Cd}_{58-62}$, and $^{108-112}\text{Sn}_{58-62}$, are good candidates for the appearance of SC.

The strong similarity between the $Z \approx 40$, $N \approx 60$ region and the $Z \approx 60$, $N \approx 90$ region has a further consequence. The $N = 90$ isotones $^{150}\text{Nd}_{90}$, $^{152}\text{Sm}_{90}$, $^{154}\text{Gd}_{90}$, and $^{156}\text{Dy}_{90}$ are known to be very good examples of the X(5) CPS, occurring at the critical point of a first order SPT from spherical to prolate deformed shapes. Because of the similarities mentioned above, it should be expected that in the region with $Z = 38$–42 and $N = 58$–62, i.e., among $^{96-100}\text{Sr}_{58-62}$, $^{98-102}\text{Zr}_{58-62}$, and $^{100-104}\text{Mo}_{58-62}$, some good examples of this SPT should occur, thus providing further connection between the SC and SPT concepts, as proposed by Heyde et al. [438,439,445,469,683]. However, it should be pointed out that the numerical hallmarks of this critical point symmetry might be different from those of X(5), since in the $Z \approx 40$, $N \approx 60$ region the $0_2^+$ state appears to be closer to the $2_1^+$ and/or $4_1^+$ states and not nearly degenerate to the $6_1^+$ state. In other words, some formulation of the CPS more flexible than X(5) would be required in the $Z \approx 40$, $N \approx 60$ region.

The above observations are compatible with past findings regarding the $^{104}\text{Mo}_{62}$ nucleus, which had been suggested [353] as an example of X(5) based on spectral char-

acteristics, but later measurements of B(E2) transition rates [351] have shown that the $^{104,106}$Mo$_{62,64}$ isotopes are of rotational and not of critical nature.

It should be recalled that the strong similarity between the $N \approx 90$, $Z \approx 64$ and the $N \approx 60$, $Z \approx 40$ regions has been pointed out by Casten et al. already in 1981 [630], suggesting a common microscopic interpretation of the onset of deformation in these two regions.

## 9. The $Z \approx 34$, $N \approx 40$ Region

### 9.1. The Kr (Z = 36) Isotopes

9.1.1. Kr (Z = 36) Isotopes below $N = 50$

Coexistence of a prolate and an oblate band has been seen experimentally in the region $^{72-76}$Kr$_{36-40}$ [689,697–704], recently extended to $^{70}$Kr$_{34}$ [705,706]. Plotting in Figure 11d the $0_2^+$ states [406] in the region $^{72-86}$Kr$_{36-50}$, one sees a rapid fall with decreasing $N$, reaching a plateau at $^{72-76}$Kr$_{36-40}$, which looks like the bottom of a parabola, of which the left branch is still missing. It should be noticed that $^{72}$Kr$_{36}$ is one of the very few nuclei in which the $0_2^+$ state falls below the $2_1^+$ state.

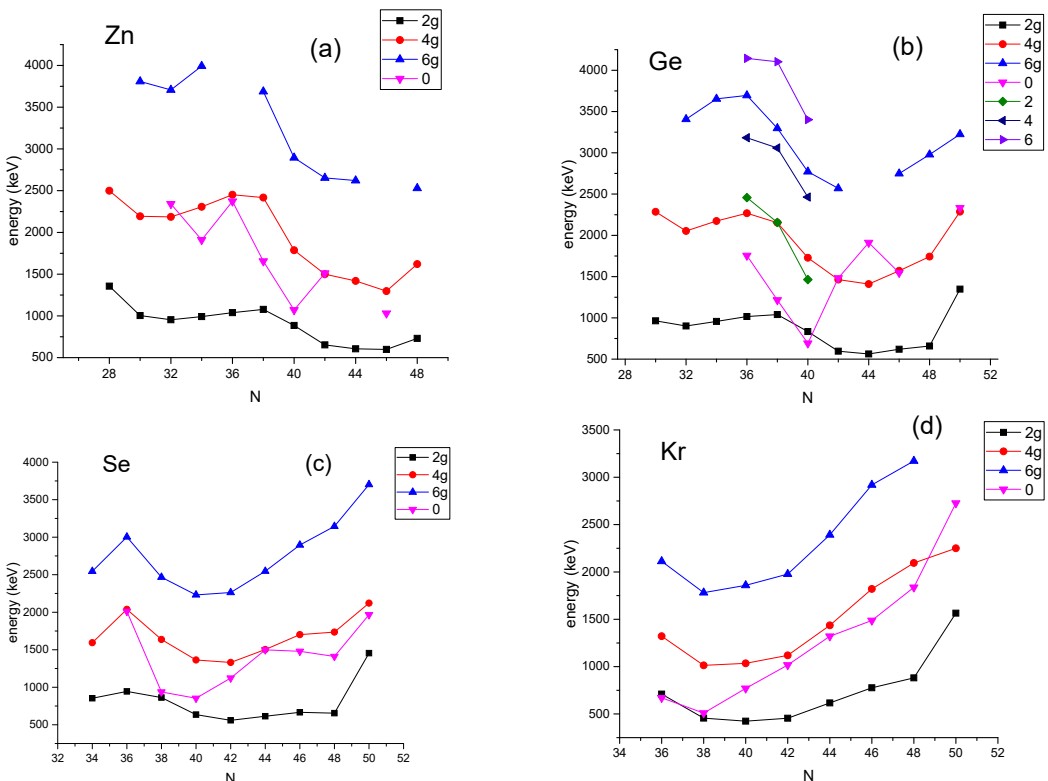

**Figure 11.** Experimental energy levels in keV [406] for the Zn ($Z = 30$) (**a**), Ge ($Z = 32$) (**b**), Se ($Z = 34$) (**c**), and Kr ($Z = 36$) (**d**) series of isotopes. The levels of the ground state band bear the subscript g, while the levels of the excited $K^\pi = 0^+$ band bear no subscript. See Sections 9.1–9.4 for further discussion.

Theoretical calculations have been focused on the $^{70-76}$Kr$_{34-40}$ region. Shell-model calculations have shown the oblate shape of the ground state and the prolate–oblate SC in $^{72}$Kr$_{36}$ [61], as well as the importance of 4p-4h excitations from $(2p_{3/2}1f_{5/2})$ to $(2p_{1/2}1g_{9/2})$ for the oblate configurations. The sudden structural change occurring along the $N = Z$ line when passing from $N = Z \leq 35$ to $N = Z \geq 36$ is attributed to a sudden jump of nucleons into the $1g_{9/2}2d_{5/2}$ orbitals [60]. It has been shown [65] that this jump favors the creation of configurations with large prolate deformation, while configurations within the $pf$ shell favor oblate deformation. The role of the tensor force in keeping these two

different configurations close in energy, which is a prerequisite for obtaining SC, has been clarified [65]. The mixing between the oblate and prolate configurations in $^{72}$Kr$_{36}$ has also been discussed in [46].

SC in $^{70-74}$Kr$_{34-38}$ has been extensively discussed in the framework of the excited VAMPIR [184,185] and the complex excited VAMPIR [186,188,191–195]. A microscopic quadrupole collective Hamiltonian has been derived considering large amplitude collective motion [198,203,707], and SC and prolate–oblate mixing in $^{72-76}$Kr$_{36-40}$ have been considered in it (see Figures 6, 8 and 11 of [203] and Figure 2 of [198] for the relevant level schemes). Mean-field calculations for $^{70-78}$Kr$_{34-42}$ have been performed with the Skyrme interaction [141] and the Gogny D1S interaction [132,134] (for relevant level schemes see Figures 5, 9, and 13 of [141] and Figure 2 of [132]), as well as with a Bohr Hamiltonian with a sextic potential possessing two minima [279,708] (see Figure 5 of [279] and Figure 3 of [708] for relevant level schemes).

In addition, extensive mean-field calculations covering the whole $^{70-98}$Kr$_{34-62}$ region have been performed [139] with the Gogny D1S interaction (see Figures 6–9 of [139] for relevant level schemes). SC is seen in $^{70-78}$Kr$_{34-42}$, and then again in $^{94-98}$Kr$_{58-62}$, with a gap with no SC appearing in between ($N = 44$–56).

Extensive calculations covering the whole $^{70-100}$Kr$_{34-64}$ region have been performed [150] in the relativistic HF framework with the DD-PC1 parametrization, leading to two coexisting minima in $^{72-78}$Kr$_{36-42}$ and then again in $^{92,96-100}$Kr$_{56,60-64}$ (see Table I of [150]), as well as with the DD-ME2 parametrization, leading to two coexisting minima in $^{70-78}$Kr$_{34-42}$ and then again in $^{92-100}$Kr$_{56-64}$ (see Table II of [150]). Once more, there is some dependence on the details of the interactions, but they both agree that SC is observed roughly in the regions $N = 34$–42 and $N = 56$–64, with a large gap with no SC in between.

Extensive calculations covering the whole $^{70-100}$Kr$_{34-64}$ region have also been performed [224] using an IBM Hamiltonian with parameters determined from a Gogny energy density functional. Two coexisting oblate and prolate minima have been found in $^{70,78}$Kr$_{34,42}$, and then again in $^{92-100}$Kr$_{56-64}$, while triple SC of spherical, prolate and oblate minima has been seen in $^{72-76}$Kr$_{36-40}$ (see Table I of [224]). The same gap without SC in the region $N = 44$–54 is seen, as in [150]. Level schemes can be seen in Figures 7–9 of [224].

9.1.2. Kr ($Z = 36$) Isotopes above $N = 50$

Experimental evidence for a smooth onset of deformation has been seen in $^{94-96}$Kr$_{58-60}$ [709], with the low-$Z$ boundary of the island of deformation seen at $^{96}$Kr$_{60}$ [710]. Oblate-prolate SC has been in $^{98}$Kr$_{62}$ [711]. A relevant review has been given in [689].

On the theoretical side, extensive RMF calculations have been performed [242] in the $^{86-104}$Kr$_{50-68}$ region, using a 5DQCH with parameters determined from the PC-PK1 energy density functional. In addition, SC in $^{94}$Kr$_{58}$ has been considered in the complex excited VAMPIR approach [194].

As mentioned above, extensive mean-field calculations covering the whole $^{70-98}$Kr$_{34-62}$ region have been performed [139] with the Gogny D1S interaction, showing SC in $^{94-98}$Kr$_{58-62}$ (see Figure 9 of [139] for relevant level schemes).

Again, as mentioned above, extensive calculations covering the whole $^{70-100}$Kr$_{34-64}$ region have been performed [150] in the relativistic HF framework with the DD-PC1 parametrization, leading to SC in $^{92,96-100}$Kr$_{56,60-64}$ (see Table I of [150]), as well as with the DD-ME2 parametrization, leading to SC in $^{92-100}$Kr$_{56-64}$ (see Table II of [150]). Once more, there is some dependence on the details of the interactions, but they both agree that SC is observed roughly in the region $N = 56$–64.

Again, as mentioned above, extensive calculations covering the whole $^{70-100}$Kr$_{34-64}$ region have also been performed [224] using an IBM Hamiltonian with parameters determined from a Gogny energy density functional. SC is seen in $^{92-100}$Kr$_{56-64}$ (see Table I of [224]). Relevant level schemes can be seen in Figures 8, 9 of [224].

Experimental evidence for SC has also been seen in $^{88}$Kr$_{52}$ [712], as well as in $^{86}$Se$_{52}$ and $^{84}$Ge$_{52}$. These observations suggest that the $N = 50$ island of inversion [534] is extended beyond $^{78}$Ni$_{50}$.

In conclusion, SC is well established both experimentally and theoretically in the region $^{70-76}$Kr$_{34-40}$, while in addition SC related to the $N = 50$ island of inversion is seen in $^{88}$Kr$_{52}$ and both experimental and theoretical signs for the beginning of a new region of SC starting at $N = 60$ have been seen in $^{94-100}$Kr$_{58-64}$.

### 9.2. The Se (Z = 34) Isotopes

#### 9.2.1. Se (Z = 34) Isotopes below $N = 50$

Coexistence of a nearly spherical ground state band with a prolate deformed band built on the $0_2^+$ state (at 937 keV) has been first observed in 1974 by Hamilton et al. [713,714] in $^{72}$Se$_{38}$. Later experiments pointed to a slightly oblate shape for the ground state [715,716], while negative parity bands have also been observed [717]. An up-to-date level scheme can be seen in Figure 1 of [718]. A very similar situation, with a prolate deformed band built on the $0_2^+$ state (at 854 keV) and a near-spherical ground state has been seen also in $^{74}$Se$_{40}$ [719,720], while in $^{68}$Se$_{34}$ an excited prolate deformed band built on $0_2^+$ and a slightly oblate ground state band have been seen [715,721,722]. A similar picture is seen in $^{70}$Se$_{36}$ [706,715]. A short review of the experimental situation in $^{68-72}$Se$_{34-38}$ has been given in [689]. An oblate ground state has also been seen in $^{66}$Se$_{32}$ [723].

The $0_2^+$ states in the region $^{68-82}$Se$_{34-48}$, plotted vs. $N$, form a parabola with its bottom located at $N = 38, 40$ (see Figure 17 of [724] and Figure 11c), while the $0_2^+$ states of the $N = 40$ isotones in the $Z = 28-36$ region also appear to form a parabola with its bottom lying at $Z = 32-36$ (see Figure 18 of [724] and Figure 12b). These curves support the occurrence of SC in the region around $Z = 34$, $N = 40$.

Theoretical calculations have been focused on the $^{68-74}$Se$_{34-40}$ region. Shell-model calculations have shown the oblate shape of the ground state and the prolate–oblate SC in $^{68}$Se$_{34}$ [61], as well as the importance of 4p-4h excitations from $(2p_{3/2}1f_{5/2})$ to $(2p_{1/2}1g_{9/2})$ for the oblate configurations. The sudden structural change occurring along the $N = Z$ line when passing from $N = Z \leq 35$ to $N = Z \geq 36$ is attributed to a sudden jump of nucleons into the $1g_{9/2}2d_{5/2}$ orbitals [60]. A systematic study of $^{68-74}$Se$_{34-40}$ has been performed in the shell-model framework using the $2p1f_{5/2}1g_{9/2}$ model space [64], taking into account a Gaussian central force and the tensor force. A simple two-state mixing model [90] has shown that the excited $0_2^+$ band is significantly more collective than the ground state band in $^{76}$Se$_{42}$.

SC in $^{68-72}$Se$_{34-38}$ has been accounted for in the excited VAMPIR [182,183] and the complex excited VAMPIR [189,191,193,195] mean-field approaches. The oblate–prolate shape mixing in these nuclei has also been described within the adiabatic self-consistent collective coordinate (ASCC) method [199]. Spectra and B(E2)s have been calculated using a 5DQCH (see Figures 10, 12, and 14 of [200] for the relevant level schemes and [134] for a relevant review).

Spectra and B(E2)s in the extensive region $^{68-96}$Se$_{34-62}$ have been calculated [223] using an IBM Hamiltonian with parameters determined from the Gogny-D1M energy density functional. SC has been seen in $^{72,74}$Se$_{38,40}$, and also in the neutron-rich $^{94}$Se$_{60}$, which will be discussed in Section 9.2.2 (see Figures 15, 17 and 18 of [223] for the relevant level schemes).

Finally, SC in $^{72-76}$Se$_{38-42}$ has been described [283] using a Bohr Hamiltonian with a sextic potential possessing two unequal, competing minima (see Figure 3 of [283] for the potentials and Figure 1 of [283] for the relevant level schemes).

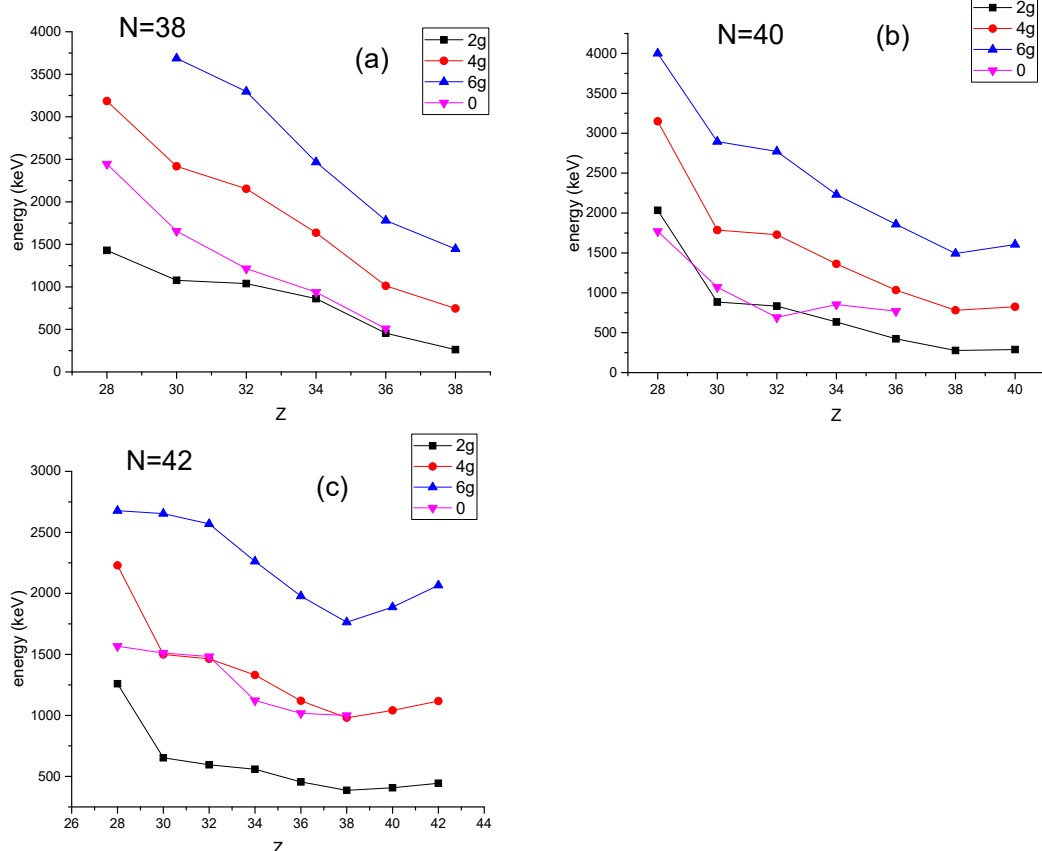

**Figure 12.** Experimental energy levels in keV [406] for the $N = 38$ (**a**), $N = 40$ (**b**), and $N = 42$ (**c**) series of isotones. The levels of the ground-state band bear the subscript g, while the levels of the excited $K^\pi = 0^+$ band bear no subscript. See Section 9.5 for further discussion.

9.2.2. Se ($Z = 34$) Isotopes above $N = 50$

Experimental signs of SC have been seen in $^{86}$Se$_{52}$ [712], while in $^{88}$Se$_{54}$ excited states have been detected [725], indicating increased collectivity in relation to $^{86}$Se$_{52}$. A transition from $\gamma$-soft to rigid oblate ground state is seen between $^{92}$Se$_{58}$ and $^{94}$Se$_{60}$ [726]. Low-lying structures and shape evolution in the region $^{88-94}$Se$_{54-60}$ have been considered in [727].

As mentioned above, spectra and B(E2)s in the extensive region $^{68-96}$Se$_{34-62}$ have been calculated [223] using an IBM Hamiltonian with parameters determined from the Gogny-D1M energy density functional, with SC seen in $^{94}$Se$_{60}$ (see Figure 18 of [223] for the relevant level scheme).

In conclusion, SC is well established both experimentally and theoretically in the region $^{68-74}$Se$_{34-40}$, while signs of SC have been seen in $^{86}$Se$_{52}$, related to the $N = 50$ island of inversion, and in $^{94}$Se$_{60}$, signing the beginning of a new region of SC starting at $N = 60$.

*9.3. The Ge ($Z = 32$) Isotopes*

9.3.1. Ge ($Z = 32$) Isotopes below $N = 50$

Experimental evidence for SC in $^{68-74}$Ge$_{36-42}$ has been reported in [728], in which a transition from spherical to deformed shapes is also seen between $^{72}$Ge$_{40}$ and $^{74}$Ge$_{42}$. Further evidence for SC in $^{70}$Ge$_{38}$ and $^{72}$Ge$_{40}$ has been reported in [626] and [729] respectively. Experimental evidence is reviewed in [724], in Figure 10 of which the $0_2^+$ states of $^{68-78}$Ge$_{36-46}$ are seen to exhibit a parabolic behavior with a sharp minimum at $N = 40$ (see also Figure 11b).

It should be noticed (see Figure 11b) that $^{72}$Ge$_{40}$ represents a rare example in which the $0_2^+$ band-head of a well-developed $K^\pi = 0^+$ band lies below the $2_1^+$ state of the ground

state band [335]. It has been found [335] that such nuclei tend to be surrounded by nuclei exhibiting SC.

Theoretical corroboration for SC in $^{68}$Ge$_{36}$ and $^{72}$Ge$_{40}$ has been given by mean-field calculations in the excited VAMPIR approach [182–184]. Earlier shell-model calculations [63,70], corroborating SC in $^{72}$Ge$_{40}$, have been recently extended [64] to the whole $^{64–76}$Ge$_{32–44}$ region, emphasizing the role played by the tensor force and supporting SC in $^{68–76}$Ge$_{36–44}$, as seen in Figures 15–19 of [64]. SC of spherical and oblate shapes in $^{72}$Ge$_{40}$ has also been found in extended IBM calculations with parameters determined through CDFT using the Gogny-D1M energy density functional [223]. IBM calculations in $^{64–80}$Ge$_{32–48}$ have also been carried out and compared to solutions of a $\gamma$-unstable Bohr Hamiltonian with a Davidson potential, with both approaches giving satisfactory results [730]. SC in $^{74}$Ge$_{42}$ has also been studied [279] by a Bohr Hamiltonian with a sextic potential in the $\beta$ variable, exhibiting two unequal minima at different deformations, as seen in Figure 6 of [279]. A similar Bohr calculation [280] in $^{80}$Ge$_{48}$ has shown absence of SC in this isotope.

Shell-model calculations [61] for $N = Z$ nuclei in the $Z = 30$–36 region have found coexistence of prolate and oblate shapes in $^{68}$Se$_{34}$ and $^{72}$Kr$_{36}$, but not in $^{60}$Zn$_{30}$ and $^{64}$Ge$_{32}$.

### 9.3.2. Ge ($Z = 32$) Isot4opes above $N = 50$

Experimental signs of SC in $^{82}$Ge$_{50}$ and $^{84}$Ge$_{52}$ have been given in [731] and [712], respectively. Signs of SC in $^{80}$Ge$_{48}$ given in [732,733] have been questioned by [734].

Gogny HFB, as well as Skyrme HF plus BCS calculations, have been carried out in $^{58–108}$Ge$_{26–76}$ [133], along with IBM calculations in $^{66–94}$Ge$_{34–62}$ with parameters determined through CDFT using the Gogny-D1M energy density functional [223]. In the latter calculation, nuclei exhibiting $\gamma$-soft shapes and coexistence of prolate and oblate shapes are seen in $52 \leq N \leq 62$ [224].

In summary, SC is experimentally and theoretically established in $^{68–76}$Ge$_{36–44}$, while evidence for SC is also seen in $^{80–94}$Ge$_{48–62}$.

### 9.4. The Zn ($Z = 30$) Isotopes

Experimental evidence for SC in $^{68,70}$Zn$_{38,40}$ has been seen in [724,735,736], backed by Nilsson–Strutinsky [735] and large-scale shell-model [736] calculations. The $0_2^+$ states in $^{66–72}$Zn$_{36–42}$ form a parabola with a minimum at $N = 40$ (see Figure 1 of [735], Figure 4 of [724], Figure 1 of [736], as well as Figure 11a).

Evidence for the robustness of the $Z = 50$ gap and the possibility of inversion of proton orbitals has been seen in $^{78,80}$Zn$_{48,50}$ [737,738], supported by evidence for SC in $^{79}$Zn$_{49}$ [739,740]. Large-scale shell-model [738–740] and MCSM [737] calculations support these findings.

Shell-model calculations [61] for $N = Z$ nuclei in the $Z = 30$–36 region have found coexistence of prolate and oblate shapes in $^{68}$Se$_{34}$ and $^{72}$Kr$_{36}$, but not in $^{60}$Zn$_{30}$ and $^{64}$Ge$_{32}$.

In summary, SC is established in $^{68,70}$Zn$_{38,40}$, related to the $N = 40$ island of inversion, while evidence for SC is also seen in $^{78–80}$Zn$_{48–50}$, related to the $N = 50$ IoI.

### 9.5. Shape Coexistence and Shape/Phase Transition at $N \approx 40$

In Figure 12 the ground state bands and the intruder levels are plotted for $N = 38$, 40, 42 in the $Z \approx 34$ region.

In Figure 12a we see that for $N = 38$, the intruder levels seem to form the left branch of a parabola, reaching a minimum at $Z = 36$, i.e., near the proton midshell ($Z = 34$) between the major shell closure at $N = 28$ and the subshell closure at $Z = 40$. The same happens in Figure 12b for $N = 40$ and in Figure 12c for $N = 42$. The difference among the three cases is that at $N = 38$ the intruder $0_2^+$ state starts between the $2_1^+$ and $4_1^+$ states at $N = 28$ and gradually approaches $2_1^+$, with which it becomes practically degenerate for $Z = 32$–36, while in the $N = 40$ case this degeneracy occurs in the whole $Z = 28$–34 region. In contrast, for $N = 42$, the $0_2^+$ state becomes practically degenerate with the $4_1^+$ state for $Z = 30$–38.

The evolution from lower $N$ to higher $N$ seen in Figure 12 for the $N \approx 40$ region strongly resembles the evolution seen in Figures 8 and 10 for the $N \approx 60$ and $N \approx 90$ regions, respectively, the main difference being that approximate degeneracy of the $0_2^+$ state is seen in $N = 90$ with the $6_1^+$ state, in $N = 60$ with the the $4_1^+$ state, and in $N = 40$ with the the $2_1^+$ state.

Several comments are in place.

In Sections 5.5 and 8.5 we have seen that the parabolas appearing at $N \approx 90$ and $N \approx 60$ have been attributed to neutron particle–hole excitations caused by the influence of the protons, leading to proton-induced SC. In addition, in Sections 3 and 7 we have seen that the parabolas appearing at $Z \approx 82$ and $Z \approx 50$ have been attributed to proton particle–hole excitations caused by the influence of the neutrons, leading to neutron-induced SC. It has been suggested [579,580] that in the region $Z = 38–40$, $N = 38–40$, both mechanisms are simultaneously active. This interpretation is consistent with the role played in the shell-model framework by the tensor force [102,105–107], which is known to be responsible for the influence of the protons on the neutron gaps, as well as the influence of the neutrons on the proton gaps, as described in Section 2.1.1.

The fact that in practically all cases for $N = 38–42$ and $Z = 28–38$ the intruder $0_2^+$ levels remain below the $4_1^+$ members of the gsb, or are almost degenerate to them, suggests that the appearance of SC in all these nuclei is plausible. In different words, $^{56-66}\text{Ni}_{28-38}$, $^{58-68}\text{Zn}_{28-38}$, $^{60-70}\text{Ge}_{28-38}$, $^{62-72}\text{Se}_{28-38}$, $^{64-74}\text{Kr}_{28-38}$, and $^{66-76}\text{Sr}_{28-38}$, are good candidates for the appearance of SC.

The strong similarity between the $Z \approx 34$, $N \approx 40$ region and the $Z \approx 40$, $N \approx 60$ and $Z \approx 60$, $N \approx 90$ regions has a further consequence. The $N = 90$ isotones $^{150}\text{Nd}_{90}$, $^{152}\text{Sm}_{90}$, $^{154}\text{Gd}_{90}$, and $^{156}\text{Dy}_{90}$ are known to be very good examples of the X(5) CPS, occurring at the critical point of a first-order SPT from spherical to prolate deformed shapes. Because of the similarities mentioned above, it should be expected that the $N = 40$ isotones could also be good candidates for the X(5) CPS, thus providing further connection between the SC and SPT concepts, as proposed by Heyde et al. [438,439,445,469,683]. However, it should be pointed out that the numerical hallmarks of this critical point symmetry might be different from those of X(5), since in the $Z \approx 34$, $N \approx 40$ region the $0_2^+$ states appears to be nearly degenerate with the $2_1^+$ states and not nearly degenerate to the $6_1^+$ states. In other words, the $N = 40$ isotones $^{68}\text{Ni}_{40}$, $^{70}\text{Zn}_{40}$, $^{72}\text{Ge}_{40}$, $^{74}\text{Se}_{40}$, and $^{76}\text{Kr}_{40}$ may be good candidates for the first order SPT, but some formulation of the CPS more flexible than X(5) would be required in the $Z \approx 34$, $N \approx 40$ region.

## 10. The Light Nuclei at and below $Z \approx 28$

### 10.1. The Ni (Z = 28) Isotopes

SC in this region has first been observed in $^{68}\text{Ni}_{40}$ [741–747] and $^{70}\text{Ni}_{42}$ [742,748,749], attributed to proton 2p-2h excitations across the $Z = 28$ shell. The early implications [750] that configuration mixing is needed in order to interpret the spectra of $^{64,66}\text{Ni}_{36,38}$ was followed by the observation of SC in these two isotopes, see [751,752] for $^{66}\text{Ni}_{38}$ and [532,753] for $^{64}\text{Ni}_{36}$. Existing evidence for SC in these isotopes is reviewed in [754], in which it is seen (in Figure 7 of [754]) that the $0_2^+$ states in $^{64-70}\text{Ni}_{36-42}$ resemble the left branch of a parabola reaching a minimum/plateau at $N = 40, 42$ (see Figure 13c).

The above observations indicate the occurrence of SC in the neighborhood of $Z = 28$ near the neutron 28–50 midshell ($N = 39$), in direct analogy to what has been seen in the neighborhood of $Z = 50$ near the neutron 50–82 mideshell ($N = 66$), as well as in the neighborhood of $Z = 82$ near the neutron 82–126 midshell ($N = 104$).

Recent experiments [755,756] suggest that SC might also appear in the doubly magic nucleus $^{78}\text{Ni}_{50}$, signifying a new island of inversion for $Z \leq 28$.

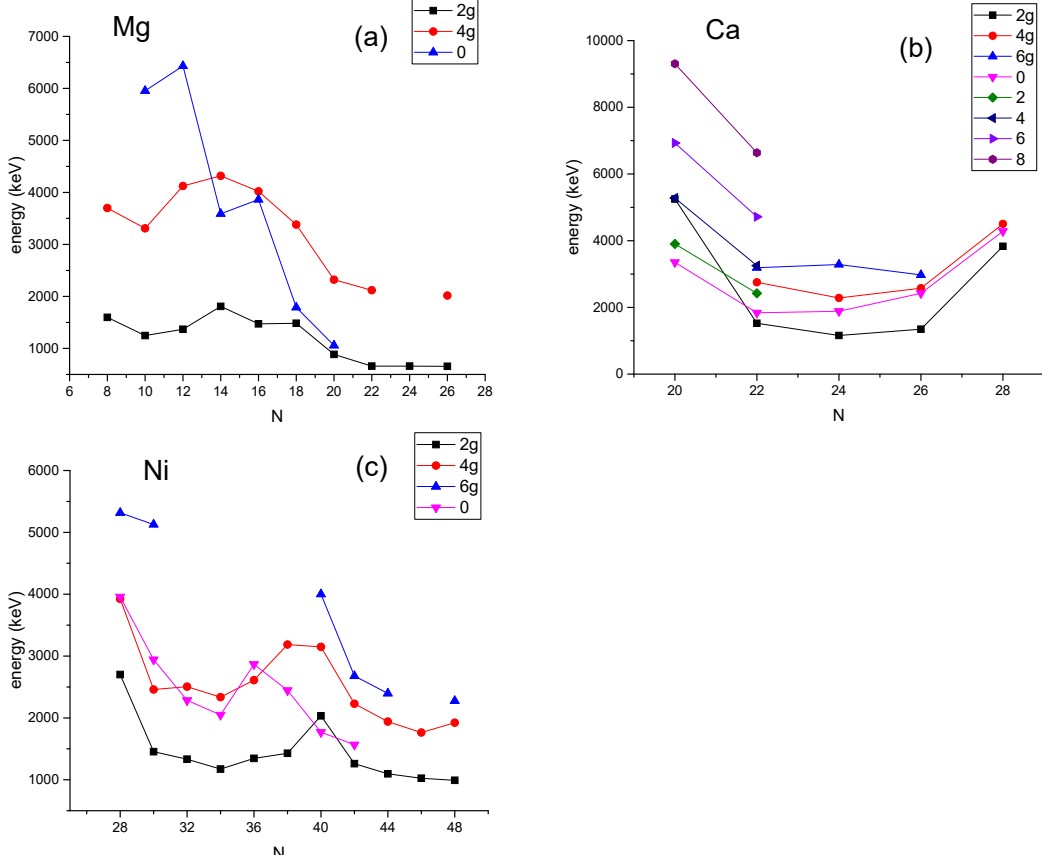

**Figure 13.** Experimental energy levels in keV [406] for the Mg ($Z = 12$) (**a**), Ca ($Z = 20$) (**b**), and Ni ($Z = 28$) (**c**) series of isotopes. The levels of the ground state band bear the subscript g, while the levels of the excited $K^\pi = 0^+$ band bear no subscript. See Sections 10.1, 10.5 and 10.9 and for further discussion.

$^{68}$Ni$_{40}$ has been considered as a test ground for examining the roles played by the various components of the nucleon–nucleon interaction in the shell-model framework in the appearance of SC. The balance of the tensor and the central force has been considered in the MCSM framework [105], while the competition between the spherical mean field and the pairing and quadrupole–quadrupole interactions has been studied in large-scale shell-model calculations taking advantage of the pseudo-SU(3) and quasi-SU(3) symmetries [46]. Calculations within these two frameworks extending to several Ni isotopes can be found in [58,73] and [47], respectively.

SC in $^{78}$Ni$_{50}$ has been shown to occur within large-scale shell-model calculations taking advantage of the pseudo-SU(3) and quasi-SU(3) symmetries [534], signifying a fifth island of inversion at $N = 50$, in addition to the ones known at $N = 8, 20, 28, 40$. Earlier considerations on the evolution of the $N = 50$ gap can be found in [757].

SC in the doubly magic $^{56}$Ni$_{28}$ has been studied in the MCSM framework [56,58,67,73], as well as with a pairing plus multipole Hamiltonian with a monopole interaction [63] and within the complex excited VAMPIR mean-field approach [187].

Extended MCSM calculations incorporating SC at the doubly magic nuclei $^{56,68,78}$Ni$_{28,40,50}$ can be found in [58,73].

In summary, much experimental and theoretical evidence for SC is accumulated in the region $^{64-70}$Ni$_{36-42}$, i.e., around $N = 40$, while some experimental and theoretical evidence for SC exists in $^{78}$Ni$_{50}$, signifying an island of inversion at $N = 50$. Theoretical predictions for SC in the $N = Z$ nucleus $^{56}$Ni$_{28}$ also exist.

### 10.2. The Fe (Z = 26) Isotopes

Experimental evidence for particle–hole excitations across the $N = 40$ subshell closure and enhanced collectivity, related to the $N = 40$ island of inversion, have been observed in $^{62}Fe_{36}$ [548,758–760], $^{64}Fe_{38}$ [548,758–760], $^{66}Fe_{40}$ [546,759–762], $^{68}Fe_{42}$ [546,762]. Strong experimental evidence for particle–hole cross-shell excitations has been seen [763] in the $N = Z$ isotope $^{52}Fe_{26}$.

Large-scale shell-model [47,50] as well as two-band mixing [764] calculations in the $^{60–66}Fe_{34–40}$ region have indicated the existence of strongly mixed spherical and deformed configurations, as well as the occupation of neutron intruder orbitals as $N = 40$ is approached, related to the $N = 40$ island of inversion. Shell-model calculations show evidence for particle–hole excitations and SC in $^{54}Fe_{28}$ [68] and $^{76}Fe_{50}$ [534], in relation to the $N = 28$ and $N = 50$ islands of inversion, while the excited $0^+$ states in the $N = Z$ isotope $^{52}Fe_{26}$ have been considered in [62].

In summary, experimental and theoretical evidence for SC is accumulated for $^{62–66}Fe_{36–40}$, especially for $^{66}Fe_{40}$, while theoretical evidence also exists for $^{54}Fe_{28}$ and $^{76}Fe_{50}$, in relation to the $N = 28$ and $N = 50$ islands of inversion, as well as for the $N = Z$ isotope $^{52}Fe_{26}$.

### 10.3. The Cr (Z = 24) Isotopes

Proton inelastic scattering experiments in inverse kinematics on $^{58}Ti_{36}$, $^{60}Cr_{36}$ and $^{62}Cr_{38}$, backed by shell-model calculations, have demonstrated [384] that the contributions of both protons and neutrons are crucial for the building up of collectivity through the admixture of the $pf$ and $dg$ shells across the $N = 40$ gap. $^{58}Cr_{34}$ has been suggested as a candidate for the E(5) CPS in several works [384–387]. SC in $^{64,66}Mn_{39,41}$ has been found through the population of levels following the $\beta$ decay of $^{64}Cr_{40}$ and $^{66}Cr_{42}$ [765]. These findings cover the whole region $N = 34$–40, in which SC is expected to occur according to the dual-shell mechanism of the proxy-SU(3) symmetry [334]. Furthermore, the appearance of the CPS E(5) has been suggested to occur at the left border of this region, while the $N = 40$ island of inversion appears at its right border [536,546].

Furthermore, experimental evidence for SC has been seen in $^{52}Cr_{28}$ [766], related to the $N = 28$ island of inversion, as well as in the $N = Z$ isotope $^{48}Cr_{24}$ [763].

Shell-model calculations focused on the IoI around $^{64}Cr_{40}$ have been performed in [46,50]. Extended large-scale shell-model [47] and MCSM [58] calculations covering the whole $^{48–64}Cr_{24–40}$ region, as well as calculations using a 5DQCH [204,205] covering the $^{58–68}Cr_{34–44}$ region, corroborate the evolution of shape changes and the occurrence of shape mixing in the $^{58–64}Cr_{34–40}$ region, in agreement with a two-band mixing calculation [764] in this region.

The relation between SC and isobaric analog states has been considered in $^{44}Cr_{20}$ within an algebraic framework in [767]. The $N = Z$ isotope $^{48}Cr_{24}$ has been considered by shell-model calculations in [64], while $^{74}Cr_{50}$ has been considered by large-scale shell-model calculations in relation to the $N = 50$ IoI in [534].

In summary, much experimental and theoretical evidence for SC is accumulated for $^{58–64}Cr_{34–40}$, while experimental and/or theoretical evidence also exists for $^{44}Cr_{20}$, $^{52}Cr_{28}$, and $^{74}Cr_{50}$, in relation to the $N = 20, 28, 50$ islands of inversion, as well as for the $N = Z$ isotope $^{48}Cr_{24}$.

### 10.4. The Ti (Z = 22) Isotopes

Experimental evidence [763,768–770] for SC and particle–hole excitations has been found in $^{44}Ti_{22}$, corroborated by $df$ shell-model [768] and $pf$ shell-model [763,771] calculations, while theoretical considerations [767] in an algebraic framework have revealed the relation between isobaric analog states and SC.

The lowest four $0^+$ states of $^{50}Ti_{28}$ have been considered [197] within the excited VAMPIR framework, with considerable mixing among them found. Shell-model calculations [50,546] have considered $^{60,62}Ti_{38,40}$ as members of the $N = 40$ island of inversion, while experiments on $^{58}Ti_{36}$, $^{60}Cr_{36}$ and $^{62}Cr_{38}$ have demonstrated [384] that the contributions of both

protons and neutrons are crucial for the building up of collectivity through neutron excitations across the $N = 40$ gap. Large-scale shell-model calculations [534] have been performed for $^{72}$Ti$_{50}$ in relation to the $N = 50$ island of inversion.

In summary, much experimental and theoretical evidence for SC is accumulated for $^{44}$Ti$_{22}$, i.e., at $N = Z$, while theoretical studies also exist for $^{50}$Ti$_{28}$, $^{60,62}$Ti$_{38,40}$, and $^{72}$Ti$_{50}$, in relation to the $N = 28, 40, 50$ islands of inversion.

### 10.5. The Ca (Z = 20) Isotopes

SC in $^{40}$Ca$_{20}$ [772] and $^{42}$Ca$_{22}$ [773] have been known experimentally since 1972, and have been interpreted as multi-paricle–multi-hole excitations in the framework of large-scale shell-model calculations [46,48]. Recent experimental evidence [769,774,775] corroborated SC in $^{42}$Ca$_{22}$ and extended it to $^{44}$Ca$_{24}$, the latter having been theoretically considered [767] in an algebraic framework revealing the relation between isobaric analog states and SC.

It should be noticed that $^{40}$Ca$_{20}$ represents a rare example in which the $0_2^+$ band-head of a well-developed $K^\pi = 0^+$ band lies below the $2_1^+$ state of the ground state band [335] (see Figure 13b). It has been found [335] that such nuclei tend to be surrounded by nuclei exhibiting SC. Indeed the intruder states in Figure 13b seem to form the bottom of a parabola around the $N = 24$ midshell.

Theoretical studies in relation to SC and islands of inversion have been performed for $^{48}$Ca$_{28}$ using a 5DQCH with parameters determined from the relativistic energy density funcional DD-PC1 [240], as well as within TDHFB theory [180]. Large-scale shell-model calculations for $^{36}$Ca$_{16}$ [51], $^{60}$Ca$_{40}$ [50] and $^{70}$Ca$_{50}$, related to the IoIs at $N = 20$, $N = 40$ and $N = 50$, have also been carried out.

In summary, much experimental and theoretical evidence for SC is accumulated for $^{36,40-44}$Ca$_{16,20-24}$, i.e., around $N = 20$, while theoretical studies also exist for $^{48}$Ca$_{28}$, $^{60}$Ca$_{40}$, and $^{70}$Ca$_{50}$, in relation to the $N = 20, 28, 40, 50$ islands of inversion.

### 10.6. The Ar (Z = 18) Isotopes

Experimental evidence [776–780] for SC and particle–hole excitations has been found in $^{36-40}$Ar$_{18-22}$, corroborated by shell-model [778,780], cranked Nilsson–Strutinsky [780], and cranked HFB [777] calculations. SC in $^{36}$Ar$_{18}$ has also been considered [169] within RMF theory, in conjunction with the hypernucleus $^{37}_\Lambda$Ar.

Experimental evidence for the existence of a strong $N = 28$ shell gap at $^{46}$Ar$_{28}$ has been provided by mass measurements [781]. The importance of the quadrupole interaction between protons and neutrons in coupling the proton hole states with neutron excitations across the $N = 28$ shell gap in $^{46}$Ar$_{28}$ has been demonstrated through TDHFB calculations [180]. In addition, $^{46}$Ar$_{28}$ has been considered within a 5DQCH with parameters determined through the relativistic energy density functional DD-PC1 [233,240].

In summary, much experimental and theoretical evidence for SC is accumulated for $^{36-40}$Ar$_{18-22}$, i.e., around $N = 20$, while experimental evidence and theoretical studies also exist for $^{46}$Ar$_{28}$, in relation to the $N = 28$ island of inversion.

### 10.7. The S (Z = 16) Isotopes

Experimental evidence for SC has been seen in $^{44}$S$_{28}$ [754,782,783] and $^{43}$S$_{27}$ [784,785], i.e., close to the $N = 28$ shell closure.

SC in $^{44}$S$_{28}$ has been seen in theoretical calculations within self-consistent mean-field theory with Skyrme interaction [147], TDHFB theory [180], shell model with variation after angular-momentum projection [75], as well as within a 5DQCH with parameters determined from the relativistic energy density functional DD-PC1 [233,240] (see Figure 1 of [75], Figure 7 of [240], and Figure 3 of [233] for relevant level schemes).

Calculations covering the $^{36-44}$S$_{20-28}$ region have been performed in the self-consistent HF framework with Skyrme interaction [69], as well as within various shell-model approaches [47,69,74]. A deformed ground state is predicted [69] for $^{44}$S$_{28}$, with strong mixing

of the closed-shell and excited configurations [47]. The crucial role played by the tensor force in creating SC in $^{44}$S$_{28}$ is clearly seen in Figure 5 of [74], where it is made clear that the tensor force is crucial for developing a second minimum in the potential energy surface.

Shell-model calculations with configuration interaction [51] point out the different structures of the intruder and normal states in $^{36}$S$_{20}$, thus giving a signal related to the $N = 20$ island of inversion.

In summary, much experimental and theoretical evidence for SC is accumulated for $^{44}$S$_{28}$, i.e., at the $N = 28$ island of inversion, while some theoretical evidence also exists for $^{36}$S$_{20}$, in relation to the $N = 20$ island of inversion.

*10.8. The Si (Z = 14) Isotopes*

SC has been observed in $^{34}$Si$_{20}$ [754,786,787] and $^{32}$Si$_{18}$ [788]. Large-scale shell-model calculations have also been performed for these nuclei [46,47,49].

SC in $^{42}$Si$_{28}$ has been considered in shell-model calculations [49,74], as well as in a 5DQCH with parameters determined from RMF calculations using the DD-PC1 energy density functional [233,240] and in a TDHFB framework [180].

SC in the $N = Z$ nucleus $^{28}$Si$_{14}$ has been considered by the Nilsson–Strutinsky approach [789,790], as well as by constrained HFB plus local quasiparticle RPA calculations [201] and large-scale shell-model calculations [49].

In summary, much experimental and theoretical evidence for SC exists for $^{32,34}$Si$_{18,20}$, i.e., around $N = 20$, while theoretical evidence for SC also exists both for $^{42}$Si$_{28}$ and $^{28}$Si$_{14}$, i.e., at the $N = 28$ island of inversion and along the $N = Z$ line.

*10.9. The Mg (Z = 12) Isotopes*

SC has been observed in $^{30}$Mg$_{18}$ [754,791–793] and $^{32}$Mg$_{20}$ [543,754], as well as in $^{40}$Mg$_{28}$ [754]. SC is also seen in the $N = Z$ nucleus $^{24}$Mg$_{12}$ [794]. Intruder states shown in Figure 13a seem to form the left branch of a parabola reaching a minimum at $N = 20$.

Mass systematics [533] have suggested $^{32,34}$Mg$_{20,22}$ as candidates for the island of inversion at $N = 20$ (see also [542]). SC in $^{30-34}$Mg$_{18-22}$ has been considered in self-consistent HF calculations with a Skyrme interaction [69], as well as in large-scale shell-model calculations [46,49], and within a 5DQCH [198,202].

SC in the $N = 28$ isotones with $Z = 12–20$, including $^{40}$Mg$_{28}$, has been studied in a 5DQCH with parameters determined from RMF calculations using the DD-PC1 energy density functional [233,240]. A bridge between the $N = 20$ and $N = 28$ islands of inversion has been provided for the Mg isotopes by large-scale shell-model calculations [49], supported by recent experimental evidence [545].

SC in the $N = Z$ nucleus $^{24}$Mg$_{12}$ has been considered within the extended IBM formalism [427–429], as well as within shell-model [62] and constrained HFB plus local quasiparticle RPA calculations [201]. In contrast, hindrance of shape mixing in $^{26}$Mg$_{14}$ has been found in the last calculation [201], based on the softness of the collective potential against the $\beta$ and $\gamma$ deformations.

In summary, much experimental and theoretical evidence for SC is accumulated for $^{30,32}$Mg$_{18,20}$, i.e., around $N = 20$, while experimental and theoretical evidence for SC also exists both for $^{40}$Mg$_{28}$ and $^{24}$Mg$_{12}$, i.e., at the $N = 28$ island of inversion and along the $N = Z$ line.

*10.10. The Ne (Z = 10) Isotopes*

Mass systematics [533] have suggested $^{30,32}$Ne$_{20,22}$ as candidates for the island of inversion at $N = 20$ (see also [542]). Further experimental evidence supporting that $^{32}$Ne$_{22}$ does belong to the $N = 20$ IoI has been provided in [795], while experimental evidence on $^{28}$Ne$_{18}$ [796] suggests that this nucleus does not belong to the $N = 20$ IoI. Self-consistent HF calculations with a Skyrme interaction [69] suggested that shape mixing occurs in $^{30}$Ne$_{20}$, but to a much smaller degree in comparison to $^{32}$Mg$_{20}$ (compare Figures 2 and 3 of [69]). Large-scale shell-model calculations [49] corroborated that $^{30,32}$Ne$_{20,22}$ lie in the

$N = 20$ island of inversion, and suggested that the Ne isotopes might also provide a bridge between the $N = 20$ IoI and the $N = 28$ IoI, if the neutron drip line does reach $^{38}Ne_{28}$.

In contrast, hindrance of shape mixing in $^{24}Ne_{14}$ has been found in constrained HFB plus local quasiparticle RPA calculations [201], based on the softness of the collective potential against the $\beta$ and $\gamma$ deformations.

$^{20}Ne_{10}$ has been known [473] to be a textbook example of a deformed nucleus in the very light mass region, having a high-lying $0_2^+$ state [62]. Recent calculations within the symmetry-adapted no-core shell-model have revealed [346] that $^{20}Ne_{10}$ is made out of a few equilibrium shapes in the framework of the symplectic symmetry.

In summary, both experimental and theoretical evidence for SC is concentrated on $^{30,32}Ne_{20,22}$, lying within the $N = 20$ IoI.

### 10.11. The O (Z = 8) Isotopes

$^{16}O_8$ is the nucleus in which SC was first suggested by Morinaga in 1956 [1]. Soon thereafter, the two-particle–two-hole excitation mechanism was discussed in the shell-model framework for $^{16}O_8$ [797–799] and $^{18}O_{10}$ [798].

Proton ($\pi$) and neutron ($\nu$) particle–hole (p-h) excitations in $^{16}O_8$ have been considered in the extended IBM-2 model within the Wigner SU(4) symmetry [800–803], further discussed in Section 11. In particular, the $\pi(2p\text{-}2h)\nu(2p\text{-}2h)$, $\pi(4p\text{-}4h)$, and $\nu(4p\text{-}4h)$ configurations of $^{16}O_8$ have been placed in the same multiplet with the $\pi(4p)\nu(4p)$ configuration of $^{24}Mg_{12}$ [427–429].

The emergence of deformation and SC in $^{16}O_8$ has been discussed by Rowe et al. in the framework of the symplectic model [619,620,804].

The spectrum of $^{20}O_{12}$ has been used, in comparison to the spectrum of $^{20}Ne_{10}$, in order to demonstrate the importance of the proton–neutron interaction for the development of nuclear deformation in the latter [473].

Recent experimental evidence [539] suggests $^{28}O_{20}$ as the low-$Z$ shore of the $N = 20$ IoI.

In summary, SC appears in $^{16}O_8$ and $^{28}O_{20}$.

### 10.12. The C (Z = 6) Isotopes

The $0_2^+$ state of $^{12}C_6$ at 7.65 MeV is of special interest for stellar nucleosynthesis, since it plays a key role in allowing it to proceed beyond $^{12}C_6$ to heavier elements. First suggested by the astrophysicist Hoyle in 1954 [805], it was considered from the SC point of view in 1956 by Morinaga, his work being the first study of SC in atomic nuclei [1]. An important step towards establishing a band based on the Hoyle state was the observation of a $2^+$ state around 10 MeV [806,807]. The structure of $^{12}C_6$ has also been discussed by Rowe et al. [619,620] in the framework of the symplectic model. Extended work has been done over the years on the structure of the Hoyle state and its description as an alpha-particle condensate, recently reviewed in [808,809], respectively, while a relevant review on microscopic clustering in light nuclei has been given in [810]. The Hoyle state is still an active field of research (see, for example, [811] and references therein).

### 10.13. The Be (Z = 4) Isotopes

The breakdown of the $N = 8$ shell closure in $^{12}Be_8$ has been seen experimentally in 2000 in [812,813], see [814] for recent references. This leads to the lowest island of inversion at $N = 8$ [536].

## 11. The $N = Z$ Nuclei

Summarizing the evidence for SC in $N = Z$ nuclei cited in the subsections of Sections 3–10 on individual series of isotopes and mentioned in their conclusions, the following picture emerges.

SC is seen in the isolated nuclei $^{12}_6C_6$, $^{16}_8O_8$, $^{24}_{12}Mg_{12}$, $^{28}_{14}Si_{14}$, $^{44}_{22}Ti_{22}$, $^{48}_{24}Cr_{24}$, $^{52}_{26}Fe_{26}$, $^{56}_{28}Ni_{28}$.

In contrast, SC in $N = Z$ nuclei surrounded by more isotopes exhibiting SC is seen in the regions $^{36-40}_{18}\text{Ar}_{18-22}$, $^{40-44}_{20}\text{Ca}_{20-24}$, $^{68-74}_{34}\text{Se}_{34-40}$, $^{70-76}_{36}\text{Kr}_{34-40}$, $^{74-82}_{38}\text{Sr}_{36-44}$, $^{76-84}_{40}\text{Zr}_{36-44}$.

The difference between these two groups of $N = Z$ nuclei can be explained through the dual-shell mechanism [334,335] developed in the framework of the proxy-SU(3) symmetry [323,327]. The $Z$ number of the nuclei belonging to the second group lies within the regions 7–8, 17–20, 34–40, 59–70, 96–112..., in which SC can appear according to the dual-shell mechanism; thus, the $N = Z$ nucleus can be accompanied by additional isotopes with $N$ lying within these regions. For example, $^{68}_{34}\text{Se}_{34}$ can be accompanied by its isotopes lying within the $N = 34$–40 region, i.e., the whole group $^{68-74}_{34}\text{Se}_{34-40}$ can exhibit SC.

$N = Z$ nuclei are the ideal test ground for the manifestation of Wigner's spin-isospin symmetry, introduced in 1937 [803] and called the Wigner SU(4) supermultiplet model [800–802]. Wigner suggested that, at first approximation, the nuclear forces should be independent of the orientation of the spins and the isospins of the nuclei constituting the nucleus. The double binding energy differences of Equation (3) have provided a test [815] of the Wigner SU(4) supermultiplet, while further details on its implications for heavy $N = Z$ nuclei can be found in chapter 6 of [816].

*11.1. $T = 0$ Pairs in $N = Z$ Nuclei*

The fact that neutrons and protons in $N = Z$ nuclei occupy the same orbitals gives room to the development of unique proton–neutron correlations which cannot be seen away from the $N = Z$ line. Much attention has been attracted by the experimental observation in 2011 [817] of an isoscalar ($T = 0$) spin-aligned ($S = 1$) neutron-proton phase in the spectrum of $^{92}_{46}\text{Pd}_{46}$. This phase is expected to replace the standard seniority [37,258–261] coupling in the ground and low-lying excited states of the heaviest $N = Z$ nuclei. The existence of this phase in $^{92}_{46}\text{Pd}_{46}$ and $^{96}_{48}\text{Cd}_{48}$ has been corroborated by shell-model calculations [818–822], within a model using quartets [823,824], as well as by mapping the shell-model wave functions onto bosons [825], see also [816,826] for relevant reviews. This phase has been recently seen also in $^{88}_{44}\text{Ru}_{44}$ [827].

The quartet formalism has also been successfully applied in the heavier $N = Z$ nuclei $^{104}_{52}\text{Te}_{52}$, $^{108}_{54}\text{Xe}_{54}$, and $^{112}_{56}\text{Ba}_{56}$ [828–830], as well as in the light $N = Z$ nuclei $^{20}_{10}\text{Ne}_{10}$, $^{24}_{12}\text{Mg}_{12}$, $^{28}_{14}\text{Si}_{14}$, and $^{32}_{16}\text{S}_{16}$ [823,828–830]. Shape isomers and clusterization, taking advantage of a quasidynamical SU(3) symmetry, have been studied in $^{28}_{14}\text{Si}_{14}$ [831] and $^{36}_{18}\text{Ar}_{18}$ [832,833].

The quartet formalism has also been successfully applied in the $N = Z$ nuclei $^{44}_{22}\text{Ti}_{22}$, $^{48}_{24}\text{Cr}_{24}$, and $^{52}_{26}\text{Fe}_{26}$ [828–830]. These nuclei have also been considered within cranked mean-field [834,835] and shell-model [821,836,837] calculations.

Experimental investigations in the $N = Z$ nuclei $^{56}_{28}\text{Ni}_{28}$, $^{60}_{30}\text{Zn}_{30}$, and $^{64}_{32}\text{Ge}_{32}$ [838–841] have been followed by mean-field [842–844] and shell-model [845] calculations focusing on the $T = 0$ neutron–proton pairing correlations and their competition with the $T = 1$ interactions, as well as by clusterization [846] investigations. The importance of the $T = 0$ channel has been questioned by [847]. A review of these efforts can be found in [848].

Experimental investigations at higher masses have been carried out in $^{68}_{34}\text{Se}_{34}$ [849], $^{72}_{36}\text{Kr}_{36}$ [838], and $^{76}_{38}\text{Sr}_{38}$ [850]. An overview of the $N = Z = 32$–46 region has been given in Section 11.3 of [816], while for the $N = Z = 44$–48 region one can see [826].

In summary, while SC in the $N = Z$ nuclei has been seen up to $N = Z = 42$, theoretical investigations are extended up to $N = Z = 56$, recently motivated by the observation of the isoscalar ($T = 0$) spin-aligned ($S = 1$) neutron–proton phase in the region of $^{92}_{46}\text{Pd}_{46}$. No SC is expected in the $N = Z = 44$–56 region according to the dual-shell mechanism [334,335] developed in the framework of the proxy-SU(3) symmetry [323,327], since these nuclei lie outside the 7–8, 17–20, 34–40, 59–70, 96–112...regions, in which SC can appear.

## 12. Islands of Inversion

Summarizing the evidence for IoIs cited in the subsections of Sections 3–10 on individual series of isotopes and mentioned in their conclusions, the following picture emerges.

Signs for the IoI at $N = 8$ appear at $Z = 4$, in $^{12}_{4}\text{Be}_{8}$.

Evidence for the IoI at $N = 20$ appears at $Z = 8$–20, 24, namely in $^{28}_{8}O_{20}$, $^{30,32}_{10}Ne_{20,22}$, $^{30,32}_{12}Mg_{18,20}$, $^{32,34}_{14}Ne_{18,20}$, $^{36}_{16}S_{20}$, $^{36-40}_{18}Ar_{18-22}$, $^{40-44}_{20}Ca_{20-24}$, and $^{44}_{24}Cr_{20}$. In other words, a robust $N = 20$ IoI appears at $Z = 8$-20, with an additional point at $Z = 24$. In some cases ($Z = 12, 14, 18$), SC is also seen at $N = 18$, while in other cases ($Z = 10, 18, 20$) it is also seen at $N = 22$.

Evidence for the IoI at $N = 28$ appears at $Z = 12$–28, namely in $^{40}_{12}Mg_{28}$, $^{42}_{14}Si_{28}$, $^{44}_{16}S_{28}$, $^{46}_{18}Ar_{28}$, $^{48}_{20}Ca_{28}$, $^{50}_{22}Ti_{28}$, $^{52}_{24}Cr_{28}$, $^{54}_{26}Fe_{28}$, and $^{56}_{28}Ni_{28}$. In other words, a robust $N = 28$ IoI appears at $Z = 12$–28, with no additional cases seen at $N = 26$ or $N = 30$.

Evidence for the IoI at $N = 40$ appears at $Z = 20$–40, namely in $^{60}_{20}Ca_{40}$, $^{60,62}_{22}Ti_{38,40}$, $^{58-64}_{24}Cr_{34-40}$, $^{62-66}_{26}Fe_{36-40}$, $^{64-70}_{28}Ni_{36-42}$, $^{68-70}_{30}Zn_{38,40}$, $^{68-76}_{32}Ge_{36-44}$, $^{68-72}_{34}Se_{34-40}$, $^{70-76}_{36}Kr_{34-40}$, $^{74-82}_{38}Sr_{36-44}$, and $^{76-84}_{40}Zr_{36-44}$. In other words, a robust $N = 40$ IoI appears at $Z = 20$–40, which in most cases ($Z = 22$–40) is extended to lower $N$ as well.

Evidence for the IoI at $N = 50$ appears at $Z = 20$–32, namely in $^{70}_{20}Ca_{50}$, $^{72}_{22}Ti_{50}$, $^{74}_{24}Cr_{50}$, $^{76}_{26}Fe_{50}$, $^{78}_{28}Ni_{50}$, $^{78,80}_{30}Zn_{48,50}$, $^{80-94}_{32}Ge_{48-62}$. In addition, signs of SC appear further up in $Z = 34$, 36 in the $N = 52$ isotones $^{86}_{34}Se_{52}$, and $^{88}_{36}Kr_{52}$. In other words, a robust $N = 50$ IoI appears at $Z = 30$-32, with no additional cases seen up to $Z = 28$, while above $Z = 28$ SC starts to be seen in $N = 48$ or $N = 52$ up to $Z = 36$.

It is seen that the $N = 28$ and $N = 50$ IoIs remain "narrow", while the $N = 20$ and $N = 40$ IoIs get "wide". A way to understand this difference is again offered by the dual-shell mechanism [334,335] developed in the framework of the proxy-SU(3) symmetry [323,327]. According to the dual-shell mechanism, SC can appear within the neutron intervals 7–8, 17–20, 34–40, 59–70, 96–112 .... $N = 28$ and $N = 50$ lie outside these intervals, therefore signs of inversion of single particle orbitals and/or signs of SC can appear only at the shell-model magic numbers 28 and 50, presumably due to particle–hole excitations across these gaps. On the contrary, $N = 20$ and $N = 40$ are the upper borders of two of these intervals; thus, SC can be expected also at lower $N$ below them. It should be noticed that "narrow" IoIs correspond to shell-model magic numbers, while "wide" IoIs are related to isotropic 3D-HO magic numbers.

As mentioned in Section 2.10, islands of inversion [534] are formed by nuclei appearing close to semimagic or even doubly magic nuclei, expected to be spherical in their ground state, but found to be deformed. How does this happen? The proton–neutron interaction produces a shape transition, in which highly correlated many-particles–many-holes configurations, called intruders, become more bound than the spherical ones [534]. This mechanism is the same as the one widely accepted to be the microscopic cause of SC. Therefore, SC and SPT from spherical to deformed nuclei are one and the same effect, caused by the proton–neutron interaction. This is why the dual-shell mechanism, based on the proxy-SU(3) symmetry, works well in predicting the regions in which SC can appear. The present mechanism is also consistent with the argument that proton p-h excitations are caused by the neutrons and vice versa, in agreement to the role played in the shell-model framework by the tensor force.

Large-scale shell-model calculations [49] have suggested the merging of the $N = 20$ and $N = 28$ IoIs. Indeed the intruder states shown in Figure 13a reveal such a tendency, since the $0^+_2$ states keep falling from $N = 12$ to $N = 20$. In order to clarify this issue, the $0^+_2$ states in $^{34-38}Mg_{22-26}$ should be measured. Experimental findings [545] for the $R_{4/2}$ ratios within the ground state bands of these nuclei seem to support the idea of merging.

A similar picture appears in the case of the intruder states shown in Figure 13c, in which the $0^+_2$ states keep falling from $N = 36$ to $N = 42$. In order to clarify the question of merging of the $N = 40$ and $N = 50$ [534] IoIs, the $0^+_2$ states in $^{72-76}Ni_{44-48}$ should be measured.

## 13. Unified Perspectives for Shape Coexistence

### 13.1. Islands of Shape Coexistence and Shape/Phase Transitions

The conclusions of all subsections of Sections 3–10 are depicted in Figure 1, in which the stripes among the nucleon numbers 7–8, 17–20, 34–40, 59–70, 96–112, 146–168, pre-

dicted by the dual-shell mechanism [334] developed in the framework of the proxy-SU(3) symmetry [323,327] are also shown in azure. According to the dual-shell mechanism [334], SC can occur only within these stripes. We remark that all nuclei in which SC has been found fall within these stripes, with two notable exceptions. The first exception regards some $N = Z$ nuclei, in which additional components of the nucleon–nucleon interaction are activated, as discussed in Sections 11 and 11.1. The second exception consists of some nuclei within the $N = 28$ and $N = 50$ islands of inversion, discussed in Section 12. Notice that these are the IoIs which correspond to shell-model magic numbers created by the spin–orbit interaction, while the IoIs at $N = 8$, $N = 20$, and $N = 40$, which correspond to magic numbers of the isotropic 3D-HO, fall within the relevant stripes of the dual-shell mechanism. A few other nuclei appear just outside the borders of the stripes, simply emphasizing the approximate nature of the arguments which led to their introduction (see Section 8 of [334] for relevant details).

The prediction of the proton and the neutron driplines has been recently receiving attention within several theoretical approaches, including large-scale shell-model [851], energy density functional theory [852], and *ab initio* calculations starting from chiral two- and three-nucleon interactions and taking advantage of the renormalization group [853]. The proton (neutron) driplines shown in Figure 1 and subsequent figures have been extracted from the two-proton (two-neutron) separation energies appearing in the database NuDat3.0 [349].

In relation to the microscopic mechanisms underlying the existence of these islands, Figure 14 can be helpful. According to the results [579,580] obtained through CDFT with standard amount of pairing included, the "horizontal" islands along $Z = 82$ and $Z = 50$ correspond to proton particle–hole excitations across these proton magic numbers, created by the influence of the neutrons. The term *neutron-induced shape coexistence* has been used for them [579,580]. In contrast, the "vertical" islands around $N = 90$ and $N = 60$ correspond to neutron particle–hole excitations from normal parity orbitals to intruder parity orbitals, created by the influence of the protons. The term *proton-induced shape coexistence* has been used for them [579,580]. In the island formed around $N = Z = 40$, both mechanisms are simultaneously present. As seen in Section 2.1, the influence of protons on the neutron gaps, as well as the influence of neutrons on the proton gaps, is the main effect caused by the tensor force [102,105–107] in the shell-model framework. Furthermore, it has been recently realized [445] within the IBM framework with configuration mixing that SC in the $N = 60$ region is due to the crossing of regular and intruder configurations.

While the nature of SC along the $Z = 82$ and $Z = 50$ lines has been understood since the early days of SC [2,3,83–87] as due to proton particle–hole excitations across these proton gaps, the nature of SC in the $N = 90$ and $N = 60$ regions has been mentioned as an open question (see Sections III.A.3 and III.A.4 of [4]) until very recently (see Section 4 of [8]). It seems that the recent findings clarify the nature of SC in the $N = 90$ and $N = 60$ regions in three different frameworks, namely the shell model [102,105–107], the CDFT with pairing interaction included [579,580], and the IBM-CM [445].

Within the shell-model [102,105–107] framework, the occurrence of SC is explained in terms of the action of the tensor force, which reduces the widths of certain energy gaps. Within the CDFT framework with pairing interaction included [579,580], SC is explained in terms of particle–hole excitations among shells of different parity. Within the IBM-CM [445] framework, SC is explained in terms of mixing of a configuration with particle–hole excitations with the ground state configuration. The interconnections between these three approaches is evident, reinforcing their validity. The reduced gaps predicted by the tensor force in the shell-model framework facilitate the occurrence of particle–hole excitations in the CDFT approach, which is equivalent to the appearance of a configuration with increased number of bosons in the IBM description.

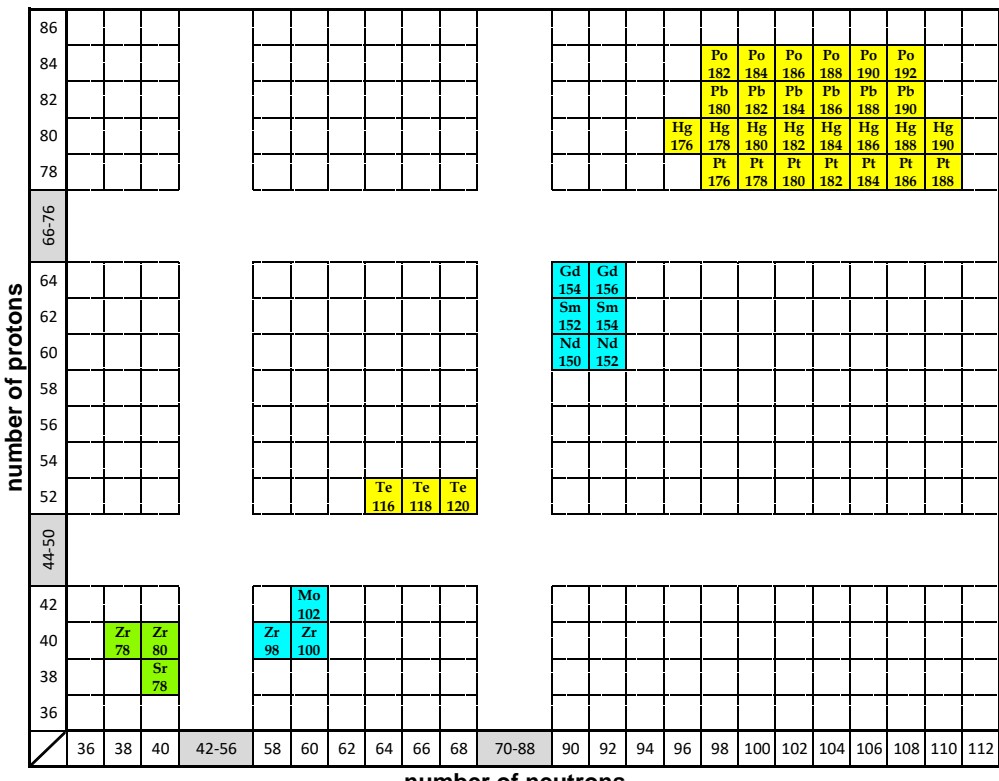

**Figure 14.** Nuclei found to exhibit particle–hole excitations, and therefore showing SC, in CDFT calculations with standard pairing included [579,580]. Islands of SC corresponding to neutron-induced SC are shown in yellow, islands due to proton-induced SC are exhibited in cyan, while islands due to both mechanisms are shown in green. Adapted from [580]. See Section 13.1 for further discussion.

The close connection between SPTs and SC, first suggested by Heyde et al. in 2004 [469], has recently been clarified through detailed studies in the Zr ($Z = 40$) [438,439,683] and Sr ($Z = 38$) [445] series of isotopes around $N = 60$, in the framework of the IBM-CM. It turns out that the $Z \approx 40$, $N \approx 60$ region offers a clear example of an SPT from spherical to prolate deformed shapes, similar to the SPT taking place in the $Z \approx 64$, $N \approx 90$ region, which is known to represent the best manifestation of the first order SPT from spherical to prolate deformed shapes in the IBM [411], described in the framework of the Bohr Hamiltonian by the X(5)CPS [451].

The similarity between these two regions is further emphasized by Figure 2, in which all nuclei reported in the literature as candidates for the X(5) CPS are shown, along with the $P \sim 5$ contours (see Equation (1)) indicating the borders of regions of deformed nuclei and therefore the regions of the onset of deformation. We see that all X(5) candidates in the $Z \approx 40$, $N \approx 60$ and $Z \approx 64$, $N \approx 90$ regions lie on the $P \sim 5$ contours or they are touching them. The same happens in the $Z \approx 40$, $N \approx 40$ region, as well as for all other X(5) candidates reported in the literature. For the sake of comparison, all nuclei reported in the literature as candidates for the E(5) CPS are also shown.

It should be emphasized that the islands of SC appearing in Figure 1 are based on the mechanism of particle–hole excitations, which can also be described as a change of the order of single-particle orbitals in the case of the islands of inversion. It might be possible that some other mechanism might exist, leading to SC far from the shores of these islands, but up to now signs of SC far from these islands is practically missing.

### 13.2. Systematics of Data

In order to observe SC, a $K^{\pi} = 0^{+}$ band lying close to the ground state band is needed. In Figure 3, the nuclei in which well-developed $K^{\pi} = 0^{+}$ bands are experimentally known are shown. It is seen that they form islands lying within the stripes predicted by the dual-shell mechanism [334] developed in the framework of the proxy-SU(3) symmetry [323,327].

For SC to occur, the $K^{\pi} = 0^{+}$ band should not be too far away from the ground state band (gsb). In Figure 4, the nuclei in which the energy difference $E(0_2^{+}) - E(2_1^{+})$ is less that 800 keV [406] are shown. Again they are seen to cluster together into islands, lying within the stripes predicted by the dual-shell mechanism [334] developed in the framework of the proxy-SU(3) symmetry [323,327].

In Ref. [407] it has been shown that the proximity of the $K^{\pi} = 0^{+}$ band to the gsb can be expressed quantitatively through the energy ratio

$$R_{0/2} = \frac{E(0_2^{+})}{E(2_1^{+})}, \tag{4}$$

by requiring $R_{0/2} < 5.7$, which is the parameter-independent value predicted by the X(5) CPS [451]. In parallel, it has been found [407] that for SC to occur, the energy ratio $R_{4/2} = E(4_1^{+})/E(2_1^{+})$ should be less than 3.05, which guarantees that the nucleus stays below the well-deformed region (notice that $R_{4/2} = 2.9$ is the parameter-independent value predicted by the X(5) CPS) [451]). In other words, these two conditions guarantee that the nucleus is not getting well deformed, since in that case the $K^{\pi} = 0^{+}$ band goes to very high-energy values.

In addition, for SC to occur, the $K^{\pi} = 0^{+}$ band should be strongly connected to the ground-state band. A measure of this connection is the ratio

$$B_{02} = \frac{B(E2; 0_2^{+} \rightarrow 2_1^{+})}{B(E2; 2_1^{+} \rightarrow 0_1^{+})}, \tag{5}$$

connecting the band-head of the excited $K^{\pi} = 0^{+}$ band to the first excited state of the ground state band. In Ref. [407] it has been found that for SC to occur one should have $B_{02} > 0.1$ .

In Figure 5, all nuclei with experimentally known $B_{02} > 0.1$, $R_{0/2} < 5.7$, and $R_{4/2} < 3.05$, known to exhibit SC [407], are shown. Again they are seen to cluster together into islands, lying within the stripes predicted by the dual-shell mechanism [334] developed in the framework of the proxy-SU(3) symmetry [323,327]. The only notable exception is seen in the Pt region ($^{198}Pt_{120}$), in which it is known that the level schemes can admit an alternative description not involving particle–hole excitations [614].

The $B(E0; 0_2^{+} \rightarrow 0_1^{+})$ transition rate is expected [491,527] to be a very reliable indicator of SC, since strong $B(E0)$s would indicate bands with very different structures lying close in energy. However, the existing data [406,526,527] do not suffice for making meaningful systematics.

A dimensionless ratio of B(E2)s involving the $0_1^{+}$, $2_1^{+}$, as well as the $2_2^{+}$ or $2_3^{+}$ states has been found [854] to obtain values lying within two narrow bands in the case of nuclei exhibiting SC.

As shown in the recent (2022) review article by Garrett et al. [8], several additional experimental quantities (transfer cross sections, decay properties, charge radii) can serve as indicators of the presence of SC. It would be interesting to gather existing data for each of these quantities on the nuclear chart and see to which extend they are consistent with the predictions of the dual-shell mechanism. The impression formed from Figures 7, 13, 14, 15, 21, 33, and 52 of Ref. [8], in which the information from many different experimental quantities is simultaneously shown in various regions of the nuclear chart, is that these extra quantities also form islands consistent with the predictions of the dual-shell mechanism,

but a detailed study of each quantity separately over the whole nuclear chart is necessary before reaching any final conclusions.

*13.3. Comparison to Earlier Compilations*

In Figure 8 of the authoritative review article by Heyde and Wood [4], the main regions of SC are shown. In Figure 6 a comparison between the regions of SC shown in that figure and the present findings is attempted.

The regions L and J of Figure 8 of [4] correspond to $Z \sim 82$ and $Z \sim 50$ respectively. They are elongated along the $Z = 82$ and $Z = 50$ lines and fall almost entirely within the $N = 96$–112 and $N = 59$–70 stripes predicted by the dual-shell mechanism [334] developed in the framework of the proxy-SU(3) symmetry [323,327]. Furthermore, they agree with the findings of the present work, depicted in Figure 1 of the present work, in which it is seen that these islands are actually wider in the $Z$ direction in comparison to Figure 8 of [4]. Their microscopic origin is attributed [579,580] to proton particle–hole excitations across the $Z = 82$ and $Z = 50$ energy gaps, corresponding to neutron-induced SC, in accordance to the early suggestions of Heyde et al. [2,3,83–87].

The regions K and I of Figure 8 of [4] correspond to $Z \sim 64$, $N \sim 90$ and $Z \sim 40$, $N \sim 60$ respectively. The first one falls entirely within the $Z = 59$–70 stripe of the dual-shell mechanism, while the second one falls partly within the $Z = 34$–40 stripe and partly within the $N = 59$–70 stripe of the dual-shell mechanism. Their microscopic origin is attributed [579,580] to neutron particle–hole excitations from normal parity orbitals to intruder orbitals, corresponding to proton-induced SC, thus answering the long-standing question [4,8] about their nature. The close relation of SC in these two very similar in nature regions to the shape/phase transition from spherical to deformed shapes has also been understood, as mentioned in Section 13.1. The regions K and I agree with the findings of the present work, depicted in Figure 1 of the present work, except for the upper left part of region I, for which minimal evidence has been found in the present work.

It is interesting that in Figure 6 the analogs of regions L and K remain disconnected, while the analogs of regions J and I almost join within the $N = 59$–70 stripe of the dual-shell mechanism, practically separated only by the Ru ($Z = 44$) isotopes.

The region H of Figure 8 of [4] corresponds to $Z \sim 34$, $N \sim 40$. It falls partly within the $Z = 34$–40 stripe and partly within the $N = 34$–40 stripe of the dual-shell mechanism, in agreement with the findings of the present work, depicted in Figure 1 of the present work, except for the lower right part of region H, for which minimal evidence has been found in the present work. The latter fact reinforces the interpretation of this region as due to the simultaneous presence of neutron-induced particle–hole excitations of protons and proton-induced particle–hole excitations of neutrons [579,580].

Region G of Figure 8 of [4] corresponds to superdeformed nuclei, which have not been considered in the present work, in view of the limitations set in Section 1.1.

Region A of Figure 8 of [4] corresponds to $N = Z$ nuclei, discussed in Sections 11 and 11.1. Region A extends up to $N = Z = 28$, while in the present work it goes further up to $N = Z = 42$, as seen in Figure 1 of the present work.

Regions B and C of Figure 8 of [4] correspond to the $N = 8$ and $N = 20$ islands of inversion, in agreement with what is seen in Figure 1 of the present work, in which the $N = 20$ IoI joins the $N = Z$ line and probably starts to be extended above it.

Regions D and F of Figure 8 of [4] correspond to the $N = 28$ island of inversion, in agreement to what is seen in Figure 1 of the present work, in which these two regions are merged and the $N = 28$ IoI reaches the $N = Z$ line.

Finally, the lower part of region E of Figure 8 of [4] falls within the $Z = 17$–20 stripe of the dual-shell mechanism, while the upper part reaches the $N = Z$ line, in agreement with what is seen in Figure 1 of the present work.

What is missing in Figure 8 of [4], in comparison to Figure 1 of the present work, is the $N = 50$ island of inversion, which had not been discovered yet at that time [534].

In conclusion, very close agreement is seen between the pioneering, early Figure 8 of [4] and Figure 1 of the present work, the main differences being the discovery of new examples of SC over the last 12 years, as well as the improved theoretical understanding of the nature of SC in the $Z \sim 64$, $N \sim 90$ and $Z \sim 40$, $N \sim 60$ regions, connected to the appearance of shape/phase transitions [438,439,445,683]. These regions have been found to correspond to neutron-induced particle–hole excitations caused by the protons [579,580], in accordance with the mechanism supported by the tensor force [102,105–107] of the nucleon–nucleon interaction.

It should be noticed that global calculations for low-lying levels and B(E2)s among them have been carried out [236] in the framework of CDFT. In addition to the regions of SC and the IoIs discussed above, evidence for SC is seen in the regions $Z \sim 64$, $N \sim 76$ and $Z \sim 40$, $N \sim 70$. Both regions lie within the stripes predicted by the dual-shell mechanism [334] developed in the framework of the proxy-SU(3) symmetry [323,327]. In addition, in Figure 1 of the present work, one sees that evidence corroborating these predictions already exists.

## 14. Conclusions

The last decade has seen a rapid growth in experimental searches for SC, assisted by the new possibilities provided by the modern radioactive ion beam facilities built worldwide. In parallel, our understanding of its microscopic origins has been greatly improved, both through large-scale shell-model and density functional theory calculations, as well as through the use of symmetries.

It is by now becoming evident in even–even nuclei that SC of the ground-state band with another $K^\pi = 0^+$ band, lying close in energy but possessing radically different structure, is not occurring all over the nuclear chart, but within certain regions of it, called islands of SC (Sections 13.1 and 13.3).

Within these islands, SC is created either by proton particle–hole excitations caused by the influence of neutrons (neutron-induced shape coexistence), or by neutron particle–hole excitations caused by the influence of protons (proton-induced shape coexistence), while in some regions both mechanisms can act simultaneously (Section 13.1). These mechanisms, corroborated by CDFT (Section 13.1), can be traced back to the properties of the tensor force within the nuclear shell model (Section 2.1.1).

In light- and medium-mass nuclei the shell-model theoretical approach (including symmetry-based no-core shell-model schemes) prevails, while in heavy nuclei the mean-field approaches, including both non-relativistic schemes and CDFT, as well as symmetry-based methods are used.

In light- and medium-mass nuclei, a rearrangement of nuclear orbitals, similar in nature to the particle–hole mechanism leading to SC, appears along the $N = 8, 20, 28, 40, 50$ lines of the nuclear chart, forming the $N = 8, 20, 28, 40, 50$ islands of inversion (Section 12). In addition, SC appears along the $N = Z$ line of the nuclear chart, in which the contribution of spin-aligned proton–neutron pairs becomes important (Sections 11 and 11.1).

In medium-mass and heavy nuclei, the connection between SC and critical point symmetries corresponding to a first-order shape/phase transition from spherical to deformed nuclear shapes has been clarified (Section 13.1).

Based on the systematics of data, as well as on symmetry considerations, quantitative rules have been developed, predicting regions in which SC can appear, as a possible guide for the experimental effort (Section 13.2).

## 15. Outlook

Further experimental results on SC can help in improving our understanding of the details of the nucleon–nucleon interaction, as well as of its modifications occurring far from stability.

The separate study of existing data [8] for individual structure indicators over the nuclear chart can provide further evidence for the regions in which SC can be expected (Section 13.2).

Growing evidence for multiple SC (Section 2.9) will provide a further test for the SC islands predicted by the dual-shell mechanism developed within the proxy-SU(3) symmetry. If the mechanism holds, multiple SC should be confined within the areas of these islands [335].

Superheavy nuclei present a special interest, since a rich variety of novel phenomena seems to be predicted there (Section 3.5).

Theoretical predictions have been found in some cases to be sensitive to the parameter sets used (Section 5.2). Parameter-independent symmetry predictions can be useful in some cases in resolving any disagreements. The development of methods allowing the estimation of the accuracy [59] of numerical theoretical predictions can also be helpful in the long run.

It would be interesting to examine if the particle–hole mechanism found in CDFT calculations using the DDME2 functional and including standard pairing [579,580] in the $N = 90$, $N = 60$ and $N = 40$ regions is corroborated by calculations in other relativistic or non-relativistic calculations using different forces and/or parametrizations.

While the CPS X(5) is well established for the description of the first-order shape/phase transition seen in the $N \approx 90$, $Z \approx 60$ region (Section 5.5) and the SC occurring in relation to it, it seems that a more flexible scheme is needed for the description of the very similar SPTs seen in the regions $N \approx 60$, $Z \approx 40$ (Section 8.5) and $N \approx 40$, $Z \approx 34$ (Section 9.5), and the SC occurring in relation to them.

The relation between the upper limits of SC and the prolate–oblate shape/phase transition should also be explored. The upper limit of the $N = 96$–112 region, in which SC can be expected according the dual-shell mechanism developed within the proxy-SU(3) symmetry in the heavy rare earths, touches $N = 114$, at which the shape/phase transition from prolate to oblate shapes is expected to occur in this region (see Table II of [324]). It should be noticed that the textbook example [214] of the O(6) dynamical symmetry of IBM, $^{196}$Pt$_{118}$, lies next to this point. The relation among the upper border of SC, the prolate-to-oblate shape/phase transition, and the O(6) dynamical symmetry of IBM needs to be clarified.

A similar situation occurs theoretically for the rare earths with neutrons in the 50–82 shell. The upper limit of the $N = 59$–70 region, in which SC can be expected according the dual-shell mechanism developed within the proxy-SU(3) symmetry in the heavy rare earths, touches $N = 72$, at which the shape/phase transition from prolate to oblate shapes is expected to occur in this region (see Table III of [324]).

The problem of clarifying the relation among SC, the E(5) CPS related to the second order shape/phase transition between spherical and $\gamma$-unstable shapes, and the dynamical symmetry O(6) of IBM has been described in detail in Section 6.2, to which the reader is referred.

Negative parity bands related to the octupole degree of freedom [855–857], not considered in the present review, might offer a fertile ground for the search for SC of positive- and negative-parity bands, since it is known [858,859] that octupole deformation occurs when a negative-parity band with levels characterized by $J^\pi = 1^-, 3^-, 5^-, 7^-, 9^-, \ldots$ joins the ground-state band, which has levels with $J^\pi = 0^+, 2^+, 4^+, 6^+, 8^+, \ldots$, thus forming a single band with $J^\pi = 0^+, 1^-, 2^+, 3^-, 4^+, 5^-, 6^+, 7^-, 8^+, \ldots$. The fact that the bands are lying close in energy and that a double-well potential [860] in the octupole deformation collective variable $\beta_3$ is needed in order to describe the octupole deformed shape may favor the appearance of SC. The regions to be investigated are the ones lying close to the octupole magic numbers 34, 56, 88, 134 (see Figure 1 of [857]), where the manifestation of octupole effects is expected. For some examples, see [29,30] for the $N \sim 88$ region, and [31] in the $Z \sim 56$ region.

**Author Contributions:** Conceptualization, D.B., A.M., S.K.P., T.J.M. and N.M.; methodology, D.B., A.M., S.K.P., T.J.M. and N.M.; writing–original draft preparation, D.B.; writing–review and editing, D.B., A.M., S.K.P., T.J.M. and N.M.; supervision, D.B. All authors have read and agreed to the published version of the manuscript.

**Funding:** This research was funded by the Bulgarian National Science Foundation (BNSF) under contract no. KP-06-N48/1.

**Data Availability Statement:** No new data have been reported in this review article.

**Conflicts of Interest:** The authors declare no conflict of interest.

**Abbreviations**

| | |
|---|---|
| 5DQCH | five-dimensional quadrupole collective Hamiltonian |
| CDFT | covariant density functional theory |
| CPS | critical point symmetry |
| GCM | generator coordinate method |
| HF | Hartree–Fock |
| HFB | Hartree–Fock–Bogoliubov |
| IBM | interacting boson model |
| IBM-CM | interacting boson model with configuration mixing |
| MCSM | Monte Carlo shell model |
| PDS | partial dynamical symmetry |
| PPQ | pairing plus quadrupole |
| RHB | relativistic Hartree–Bogoliubov |
| RMF | relativistic mean field |
| SANCSM | symmetry-adapted no-core shell model |
| SC | shape coexistence |
| SPT | shape/phase transition |
| TDHFB | time-dependent Hartree–Fock–Bogoliubov |
| VAMPIR | variation after mean-field projection in realistic model spaces |

**Appendix A. Theoretical Methods**

In this Appendix, a list of theoretical terms and models used in the text of this review are listed (in alphabetical order) first, along with the section in which their definition and/or brief description appears. Below this list, additional theoretical terms and models used in the text of this review without definition, mostly in Sections 3–10, are briefly discussed, again in alphabetical order.

*Appendix A.1. Theoretical Approaches*

| | |
|---|---|
| BCS approximation | Section 2.2 |
| beyond-mean-field approach | Section 2.2 |
| collective model of Bohr and Mottelson | Section 2.3 |
| covariant density functional theory (CDFT) | Section 3.1 |
| critical point symmetry (CPS) | Section 2.6 |
| density functional theory (DFT) | Section 2.2 |
| dual shell mechanism | Section 2.4.6 |
| E(5) CPS | Section 2.6 |
| Elliott SU(3) model | Section 2.4.1 |
| extended IBM | Section 2.5 |
| five-dimensional quadrupole collective Hamiltonian (5DQCH) | Section 2.2 |
| generator coordinate method (GCM) | Section 2.2 |
| Gogny interaction | Section 2.2 |
| Hartree–Fock (HF) | Section 2.2 |
| Hartree–Fock–Bogoliubov (HFB) | Section 2.2 |
| IBM with configuration mixing (IBM-CM) | Section 2.5 |
| interacting boson model (IBM) | Section 2.5 |

Monte Carlo shell model  (MCSM)                                          Section 2.1
Nilsson model                                                            Section 2.3
no-core shell model                                                      Section 2.1
O(6) symmetry                                                            Section 2.7
pairing interaction                                                      Section 2.3
pairing plus quadrupole (PPQ) model                                      Section 2.3
partial dynamical symmetry (PDS)                                         Section 2.5
proxy-SU(3) symmetry                                                   Section 2.4.4
quasi-SU(3) symmetry                                                   Section 2.4.3
pseudo-SU(3) symmetry                                                  Section 2.4.2
relativistic mean field (RMF)                                            Section 2.2
shape/phase transition (SPT)                                             Section 2.6
Skyrme interaction                                                       Section 2.2
spdf-IBM                                                                  Section 2.5
symmetry-adapted no-core shell model (SANCSM)                            Section 2.1
symplectic model                                                       Section 2.4.5
tensor force                                                           Section 2.1.1
time-dependent Hartree–Fock–Bogoliubov (TDHFB)                           Section 2.2
VAMPIR                                                                    Section 2.2
X(5) CPS                                                                  Section 2.6
Z(5) CPS                                                                  Section 2.6

*Appendix A.2. The Constrained HFB Theory*

The (HF) and (HFB) variational methods, mentioned in Section 2.2, determine only one point on the potential energy surface, namely the minimum. If somebody wishes to locate additional local minima, one has to introduce relevant constraints (i.e., subsidiary conditions) and look for a minimum obeying these additional restrictions (see Section 7.6 of [111]). For an application of configuration-constrained calculations to SC, see [575].

*Appendix A.3. The Cranking Model and the Routhians*

The cranking model, introduced by Inglis in 1954 [861,862] (see also Section 3.4 of [111]), in its initial form describes nuclear motion in a rotating potential well. However, a similar formulation can be applied to more general collective motions [863]; see, for example, [24,25,618,789] for applications related to SC.

The single particle energies in the rotating system are called Routhians (see Section 12.1 of [104]). Total-Routhian-surface (TRS) calculations related to SC have been performed, for example, in [574,686].

*Appendix A.4. The Deformed Woods–Saxon Potential*

The deformed Woods–Saxon potential [864] is a generalization of the spherical Woods–Saxon potential, in which the distance of a point from the nuclear surface is specified by a set of shape parameters and is generated numerically [865]. Applications of the deformed Woods–Saxon potential related to SC can be found in [25,572,573,575,578].

*Appendix A.5. The Nilsson–Strutinsky Model*

In phenomenological models, like the liquid drop model of vibrations and rotations (see Appendix 6A of [249]), the distribution of nucleons in phase space is supposed to be homogeneous, which is not the case in the nuclear shell model, in which the distribution becomes inhomogeneous. Strutinsky's method ([866], see also Section 2.9 of [111]) allows for calculations of the shell-model corrections to the liquid drop energy of the nucleus, starting from Nilsson's level scheme [104,250,251], which is a simplified deformed shell model. The corrections depend on the occupation number and on the deformation. The Nilsson–Strutinsky model has been used for the description of SC, for example in [572,577,789,790].

*Appendix A.6. The Particle-Plus-Rotor Model*

In order to describe the interplay between the independent motion of individual particles and the collective motion of the nucleus as a whole, Bohr and Mottelson in 1953 ([247], see also Section 3.3 of [111]) suggested to describe the nucleus in terms of a few valence particles, which move more or less independently in the deformed well of the core, and couple them to the collective rotor, which represents the rest of the particles. This particle-plus-rotor or particle-core model has been used for the description of SC for example in [576,600].

*Appendix A.7. The Projection Method*

The projection method is a method of restoring broken symmetries. It is used in cases in which one has a Hamiltonian possessing a certain symmetry and a wave function of this Hamiltonian violating this symmetry. It is a variational procedure which minimizes the energy and at the same time produces a wave function satisfying the symmetry of the Hamiltonian [111]. For details on particle number projection and angular momentum projection one may see Section 11.4 of the book [111]. For example, spin and number projection has been used in the excited VAMPIR [197] approach.

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
