# Peer review of "Shape Coexistence in Even–Even Nuclei: A Theoretical Overview"

_atoms, doi:10.3390/atoms11090117_

Round 1
Reviewer 2 Report
I would like to commend the tremendous work done by the authors which realised a very good review of theoretical studies on shape coexistence in even-even nuclei split on methods as well as parts of the nuclide chart. The manuscript deserves to be published, and will certainly become a major reference in the field. The large dimension of the manuscript did not generate many mistakes. Therefore I have only a few mistakes to report and some references to be added if the authors would consider them worth to mention:
1) Page 7, lines 287-289. It would be better to mention the phenomenological Bohr model with general double-well potentials, sextic potential being one of them. Other studies in this direction with different double-well potentials are:
[96Zr] E. V. Mardyban, E. A. Kolganova, T. M. Shneidman, R. V. Jolos, and N. Pietralla, Phys. Rev. C 102, 034308 (2020);
[92-102Zr] E. V. Mardyban, E. A. Kolganova, T. M. Shneidman, and R. V. Jolos, Phys. Rev. C 105, 024321 (2022);
[114-124Te] R. Budaca, A. I. Budaca, Eur. Phys. J. Plus 136, 983 (2021);
where the nuclei described are also shown for a better integration into the reference list.
The paper
Koichi Sato, Nobuo Hinohara, Takashi Nakatsukasa, Masayuki Matsuo, Kenichi Matsuyanagi, Prog. Th. Phys. 123, 129–155 (2010)
also deserves to be mentioned as it gives a Bohr solution for a potential with prolate and oblate minima.
2) Page 10, lines 458, 488; page 11, line 496. The start of the sentence "A convenient way to visualize..." is exactly repeated in the three instances.
3)Sec. 2.6, For the close connection between shape phase transition and shape coexistence one could indicate Ref.[479] and [R. Budaca, A. I. Budaca, EPL, 123 (2018) 42001] which also has a very good application to the shape coexistence in 76Kr.
4) Page 11, line 501. The sentence is ended with a comma instead of a period. At this point it would be instructive to mention other critical point solutions such as:
X(3) - D. Bonatsos, D. Lenis, D. Petrellis, P.A. Terziev, I. Yigitoglu, Phys. Lett. B 632, 238 (2006)
Z(4) - D. Bonatsos, D. Lenis, D. Petrellis, P.A. Terziev, I. Yigitoglu, Phys. Lett. B 621, 102 (2005)
T(4) - Y. Zhang, F. Pan, Y.X. Liu, Y.A. Luo, J.P. Draayer, Phys. Rev. C 96, 034323 (2017)
X(4) - R. Budaca, A.I. Budaca, Phys. Lett. B 759, 349 (2016)
Fractional critical points - M.M. Hammad, Nucl. Phys. A 1011, 122203 (2021)
5) Page 12, lines 573, 574, 575. The characteristics of a beta vibration band are more nuanced than that. Ref. [266] provide some guidance criteria based on simplistic assumptions of the original Bohr picture with harmonic potentials in deformation variables. It disregards for example anharmonic vibrations in transitional nuclei, which would have distinct spectral signatures. The connecting B(E2) transition rates are actually small, comparing to the inband ones. This is more obvious in the IBM model as pointed out in the same Ref.[266]. Also, although E0 strength is supposed to be moderate between ground and beta bands, it cannot be used as a definite criterium for beta bands, because similar values can signal an altogether different mechanism (SC for example).
6) It would be opportune to add in Sec 2.8, few comments regarding the shape mixing, that is the distinction between SC with or without shape mixing.
7) Perhaps a small section or subsection could be devoted to the "two state mixing model" (See in Ref.[276]), which is a preferred tool of experimentalists to establish SC from experimental data. This model have been extensively used (papers of T. Fortune and many others).
8) Page 13, line 631, "SO-HO" instead of "SH-HO".
9) Page 17, line 817. I would say "hints" instead of "incomplete evidence".
10) Large portions of 4.1, 4.2, 4.3, and 4.4, have very obvious similar text, mainly because the same literature is cited. I suggest either combine these subsection into one or two (Os/W and Hf/Yb), or to make an effort to change more every one of the four subsections.
11) Page 20, line 994, and page 21, line 1031, and in other places. (SPT) notation was already defined, and one can just write "SPT" instead of "shape/phase transition (SPT)". A similar comment is valid for
"5-dimensional quadrupole collective Hamiltonian (5DQCH)" which can be replaced by "5DQCH" throughout the rest of the paper.
"IBM with configuration mixing (IBM-CM)" which can be replaced by just "IBM-CM" throughout the rest of the paper.
12) "Covariant density functional theory" appears very often and can be replaced by an easily recognizable "CDFT".
13) The above mentioned abbreviations must also be completed in the list on page 55.
14) If I am not mistaken, "PES" notion appears first on page 18, line 905. It should be spelled "Potential energy surface (PES)" for future reference. Maybe it is the right place to warn the reader about the danger of proposing SC based solely on PES. The right way is to check the many body wave function when available, i.e. its probability distribution in the deformation variables. Most often, these do not show any signs of shape coexistence as suggested by PES which lack additional information. For example a gamma-soft PES will actually correspond to an extended wave function distribution around maximal triaxiality. Some local SC minima in PES will be just swallowed by large shape fluctuations of the many-body wave function and many other inconsistencies can be expected when comparing the two objects.
15) Ref.[8], the nucleus symbol should not be as superscript.
16) Period after Refs.[15],[27].
17) Ref.[209]. It would be better to write the citation [JPS Conf. Proc. 6, 030093 (2015)] instead of the arXiv entry.
18) Ref.[675]. It would be better to write the citation [Phys. Lett. B 751, 348 (2015)] instead of the arXiv entry.
19) I do not know the actual quality of Figures 8, 10 and 13, but the neutron and proton drip lines are not very visible. Maybe it would be better to make them thicker.
Author Response
Please see the attachement.

Round 2
Reviewer 1 Report
After reading the reply I see that the authors have taken into account some of my comments raised in my first report.
Although I can understand the authors' reasons for not moving some sections of the manuscript to appendices, I don't feel fully comfortable with the presentation of the second part of the manuscript. I still believe that the ordering in sections 3-15 could be modified without a lot of work for the authors. I mean, what about explaining first the sections that you recommend to read, namely, 1,2,5.5,8.5,9.5,11-15, and next the other sections concerning the different isotopic chains? This option, which I suggest to the authors, probably would make the paper easier to read.
